# Explainable AI unravels sepsis heterogeneity via coagulation-inflammation profiles for prognosis and stratification

Li Zhu [1,2,14], Zengtian Chen [1,14], Hong Zhang [3,14], Hongjun Chen [1], Lanqi Liu[4], Wei Yu [5], Kai Wu [1], Yijin Chen [5], Xingyu Tao [2], Zefeng Yu[1], Linhui Shi[6], Jialian Wang[2], Fan Zhang[7], Jiaying Shen[5], Fen Liu [4,8], Chongke Hu [5], Yangguang Ren [9], Tzu-Ming Liu [10], Yang Luo [3,9,11,15] ✉, Fei Guo[5,15] ✉ & Bailin Niu [2,12,13,15] ✉

Sepsis is a leading cause of hospital mortality, and its significant heterogeneity complicates prognosis and stratification. To address this challenge, we developed an explainable artificial intelligence prognostic model (SepsisFormer, a transformer-based neural network) and an automated risk-stratification tool (SMART) for sepsis. In a multi-center retrospective study of 12,408 sepsis patients, SepsisFormer achieved high predictive accuracy (AUC: 0.9301, sensitivity: 0.9346, and specificity: 0.8312). SMART (AUC: 0.7360) surpassed most established scoring systems. Seven coagulation-inflammatory routine laboratory measurements and patient age were identified to classify patients' four risk levels (mild, moderate, severe, dangerous) and two subphenotypes (CIS1 and CIS2), each with distinct clinical characteristics and mortality rates. Notably, patients with moderate/severe levels or CIS2 derive more significant benefits from anticoagulant treatment. Our work, therefore, offers a set of simple, real-time executable tools for sepsis heterogeneity, demonstrating the potential to enhance sepsis clinical practice globally, particularly in resource-constrained healthcare settings.

Sepsis, a leading cause of hospital mortality, is a serious condition characterized by a heterogeneous syndrome and a dysregulated immune response[1]. Annually, approximately 49 million sepsis cases occur globally, with sepsis-related deaths constituting 19.7% of all deaths worldwide[2]. Sepsis heterogeneity complicates risk stratification, prognostic prediction, and subtyping, as diverse clinical presentations and heterogeneity of treatment effects (HTEs) hinder outcome improvement[3,4]. HTE refers to the phenomenon where the same treatment can have different effects (beneficial, neutral, or even harmful) on different patients. Identifying practical markers and developing tools to address this variability remain critical yet challenging tasks in advancing sepsis management[5]. Current approaches for measuring sepsis heterogeneity use unsupervised machine learning methods to identify subtypes and

subphenotypes[6–8]. Currently, four main subtype strategies have been established based on clinical data from electronic health records (EHRs, α, β, γ, and δ), biological pathway data (hyper- and hypo-inflammatory states), and transcriptomic data (Mars1–Mars4 and SRS1–SRS2 classifications). Various subtype strategies use markers that fail to consistently identify homogeneous patient groups. This indicates fundamental heterogeneity in clinical presentations and biological responses[8].

Biomarker, clinical, and transcriptomic data in sepsis reflect infection, dysregulated host response, or treatment effects. However, the precise therapeutic role of biomarkers remains unclear[9]. Analysis of 5367 studies identified 258 sepsis biomarkers from multi-omics data, including complement components, cytokines, chemokines, noncoding RNAs, miRNAs, and cell proteins, underscoring sepsis's

---

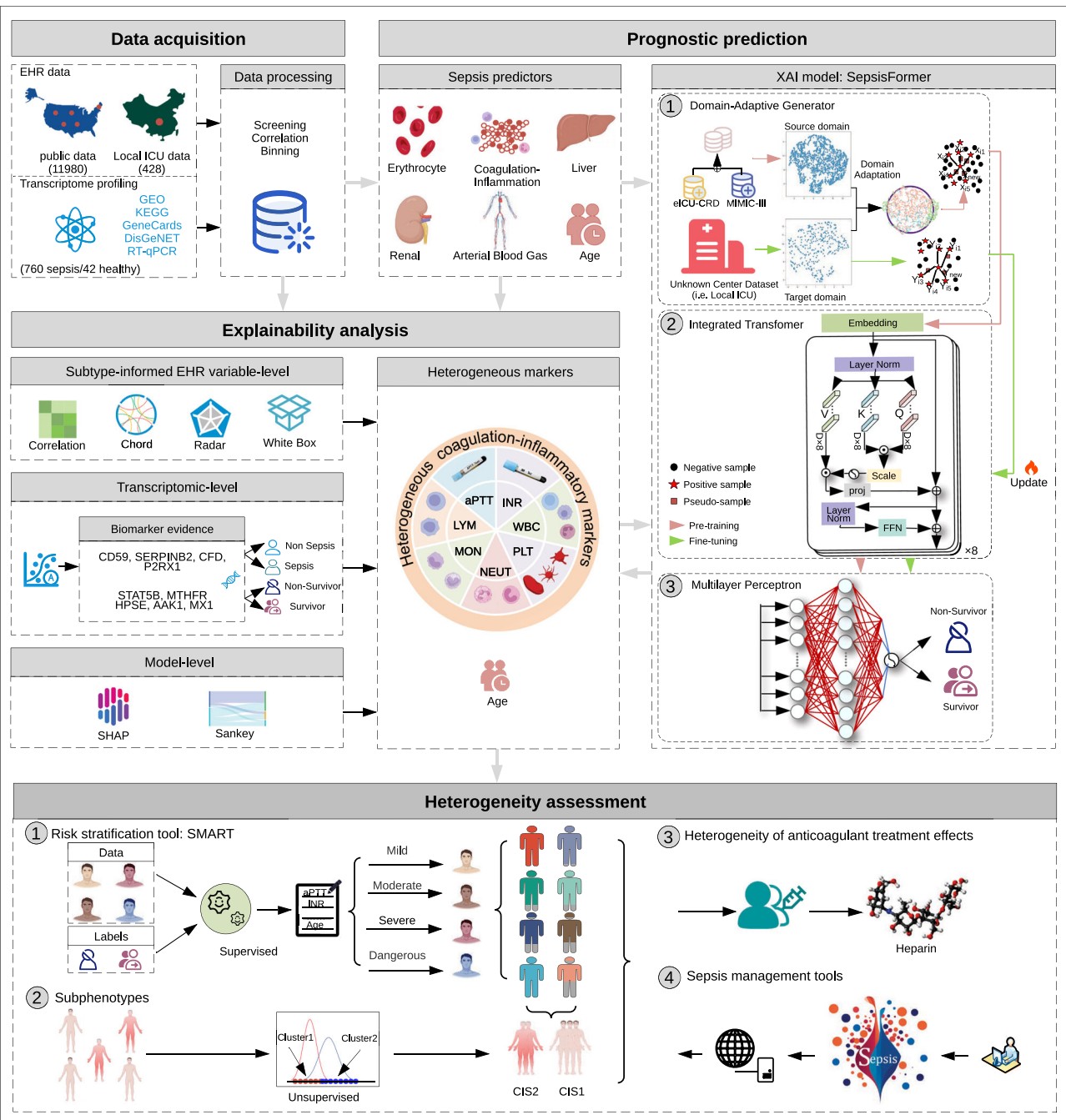

**Fig. 1 | Comprehensive framework of our study.** Data acquisition: three public cohorts (EHRs) from MIMIC-III, MIMIC-IV, and eICU-CRD, a local cohort from the First Affiliated Hospital of Chongqing Medical University, and transcriptomic data from the Ningbo Medical Center Lihuili Hospital, GEO, KEGG, GeneCards, and DisGeNET were used for model development. Prognostic prediction: predictors derived from medical practice and laboratory tests were used for model development. SepsisFormer consisted of a domain-adaptive generator that dealt with multi-center data distribution differences and class imbalance, an integrated transformer that extracted information from the EHRs of sepsis patients into fixed-length semantic vectors, and a multilayer perceptron that make predictions. Explainability analysis: the efficiency of heterogeneous markers, including seven coagulation-inflammatory markers and patient age, was explained through EHR-level, transcriptomic-level, and model-level analysis. Heterogeneity assessment: 1, an automated risk stratification tool (SMART), assessed the risk level of sepsis patients based on eight routine medical measurements. 2, Subphenotypes classified sepsis patients as CIS1 and CIS2 via unsupervised methods. 3, Heterogeneity of anticoagulant treatment (e.g., Heparin) effects was evaluated in subgroups with distinct subphenotypes and risk levels. 4, Sepsis management tools. Created in BioRender. Niu, B. (2025) https://BioRender.com/8py5ibd. EHR Electronic health record, XAI Explainable artificial intelligence, CIS1 Coagulation-inflammatory subphenotype 1, and CIS2 Coagulation–inflammatory subphenotype 2.

complex pathophysiology[10,11]. Blood-based biomarkers like gene expression profiles and routine tests show promise for diagnosis and prognosis. For example, altered expression of immune and inflammatory genes (e.g., CD59, SERPINB2, LPIN1) correlates with disease stratification. However, transcriptomic methods face challenges such as cost, time, and host variability, while routine clinical tests (e.g., activated partial thromboplastin time (APTT), platelet count (PLT), international normalized ratio (INR), and white blood cell count (WBC)) offer fast, affordable, and practical alternatives for assessing sepsis heterogeneity[11,12].

Artificial intelligence (AI) prediction models and prognostic warning score systems have transformed sepsis care and management with advanced prediction and intervention capabilities[13,14], demonstrating high diagnostic accuracy in the ICU[15], accurate prediction prior to sepsis onset[16], and even the potential to optimize antibiotic stewardship through HTE estimation[13]. A total of 256 AI-based sepsis prediction models from 73 studies (2016–2023, $n = 457,932$) showed a pooled AUC of 0.825 (95% CI: 0.809–0.840)[15]. Models mainly include machine learning approaches (Decision Tree, Logistic Regression, Support Vector Machine, Generalized Linear Model, Naïve Bayes) and neural networks (Multilayer Perceptron, Long Short-term Memory, Convolutional Neural Network, Gated Recurrent Unit, and two attention-based explainable models: RETAIN and Dipole). Public datasets (e.g., MIMIC-III/IV, eICU, Computing in Cardiology) were used in 53% of studies. However, only 21.9% performed external validation; data-sharing transparency was critically limited—only three studies disclosed data, and no studies released code[17]. Meanwhile, although Transformer-based models (e.g., RETAIN, BEHRT, Med-BERT)[18] have achieved strong performance in EHR-driven disease risk prediction, their use in sepsis remains relatively limited. On the other hand, in clinical practice, several well-established prognostic warning score systems (SOFA[1], APACHE II, LODS[19], qSOFA[1], SIRS[1]) remain benchmarks. In an analysis of 148,907 EHRs of suspected infection cases, the area under the receiver operating characteristic curve (AUC) for patients admitted to the ICU ranged from 0.64 to 0.75 for existing scoring systems[14]. In our previous study, we developed the LIP scoring system, which incorporates lymphocyte count, INR, and procalcitonin as a simple sepsis screening tool, achieving 92.8% sensitivity and 94.1% specificity[20]. The LIP tool is well-suited for rapid clinical screening and is particularly beneficial in resource-limited settings.

Although progress has been made, some limitations still exist: (1) These traditional prognostic warning score systems have inherent deficiencies, including a lack of refined risk stratification, uncertain applicability across various patient subgroups, and insufficiently validated efficacy in improving treatment outcomes[21]. (2) Despite demonstrating superior performance, AI models encounter two primary challenges in clinical application: constrained predictive performance and generalization capability, largely attributable to class imbalance and multi-center data heterogeneity. The presence of class imbalance in mortality outcomes among sepsis patients is a widely held consensus, substantiated by empirical analyses of clinical data[4,5,8]. Concurrently, the "black-box" nature of the model impedes clinicians comprehension and trust in their decision-making.

In this study, we developed a Transformer-based prognostic model (SepsisFormer) interpreted via post-hoc XAI techniques, an automated risk stratification system (Sepsis Mortality and Risk Tool, SMART), and identified two distinct subphenotypes (CIS1/CIS2). An open-access sepsis risk assessment platform (http://smartsepsis.org.cn) was established to provide real-time outputs of risk levels (Mild, Moderate, Severe, Dangerous) and subphenotypes. Both SMART and subphenotyping require only patient age and seven routine coagulation-inflammatory markers. These markers were selected and validated using multi-view explainability analyses across EHR variables, models, and transcriptomic levels. SepsisFormer's prognostic performance was comprehensively benchmarked against a wide range of machine learning and deep learning models. In parallel, SMART's risk stratification capability was systematically evaluated in comparison with established clinical scoring systems. To investigate the influence of SMART-derived risk levels and the identified CIS1/CIS2 subphenotypes on clinical outcomes, we conducted further heterogeneous treatment effects analyses, specifically examining the efficacy of heparin anticoagulation. Overall, this work, by combining XAI and coagulation-inflammatory markers, deeply explores sepsis heterogeneity and develops high-performance, real-time tools for clinical practice.

## Results

The development of SepsisFormer and SMART were illustrated in Fig.1 and Supplementary Fig. 1. The performance of SepsisFormer was assessed in an extensive, multi-center retrospective cohort study using EHR data collected from 12,408 septic patients across our local ICU, the Medical Information Mart for Intensive Care III/IV Database (MIMIC-III/IV), and the eICU Collaborative Research Database (eICU-CRD). Eight markers (APTT, INR, lymphocyte, monocyte, neutrophil, WBC, PLT counts, and patient age) were identified and validated to delineate risk stratification (mild, moderate, severe, and dangerous) and sepsis subphenotypes (CIS1 and CIS2) (Supplementary Fig. 2). SMART can automatically assess the risk level of septic patients. We tested the effects of anticoagulant therapy across patient subgroups stratified by our model.

### Performance of SepsisFormer based on sepsis predictors

SepsisFormer is trained with 36 sepsis-related predictors, including patient age and 35 objective routine laboratory measurements, derived from Sepsis-3 criteria (SOFA based)[14] criteria for organ dysfunction assessment (e.g., respiratory, hepatic, renal, coagulation), while exclusion for subjective factors like GCS score and respiratory rate. These 35 indicators enable targeted evaluation of infection-related organ injury and inflammation dynamics, thereby supporting reliable sepsis diagnosis and prognostic applications. Firstly, the performance of SepsisFormer was compared with five machine learning approaches and three state-of-the-art deep learning models (Fig. 2a, b and Supplementary Table 1). SepsisFormer demonstrated cross-database generalizability, achieving superior predictive performance (AUC: 0.9301; sensitivity: 0.9346; and specificity: 0.8312). The deep learning-based models (AUC: 0.9109–0.9301) achieved higher prediction performance than their machine-learning counterparts (AUC: 0.7761–0.9067). The hyperparameters used for training SepsisFormer include a learning rate of 0.0010, a batch size of 5000, a dropout rate of 0.1000, 1400 training epochs, eight parallel self-attention heads, and an eight integrated Transformer architecture (Supplementary Table 15). Secondly, predictor interrelationships of all 36 predictors were explained via a correlation network diagram, with Pearson correlation coefficients and corresponding $p$ values (Fig. 2c). Then, statistically significant differences were observed for most predictors between survivors and non-survivors across multi-center cohorts (Supplementary Tables 2 and 3).

### Explainability analysis of coagulation-inflammatory dysfunction

The 35 laboratory measurements collectively capture multi-organ dysfunction spanning five categories: coagulation-inflammatory, hepatic, renal, blood gas, and oxygen transport (Erythrocyte), while essentially conforming to the connotation of Sepsis-3 criteria (excluding neurological markers). However, their full clinical adoption is hindered by prohibitive costs, large blood volume requirements, and implementation barriers in resource-constrained settings. To overcome these challenges, explainability analyses were performed across five categories to achieve two clinical goals: (a) select an efficient unsupervised clustering method to uncover clinically meaningful sepsis subphenotypes, essential for understanding disease heterogeneity and guiding clinical insights[4]; (b) identify a sepsis mechanism relevant and clinically feasible subset of laboratory measurements within one category for real-time and cost-effective application.

We explained the effectiveness of coagulation-inflammatory markers in predicting sepsis outcomes through multi-view explainability analyses of cluster-informed EHR variables, model, and transcriptomics. We identified coagulation-inflammation-related variables that are essential for sepsis subtyping. Exploratory clusters α and β were automatically derived using unsupervised clustering methods using 36 sepsis predictors, requiring no prior information (e.g., mortality, disease outcomes, or treatment medications), based on the

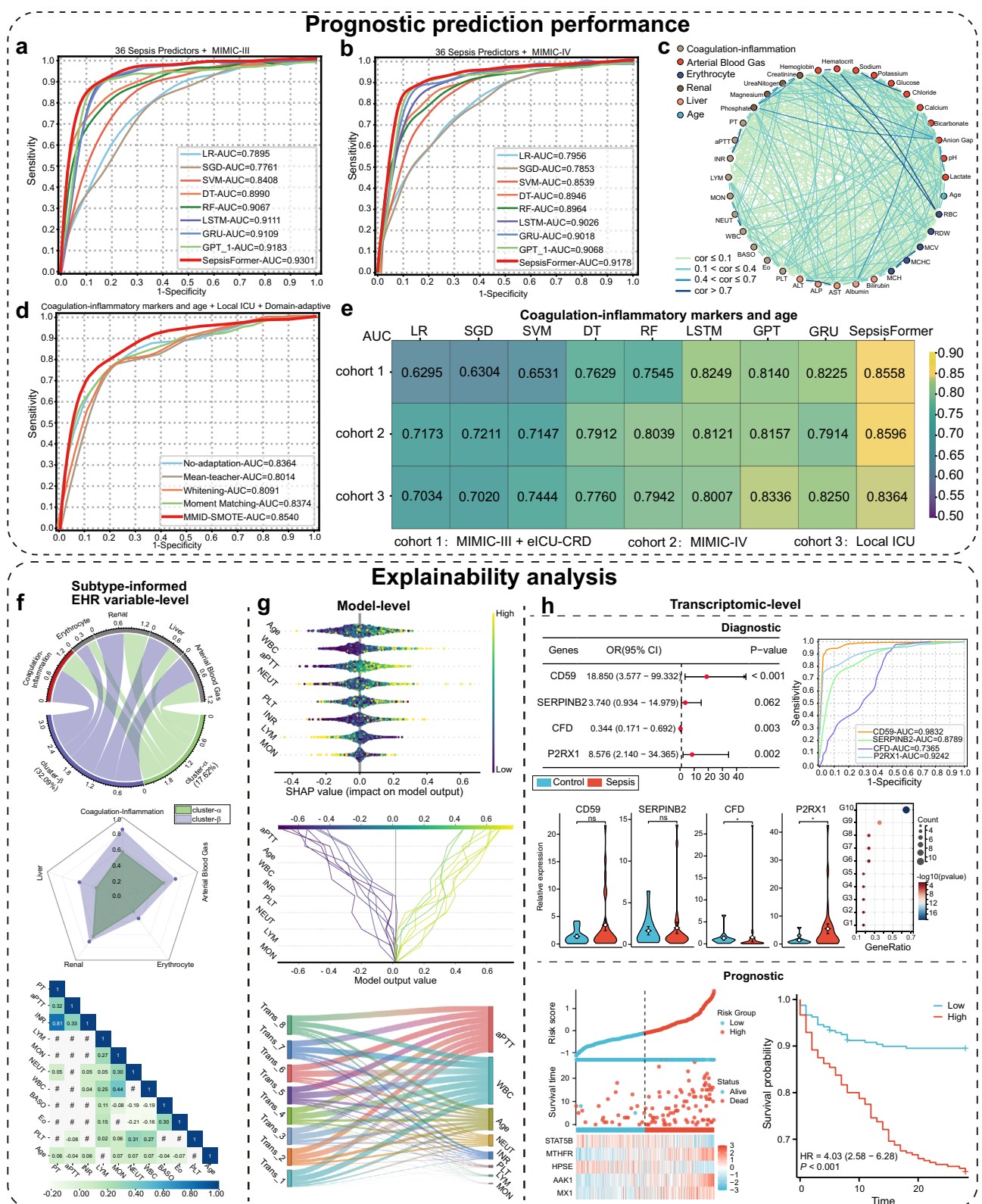

optimal number of clusters (Supplementary Table 4). Five different unsupervised methods—Gaussian Mixture Model (GMM), Mini-BatchKMeans, K-means, Hierarchical Agglomerative Clustering (HAC), and Birch—generated highly aligned clusters, a consistency across distinct approaches that confirms the robustness and validity of the two identified clusters. Since these data-driven clusters lack inherent clinical interpretation, we statistically analyzed subgroup mortality. The GMM-derived α and β clusters exhibited the most significant

mortality difference (32.09% vs 17.62%, respectively), indicating GMM's superior effectiveness in identifying patient subgroups with divergent mortality risks. In the chord diagram, both cluster α and cluster β consistently showed the widest chords with the coagulation-inflammatory category (ribbons connect with these portions of the circle), indicating coagulation-inflammatory predictors are key defining features for the clustering. Independently, the radar plot also confirmed the coagulation-inflammatory category as the highest

**Fig. 2 | Prognostic prediction performance of SepsisFormer and explainability analysis. a, b** ROC curves show superior prognostic prediction by SepsisFormer using 36 predictors compared to baseline models on MIMIC-III (all $p < 0.01$, except LSTM, $p = 0.08$), and MIMIC-IV (all $p < 0.01$, except GRU, $p = 0.41$; DeLong's test). **c** Network diagram of predictor correlations; edges indicate significant associations ($p < 0.05$, Pearson's correlation, two-sided test). **d** ROC curves comparing MMID-SMOTE with domain adaptation approaches and the no-adaptation baseline using coagulation-inflammatory markers from local ICU. **e** AUC values confirm better performance of SepsisFormer across Cohort 1 (MIMIC-III/eICU-CRD, 7789 patients; all $p < 0.01$), Cohort 2 (MIMIC-IV, 4191 patients; all $p < 0.01$, except LSTM, $p = 0.91$), and Cohort 3 (local ICU, 428 patients; all $p < 0.05$, except GPT, $p = 0.16$, and RF, $p = 0.60$; DeLong's test). **f** Cluster-informed EHR variable-level explainability: chord/radar diagrams show category contributions; Pearson's correlation matrix highlights significant predictor relationships. Clustering optimized via Silhouette, Calinski–Harabasz, and Davies–Bouldin indices. **g** Model-level explainability: SHAP values show feature importance; decision plot visualizes individual prediction

contributions; Sankey diagram shows the cumulative overlap ordering across transformer layers. **h** Transcriptomic explainability: forest plot of diagnostic DEGs; ROC curves demonstrate the performance of the diagnostic markers; violin plots of real-time PCR validate the diagnostic genes. Results are expressed as mean ± SD and analyzed by two-tailed t-tests (sepsis: $n = 29$; healthy controls: $n = 11$; biologically independent samples).KEGG enrichment of DECRGs/DEIRGs; risk group distributions, heatmaps, and Kaplan–Meier survival analysis revealed a distinct separation between high- and low-risk groups. *$p < 0.05$; ns not significant, LR logistic regression, SGD stochastic gradient descent, SVM support vector machine, DT decision tree, RF random forest, LSTM long short-term memory, GRU gated recurrent unit, GPT generative pretrained Transformer, RBC red blood cell, RDW red blood cell distribution width, AST aminotransferase, ALP alkaline phosphatase, ALT alanine aminotransferase, PLT platelet count, Eo eosinophil, BASO basophil, WBC white blood cell, NEUT neutrophil, MON monocyte, LYM lymphocyte, MMID max-min interval discrepancy, GMM Gaussian mixture model, OR odds ratio and HR hazard ratio.

connection point. These findings were robustly replicated across all five unsupervised clustering methods (Fig. 2f and Supplementary Fig. 3).

Pearson's correlation matrix showed the interdependencies and statistically significant differences among coagulation-inflammatory predictors and patient age (Fig. 2f and Supplementary Fig. 4). A notable correlation was observed among the INR, prothrombin time (PT), and APTT, particularly underscored by a substantial correlation of 0.81 between the INR and PT. This correlation may be attributed to the fact that the INR and PT are essentially the same, but the INR, a standardized form of PT, is comparable across different laboratories. Consistently, systematic studies have demonstrated that INR and APTT are reliable predictors for preoperative coagulation screening and rapid sepsis prognostic prediction. Due to PT tests extrinsic, APTT intrinsic coagulation pathways and INR is a standardized value calculated from PT results, we retained INR and APTT while excluding PT. Meanwhile, basophils and eosinophils exhibit weak or negative correlation coefficients with other predictors, rendering them impractical to reflect the risk status of patients. These two types of WBC, influenced by external factors like allergies, display inconsistent behavior and limited prognostic value in sepsis, with their mechanisms remain unclear[22,23]. To focus our analysis on the most robust and clinically relevant markers, we excluded PT, basophils, and eosinophils. The final coagulation-inflammatory markers analyzed in this study comprise 8 variables: APTT, INR, lymphocytes, monocytes, neutrophils, WBC, PLT, and age.

At the model-level explanation scope, the SHapley Additive exPlanations (SHAP) analysis indicated that APTT, WBC, and patient age are important predictors of sepsis outcomes, regardless of the perspective (global, local, or hierarchical cumulative contribution). These factors significantly influence whether a patient's condition will deteriorate or improve, providing valuable guidance for clinicians to focus on these key indicators during patient assessment. The SHAP summary plot further quantified the contributions of predictors such as APTT to the prediction outcomes, highlighting the importance of predictor ranking in model prediction (Fig. 2g). The decision plot, with predicted values below −0.40 or above 0.40, showed how the contributions of predictors have different impacts on sepsis outcomes in specific patient populations. The Sankey diagram shows that as Transformer depth increases, the cumulative contributions of the eight predictors slightly increase yet remain generally stable, indicating consistent feature integration across layers. The results consistently indicated that coagulation-inflammatory indicators such as APTT, WBC, and the patient's age were important predictors from various perspectives. To optimize the model architecture, we conducted a systematic hyperparameter sensitivity analysis on the number of Transformer layers $L$ and attention heads $H$. We chose $L = 8$ to balance complexity and feature extraction, despite peak performance

at $L = 1 \sim 7$. With M fixed at 8, we observed a similar pattern for attention heads, with optimal performance at $H = 1 \sim 7$ before a decline at $H = 8$ (Supplementary Fig. 5). Furthermore, the consistent ranking of predictor importance underscores the model's structural robustness and deterministic nature. These findings collectively affirm the model's reliability and interpretability in multivariate prediction tasks. Therefore, compared with the functional indicators of specific organs such as the liver, kidneys, and arterial blood gas, the coagulation-inflammatory indicators reflect the systemic or general state and can better reflect the systemic pathophysiological connotation of sepsis. They are also an important basis for reflecting and inducing functional disorders in other organs[24,25].

Our transcriptomic-level explanation provides complementary evidence for the critical role of coagulation-inflammatory markers in sepsis heterogeneity, initially identified through EHR-based variable and model explanations. We retrieved the sepsis expression profile from the Gene Expression Omnibus (GEO, https://www.ncbi.nlm.nih.gov/geo/), selecting dataset GSE65682 for analysis. Disseminated intravascular coagulation (DIC)-related genes were sourced from the GeneCards (https://www.genecards.org/) and DisGeNET (https://www.disgenet.org/) databases. Genes overlapping between these DIC-related gene sets and differentially expressed genes (DEGs) were defined as DIC-related DEGs. To clarify their core biological functions and underlying mechanisms, we performed Kyoto Encyclopedia of Genes and Genomes (KEGG) and GO enrichment analyses on these genes using the clusterProfiler package (version 3.14.3) in R. We specifically examined coagulation-related genes (CRGs), inflammatory-related genes (IRGs), and DIC-related genes (Fig. 2h; Supplementary Figs. 6 and 7). This investigation reaffirmed the significance of these markers and identified five key prognostic biomarkers: STAT5B, MTHFR, HPSE, AAK1, and MX1. Patients stratified into a high-risk group based on these biomarkers exhibited significantly higher mortality. Notably, MTHFR, AAK1, and MX1 expression levels were elevated in this high-risk group, suggesting their influence on sepsis prognosis potentially through modulation of immune and coagulation pathways (Supplementary Table 8).The diagnostic efficacy of these five genes was further validated in the external dataset GSE54514 (Supplementary Fig. 8). Furthermore, transcriptome analysis confirmed the diagnostic relevance of four genes: CD59, P2RX1, CFD, and SERPINB2 (Supplementary Tables 9 and 10). These markers demonstrated robust performance as independent diagnostic biomarkers for sepsis. Their diagnostic efficacy was successfully validated in external datasets GSE26440 and GSE95233 (Supplementary Fig. 9). RT-PCR analysis of PBMCs from 29 sepsis patients and 11 healthy controls revealed significant differential expression of CFD and P2RX1 ($p < 0.05$, Fig. 2h), with greater variability observed in the sepsis group. Therefore, these transcriptomic findings further explained the important roles of

coagulation-inflammation related indicators in the diagnosis and prognosis of sepsis from multiple perspectives.

## Prognostic prediction performance of SepsisFormer based on coagulation-inflammatory markers

SepsisFormer used seven coagulation-inflammatory markers and age demonstrated high prognostic prediction capabilities. Cohort 1 included 7789 septic patients from MIMIC-III and eICU-CRD, while two external cohorts comprised 4191 from MIMIC-IV (Cohort 2) and 428 from a local ICU (Cohort 3). As previously noted, SepsisFormer demonstrated strong prognostic performance for sepsis using 36 predictors. Using only seven coagulation-inflammatory biomarkers and age, SepsisFormer achieved high prognostic performance, with AUCs of 0.8558 in internal testing (in Cohort 1, specificity: 0.7398; sensitivity: 0.9264) and 0.8596/0.8364 in external validations (Cohorts 2/3). DeLong's test confirmed its superiority over all baseline models in Cohort 1 (all $p < 0.01$) and most baseline models in the external cohorts, demonstrating the model's robustness and generalizability (Fig. 2e; Supplementary Tables 5 and 6).

Despite strong performance within each cohort, SepsisFormer's ability to generalize across different clinical settings remains a key challenge, as patient characteristics can differ significantly between datasets. To address this issue, we incorporated methods to enhance the model's adaptability, allowing it to adjust to these differences and maintain high accuracy in new settings.

As shown in Fig. 2d and Supplementary Table 7, Maximum and Minimum Interval Difference-based Synthetic Minority Oversampling Technique (MMID-SMOTE) significantly outperformed other domain adaptation techniques and the ablation experiment (no-adaptation baseline) by substantially improving the model's cross-cohort generalization capability. MMID-SMOTE performed better than other approaches across multiple evaluation metrics, including AUC, accuracy, sensitivity, specificity, and F1-score. This is demonstrated through an ablation study, which shows that MMID-SMOTE outperformed the no-adaptation baseline and other state-of-the-art methods. The Mean-teacher and Whitening methods showed poorer performance, with Mean-teacher struggling with low-quality pseudo-labels and Whitening's adjustments failing to improve generalization. In contrast, Moment Matching showed second-place performance by focusing on second-order statistics, but still did not surpass MMID-SMOTE in any metric. This highlights the importance of selecting the right method for improving model performance across diverse clinical settings. Our findings show that MMID-SMOTE improves SepsisFormer's ability to predict outcomes and adapt to new clinical environments. As a post-hoc XAI model, SepsisFormer offers reliable and interpretable support for clinical decision-making, utilizing a minimal set of routine biomarkers.

## Subphenotypic heterogeneity analysis

To explore the heterogeneity of septic patients, we identified two subphenotypes, CIS1 and CIS2 (Fig. 3a–c). This study employs GMM for subphenotype identification, as the patient subgroups derived from GMM show the most significant differences in mortality rates (shown in Section "Explainability Analysis"), indicating greater clinical relevance. The optimal number of clusters, 2, was determined according to the silhouette, Davies–Bouldin, and Calinski–Harabasz scores (Fig. 3b and Supplementary Table 11). The dimensionality reduction of each subphenotype across all cohorts is shown in Fig. 3a. Clinical outcomes and characteristics differed between the two subphenotypes. Compared to CIS1, CIS2 exhibited higher mortality rates (mean mortality rates: 27.94 and 21.65%, $p < 0.001$), longer APTT and greater INR, increased WBC, lymphocyte, monocyte, and neutrophil counts, and lower PLT counts (Fig. 3c and Supplementary Table 12), and a higher systemic inflammatory response index (SIRI) (Supplementary Table 13, $p < 0.001$). Specifically, the mortality rates of the CIS2 and CIS1 were

27.89% and 18.70% (MIMIC-III), 32.23% and 25.84% (MIMIC-IV), 24.20% and 19.38% (eICU-CRD), and 33.87% and 26.23% (Local ICU), respectively.

## SMART scoring system based on coagulation-inflammatory markers

The proposed automated risk stratification tool SMART achieved comparable performance to established clinical criteria associated with sepsis (Fig. 3d, f). In the local ICU cohort, the SMART demonstrated the highest predictive accuracy, with an AUC of 0.7360. For the five established scoring systems, including SOFA, qSOFA, LIP, APACHE II, and SIRS, the AUCs are 0.6833, 0.6441, 0.6431, 0.6222, and 0.5428 respectively (Fig. 3d). Similarly, a validated Sepsis-3 study evaluated the clinical criteria of 7932 patients with suspected or documented infection in the validation cohort and reported a similar range of AUCs, from 0.66 to 0.75[14]. The AUC of SMART was also superior to that of other scoring systems for large datasets, such as MIMIC-III (SMART: 0.6751; SOFA: 0.661, qSOFA: 0.558, and LODS: 0.668[26]), MIMIC-IV (SMART: 0.6596 and SOFA: 0.606[27]), and eICU-CRD (SMART: 0.6475 and SOFA: 0.680[28]).

A clinically relevant scorecard (Table 1) was developed for SMART based on medical knowledge, allowing clinicians to easily and directly calculate a patient's risk score. Sepsis was classified into four risk-stratified levels that significantly reflected patient heterogeneity. Across all cohorts, the mortality rate stably exhibited a clinically meaningful and statistically significant increase (all cohorts $p < 0.001$) with increasing risk levels of approximately 5, 15, 30, and 50% (Fig. 3e). This mortality-risk level relationship has clear clinical significance; for example, in intensive care unit settings, a real-time predicted mortality rate can guide treatment intensity and resource allocation. The distribution of overall scores at the four risk levels correlates consistently with the distribution of the patient's risk level. Furthermore, the overall score distribution, the distribution of scores for each marker, and the distribution of clinical values exhibited a clinically reasonable correlation (Fig. 3g). We determined this clinical reasonableness by comparing it with known pathophysiological mechanisms. For instance, coagulation-inflammatory markers such as lymphocyte count and PLT count showed monotonic changes with increasing risk levels. The lymphocyte count and PLT count monotonically decreased with increasing risk levels (lymphocyte count, $p < 0.01$; others, $p < 0.001$; Fig. 3h), while the remaining markers increased monotonically. In clinical sepsis cases, a decrease in lymphocyte counts and platelet counts is associated with disease progression, reflecting the body's deteriorating immune and coagulation functions. Moreover, the patient's SIRI increased with risk level (Supplementary Table 13, $p < 0.001$), indicating that patients with higher risk levels may present with more severe inflammatory and coagulation syndromes. This aligns with clinical observations where more severe sepsis cases are often accompanied by exacerbated inflammatory and coagulation disorders.

## Assessment of HTEs according to subphenotype and risk level

Subphenotype identification and risk stratification are promising approaches for addressing heterogeneity[3]. A total of 4191 septic patients from the MIMIC-IV were included in the study to assess HTEs associated with anticoagulant drugs, such as heparin, across the subphenotypes and risk levels classified by our findings. Among these patients, 946 received anticoagulant treatment for three consecutive days, while 3245 served as controls.

Significant heterogeneity was observed in clinical characteristics, mortality rates, and anticoagulant treatment effects. The Kaplan–Meier curves (Fig. 4a) revealed that anticoagulant treatment was associated with a significant reduction in 28-day mortality at the moderate ($p < 0.001$) and severe ($p < 0.001$) risk levels. In contrast, no statistically significant difference was found between the mild

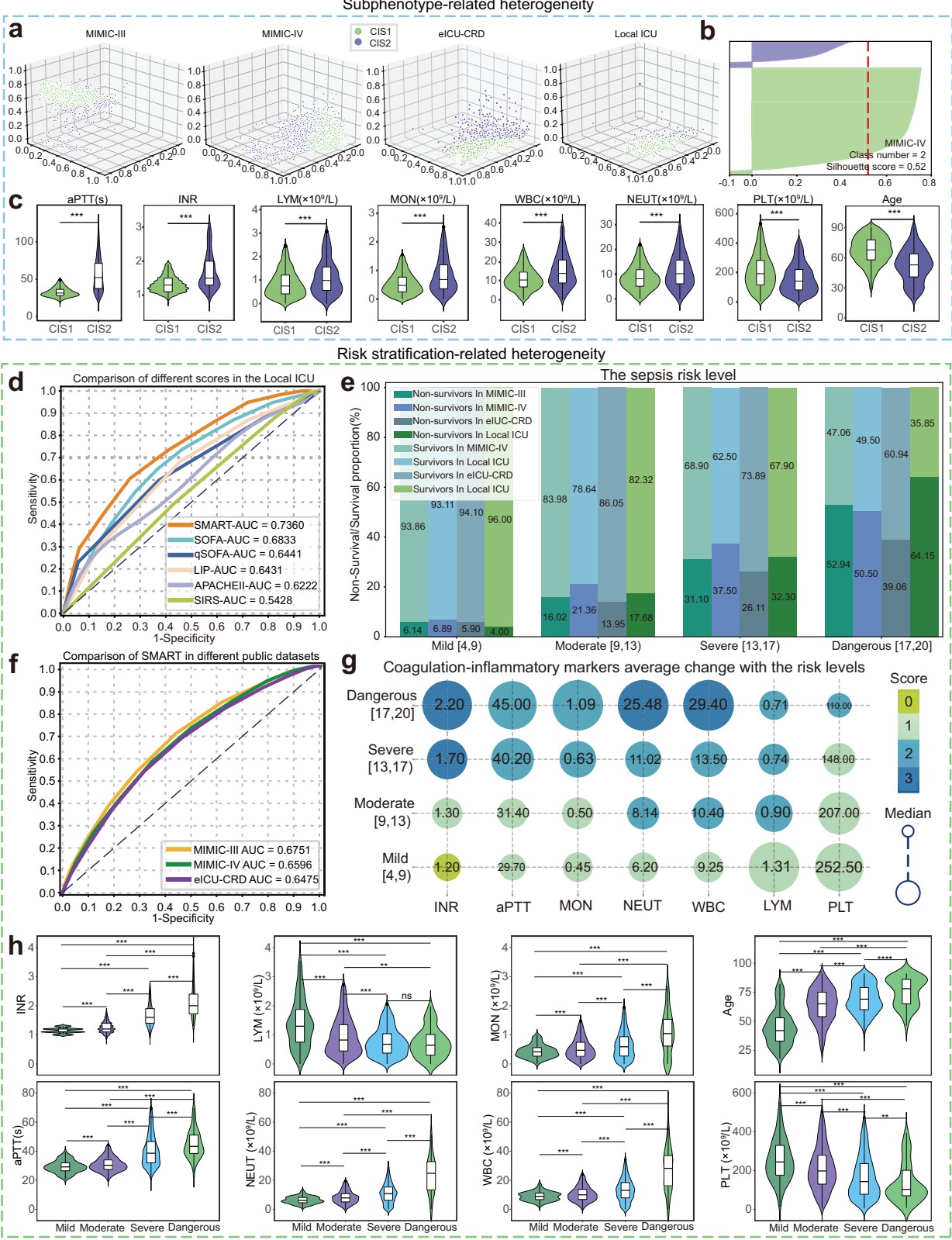

(p = 0.899) and dangerous (p = 0.052) levels. Specifically, mild-level did not demonstrate a significant survival benefit from anticoagulant treatment (hazard ratio (HR): 0.95, 95% confidence interval (CI): 0.40–2.24, p = 0.90). Moderate-level (HR: 0.60, 95% CI: 0.48–0.75, p < 0.005) and severe-level (HR: 0.57, 95% CI: 0.46–0.70, p < 0.005) patients demonstrated a significant survival benefit from anticoagulant treatment, with significant reductions in mortality (19.96% vs.

21.80% and 32.5% vs. 38.73%, respectively) and prolongation of survival time (10.28 and 10.76, respectively). However, it is statistically significant that patients at the danger level had a lower hazard ratio (HR) (0.33, 95% CI 0.10–1.07, p = 0.07). These findings are consistent with those of a previous study[29]. As shown in Fig. 4b, the radar plot demonstrates that risk stratification based solely on coagulation-inflammatory markers captures a broader pattern of multi-organ

**Fig. 3 | Subphenotype-related heterogeneity and risk stratification-related heterogeneity in sepsis. a** CIS1 and CIS2 subphenotypes across all cohorts are visualized via FastICA. **b** Silhouette plot depicting the optimal number of clusters on the basis of silhouette scores. **c** Violin diagrams showing the significant difference in the distributions of coagulation-inflammatory markers between CIS1 ($n = 3347$) and CIS2 ($n = 844$) from MIMIC IV. **d** ROC curves illustrating the SMART and established scoring systems for the local ICU. **e** Stacked bar charts showing the proportion of patients stratified by risk level across all four cohorts. $\chi^2$ tests for differences in patient proportions across risk levels and cohorts. **f** ROC curves of SMART in the MIMIC III/IV and eICU-CRD cohorts. **g** Bubble chart representing the relationship between the mean value of each coagulation-inflammatory marker and stratified risk scores and levels. **h** Violin diagrams illustrating the difference in distribution of coagulation-inflammatory markers among mild ($n = 334$), moderate ($n = 2140$), severe ($n = 1616$), and dangerous ($n = 101$) risk levels. **c** and **h** Violin diagrams show data distributions with embedded box plots (median, 25th/75th percentiles, whiskers extending to minimum and maximum values). Significances were assessed by two-tailed Mann–Whitney U tests. **$p < 0.01$; ***$p < 0.001$; ****$p < 0.0001$; ns not significant, CIS1 coagulation-inflammatory subphenotype 1, CIS2 Coagulation-inflammatory subphenotype 2, LYM lymphocyte, MON monocyte, WBC white blood cell, NEUT neutrophil, PLT platelet count, SMART sepsis mortality and risk tool, SOFA sequential organ failure assessment, qSOFA quick sequential organ failure assessment, LIP lymphocyte count, international normalized ratio, PCT level, APACHE II acute physiology and chronic health evaluation II, and SIRS systemic inflammatory response syndrome.

## Table 1 | SMART scoring system

| Markers | Point |
|---|---|
| APTT (s) | |
| <24.0 | 0.0 |
| <37.0 | 1.0 |
| ≥37.0 | 2.0 |
| Neutrophil count (×10⁹/L) | |
| <1.8 | 2.0 |
| <6.3 | 1.0 |
| <25.0 | 2.0 |
| ≥25.0 | 3.0 |
| Patient age | |
| <46 | −1.0 |
| <58 | 1.0 |
| <83 | 2.0 |
| ≥83 | 3.0 |
| White blood cell count (×10⁹/L) | |
| <4.0 | 2.0 |
| <10.0 | 1.0 |
| <28.0 | 2.0 |
| ≥28.0 | 3.0 |
| Platelet count (×10⁹/L) | |
| <125.0 | 2.0 |
| ≥125.0 | 1.0 |
| INR (PT) | |
| ≤1.2 | 0.0 |
| ≤1.4 | 1.0 |
| ≤1.6 | 2.0 |
| >1.6 | 3.0 |
| Lymphocyte count (×10⁹/L) | |
| <1.1 | 2.0 |
| ≥1.1 | 1.0 |
| Monocyte count (×10⁹/L) | |
| <0.1 | 2.0 |
| <0.6 | 1.0 |
| ≥0.6 | 2.0 |

dysfunction. Except for the erythrocyte category, laboratory values across all other systems increase consistently with risk level, peaking in the dangerous group. The consistent upward trend supports the use of coagulation-inflammatory markers as indicators of systemic severity and their utility in clinical risk stratification.

CIS1 and CIS2 exhibited significant heterogeneity in clinical characteristics, mortality rates, and coagulation-inflammatory markers (Fig. 3c, all $p < 0.001$). CIS2 patients had a significantly greater mortality rate than did CIS1 patients (32.23% vs. 25.84%, $p < 0.001$). The Kaplan–Meier curves (Fig. 4c) demonstrated that anticoagulant treatment significantly reduced 28-day mortality in both subphenotypes ($p < 0.001$). Septic patients with both CIS1 (HR: 0.59; 95% CI: 0.50–0.70, $p < 0.005$) and CIS2 (HR: 0.42; 95% CI: 0.31–0.58, $p < 0.005$) status benefitted from anticoagulant treatment (Fig. 4c). Moreover, no significant difference in mortality rate was observed between CIS1 and CIS2 patients receiving anticoagulant treatment ($p = 0.391$). This means that anticoagulant treatment can significantly benefit the CIS2 subgroup with worse outcomes. These findings are consistent with current studies; despite over 100 sepsis subtypes, it remains unclear whether patients benefit from each new subtype strategy[8]. However, subphenotyping combined with risk stratification can reveal heterogeneity in anticoagulant treatment effects, enhancing the safety of anticoagulant treatment decision-making, as illustrated in Fig. 4d and Supplementary Table 14.

Last but not least, clinicians could conduct real-time risk stratification and subphenotypic classification of sepsis based on the SMART scorecard or our open-access sepsis subphenotype and SMART platform (http://smartsepsis.org.cn). Here, we conducted risk stratification, phenotypic classification, and prognosis prediction for 40 sepsis patients locally admitted from March 21, 2025, to April 27, 2025 (external observational verification only) (Fig. 4e–h; Supplementary case materials 1 and 2). The results showed that the proportions of patients in the four risk levels of mild, moderate, severe, and dangerous were 12.5%, 30%, 32.5%, and 25%, respectively (Fig. 4e), and their 28-day actual mortality rates were 0%, 16.7%, 38.5%, and 90% respectively (Fig. 4f). The overall mortality rate of the CIS1 subphenotype was 33.3%, significantly lower than 53.8% of CIS2, and the trend was the same at different risk levels (Fig. 4g, h). The mortality rate at the dangerous level (this external observational cohort) was a little higher than the predicted rate of the model in this study (approximately 50%), as well as simultaneously increased the overall mortality rates of CIS1 and CIS2, which might be related to the small sample size, but the overall trend was consistent. Therefore, clinicians can utilize the SMART scorecard or our open-access platform to conduct real-time risk classification and subphenotyping of patients, enabling objective and accurate assessment of sepsis patients to intervene as early as possible and improve their prognosis.

## Discussion

We developed two heterogeneity-aware methods: an XAI-powered prognostic model (SepsisFormer) and an automated sepsis risk stratification tool (SMART). SepsisFormer outperformed many existing models in prognostic prediction, and SMART outperformed comparably to established scoring systems. Meanwhile, we have established a webpage related to SMART scores and subphenotypic classification, which is now open for sharing (http://smartsepsis.org.cn). Explainability analysis identified and validated the critical role of coagulation-inflammatory markers in sepsis heterogeneity. The eight markers, comprising seven coagulation-inflammatory markers and patient age, can predict sepsis prognosis, identify sepsis subphenotypes, and

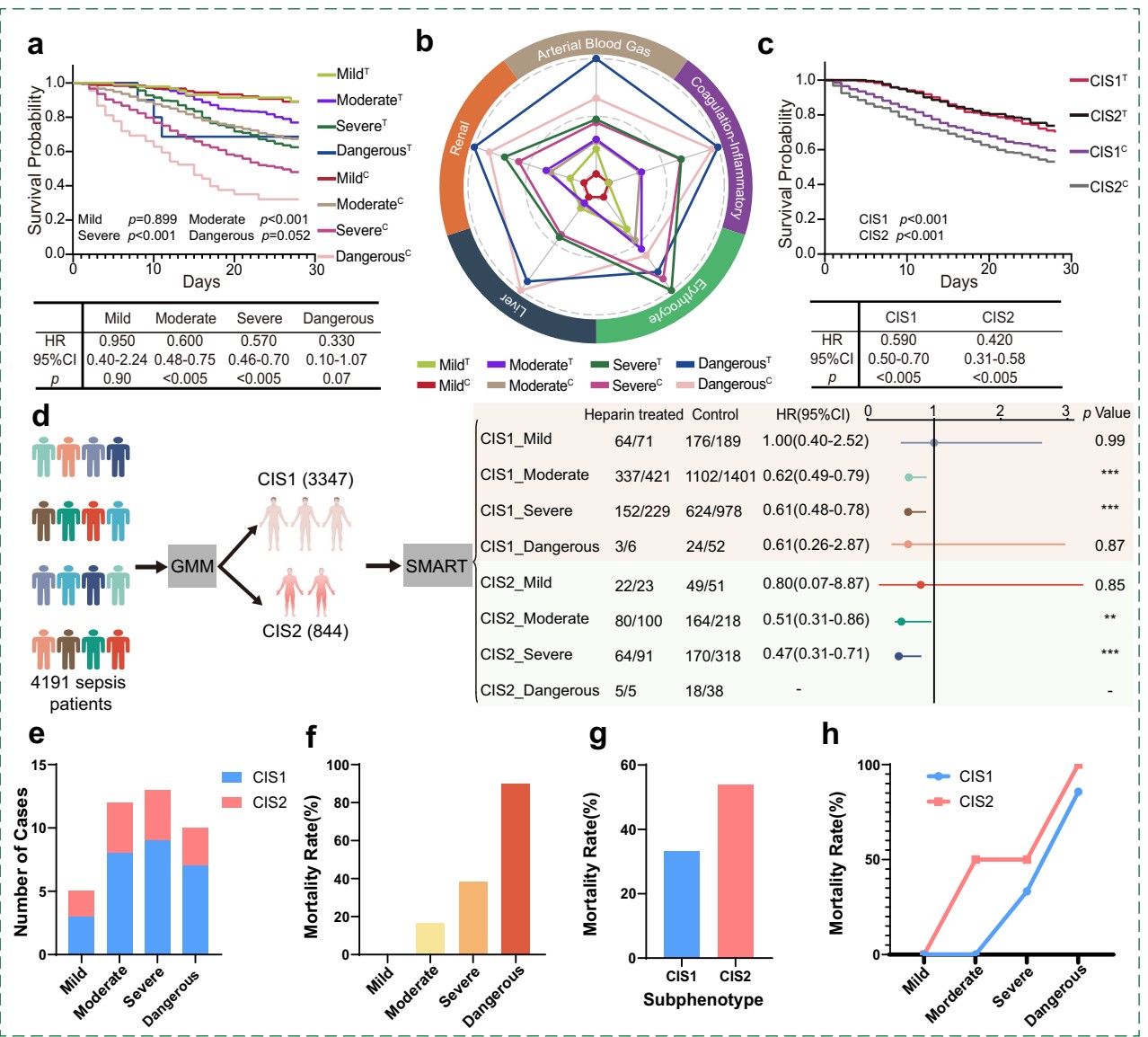

**Fig. 4 | Heterogeneity assessment of anticoagulant treatment effects and external validation of risk stratifications. a, c** Kaplan–Meier survival curves illustrating the cumulative probability of 28-day survival for patient subgroups with distinct subphenotypes (CIS1 and CIS2) or risk levels (mild, moderate, severe, and dangerous) in the heparin treatment and control subgroups. Log-rank tests for survival analysis. **b** Radar plot highlighting different degrees of severity across five categories among the four risk levels in the heparin treatment and control subgroups. **d** Septic patients were divided into eight subphenotype-based subgroups with risk stratification; hazard ratios (HR), 95% confidence intervals (CI; error bars), and two-tailed Log-rank *p* values show the subgroup-specific benefits of heparin treatment. HR > 1 denotes higher risk, HR < 1 denotes lower risk vs. reference. **e–h** External validation of real-time SMART risk stratification and subphenotype classification conducted using our open-access platform (http://smartsepsis.org.cn). **e** The real-time (day 1) risk stratification and distribution of the 40-patient

cohort. The number of patients in the four risk levels of mild, moderate, severe and dangerous were 5 (CIS1, 3 and CIS2, 2), 12 (CIS1, 8 and CIS2, 4), 13 (CIS1, 9 and CIS2, 4) and 10 (CIS1, 7 and CIS2, 3). **f** The mortality rates of different risk levels were 0%, 16.7%, 38.5% and 90% respectively. **g** The mortality rates of CIS1 and CIS2 were 33.3% and 53.8%. **h** The mortality rates of patients with different CIS classifications at the four risk levels were as follows: at the mild level, both CIS1 and CIS2 had a mortality rate of 0; at the moderate level, CIS1 had a mortality rate of 0 and CIS2 had a rate of 50%; at the severe level, CIS1 had a rate of 33.3% and CIS2 had a rate of 50%; and at the risk level, CIS1 had a rate of 85.7% and CIS2 had a rate of 100%. Created in BioRender. Niu, B. (2025) https://BioRender.com/qb3a3yr. \*\**p* < 0.01, \*\*\**p* < 0.001. T heparin treatment subgroup, C control subgroup, HR hazard ratio, CI confidence interval, CIS1 Coagulation-inflammatory subphenotype, and CIS2 coagulation-inflammatory subphenotype 2.

develop risk scores for septic patients. Different subgroups of sepsis patients were identified using unsupervised methods and SMART, and the heterogeneity of anticoagulant treatment effects across these subgroups was assessed. Patient populations stratified by subphenotype and risk level had disparate clinical characteristics and mortality rates. This study also provides a reference for decision-making regarding anticoagulant treatment, particularly for patients in the moderate and severe subgroups. In addition, the safety of anticoagulant treatment decision-making can be improved by more

profound disclosure of HTEs through subphenotypes in conjunction with risk stratification.

A significant strength of this study lies in clarifying the important functions of coagulation-inflammatory markers in sepsis progression. An APTT > 37 s and INR > 1.2 are considered prolonged clotting times, which may indicate coagulation disorder conditions involving the consumption of coagulation factors and the formation of microthrombi[30]. Alterations in white blood cell populations, which are predominantly composed of lymphocytes (20–40%), monocytes

(3–8%), and neutrophils (40–70%), are correlated with the severity of acute inflammatory responses and mortality in sepsis patients[23]. An immune-related neutrophil-to-lymphocyte ratio (NLR) > 9.8, a platelet-to-lymphocyte ratio (PLR) > 249.89, and a lymphocyte-to-monocyte ratio (LMR) ≤ 2.18 are important determinants of mortality in septic patients. Their dysregulated interplay may reflect an imbalance in the inflammatory response or immune status[31]. The SIRI is calculated based on peripheral blood neutrophil, monocyte, and lymphocyte levels. SIRI ≥ 6.32 is associated with a greater risk of short- and long-term mortality[32]. Lymphocytes in inflammation undergo apoptosis, which decreases in response to sepsis-induced stimuli. Persistently low lymphocyte counts reflecting adaptive immune function may be associated with increased mortality and a greater risk of developing chronic infections[33]. Following ischemia, PLTs are involved in sepsis-associated inflammation, vascular contracture, thrombosis, and delayed tissue damage. Both thrombocytopenia and thrombocytosis may reflect the distinct severity of sepsis. Therefore, the two measures of coagulation (INR and APTT) and the five measures of inflammation (WBC, lymphocyte, monocyte, neutrophil, and PLT counts) are reliable markers of sepsis.

Although transcriptomic data are not direct items for SMART scores and CIS typing, they further enhance the explanation of the importance of coagulation-inflammatory markers in diagnosis and prognosis assessment. To further explain the importance of coagulation-inflammatory indicators in diagnosis and prognosis assessment, we integrated transcriptomic data. Five genes (STAT5B, MTHFR, HPSE, AAK1, and MX1) had significant potential in predicting 28-day survival in septic patients. The STAT5B proteins are indispensable for immune regulation and homeostasis and influence the development and functionality of diverse hematopoietic cells[34]. Studies have suggested that both inadequate and excessive expression of MTHFR can exacerbate MTX-induced myelosuppression, leading to diminished levels of leukocytes, granulocytes, platelets, and hemoglobin[35]. HPSE inhibition conserves heparan sulfate within the glycocalyx and mitigates sepsis-induced injury[36]. AAK1, found in immune cells, impacts virus endocytosis and inflammation, playing a role in sepsis-related coagulation[37]. The MX1 genes, which encode dynamin-like GTPases, play crucial roles in the defense of mammals against a diverse array of viral infections[38]. In addition, CD59, SER-PINB2, CFD, and P2RX1 can be potential biomarkers for sepsis diagnosis. CD59 is correlated with the severity of organ damage in sepsis by inhibiting the formation of the complement membrane attack complex[39]. The presence of SERPINB2 in plasma is associated with sepsis outcomes[40]. CFD plays an important role in the coagulation process by blocking platelet activation[41]. The activation of P2RX1 causes platelets to release ATP, enhancing neutrophil glycolytic metabolism and NET production[42], whereas excessive NET production during sepsis may induce intravascular thrombosis and multi-organ failure. Moreover, external validation of transcriptomic datasets also yielded consistent results (Supplementary Figs. 8 and 9).

Another strength is the ability to distinguish subphenotypes and risk levels of septic patients using only coagulation-inflammatory markers and patient age. One of the important studies that directly applied transcriptomic data to the subtyping of sepsis came from the MARS consortium in 2017[43]. They enrolled a total of 787 cases of sepsis patients in a discovery cohort and two validation cohorts. Through machine learning and analysis of DEGs, sepsis was classified into four subtypes, MARS1-4. Among them, MARS1 had the poorest prognosis, with a mortality rate as high as 35%. This classification at the level of gene expression is also named endotype. In 2019, Seymour CW et al.[7] conducted phenotypic analysis on 20,189 patients with sepsis using statistics, machine learning, and simulation tools. Twenty-nine variables, including cardiovascular, hematopoietic, hepatic, coagulation-inflammatory, neurological, pulmonary, and renal systems were used. Finally, they divided the patients into four phenotypes: α, β, γ, and δ.

Among them, the mortality rate of phenotype α was the lowest at approximately 5%, while the mortality rates of type β, γ and δ were 13%, 24% and 40% respectively. From the perspective of mortality rates, our risk stratification results are similar to those of the Seymour CW's phenotypic classification. However, we revealed the heterogeneity of sepsis from two aspects: the stratification of risks and the subtypes. Moreover, the clinical indicators we use are fewer and more beneficial for clinical practice, which not only reduces the consumption of patients' blood samples but also saves costs, and it is also conducive to real-time dynamic assessment. Furthermore, the Seymour CW team has not yet developed a scoring system or a shared platform that can be universally implemented by clinicians. Most importantly, although many sepsis subtypes have been published at present, due to the different goals, the use of different clinical indicators, different machine learning models, or black-box algorithms, there are varying degrees of differences among these subtypes, ultimately resulting in low comparability or overlap among them (like MARS1-4, SRS1-2, Hyper or hypo inflammatory, and SENECA subtypes)[8], and no shared classification tools or platforms have been simultaneously proposed. However, it is still necessary to further conduct comparative studies on our classification and the existing subtype classifications.

The anticoagulant drug heparin may offer potential benefits in sepsis management, with clinical outcomes varying across different risk level and subphenotypes (Supplementary Fig. 11). Its primary anticoagulant mechanism involves binding with antithrombin to inhibit the activity of thrombin and factor Xa, inhibiting platelet activation, promoting the release of tissue factor pathway inhibitors, and increasing vascular permeability[44,45]. In addition to its well-known anticoagulant effects, Heparin has several immunomodulatory properties and protects the glycocalyx from shedding[46]. Heparin can also regulate the coagulation or inflammatory response by inhibiting the expression of SERPINB2, a diagnostic gene we discovered[47]. Under certain circumstances during sepsis, local thrombosis acts as an antimicrobial matrix to protect against pathogens, forming an intrinsic immune mechanism called "immunothrombosis"[48]. Patients classified in the lowest risk level (mild level) in this study did not demonstrate any clinical advantage from anticoagulant therapy. In patients with the highest risk level (dangerous level), anticoagulant therapy also did not significantly improve clinical outcomes. Still, it demonstrated some effect, which needs to be further confirmed by extensive sample RCT studies. The underlying cause of this difference may lie in the continuous and excessive activation of inflammation in patients with moderate, severe, and dangerous risk levels, which could lead to uncontrolled thrombosis activation[49]. This overwhelming thrombosis leads to the development of thrombotic disorders and the inability to engage host defense, which plays a critical role in inducing multiple organ dysfunction syndrome and subsequent death. Furthermore, for patients classified as dangerous risk levels, most patients already have severe organ dysfunction and complications, so, understandably, anticoagulation alone may show a limited difference in prognosis. However, based on SMART risk stratification, subphenotype analysis, the mechanism of action of heparin[50], and current data results, we have reason to believe that heparin anticoagulant treatment can improve the prognosis of patients with heterogeneous sepsis by selecting the appropriate population and initiation time.

In summary, this study introduces the prognostic model Sepsis-Former, the automated risk stratification tool SMART, and the subphenotypes CIS1/CIS2 to characterize sepsis heterogeneity. Through multi-level explainability analyses, age and seven routine coagulation-inflammatory markers were identified as key predictors for sepsis diagnosis and prognosis. SepsisFormer achieved an AUC of 0.9301, outperforming state-of-the-art models, while SMART reached an AUC of 0.7360, exceeding conventional clinical scores and effectively stratifying mortality risk. Notably, CIS2 patients showed higher mortality and distinct coagulation-inflammatory profiles compared to CIS1.

Integrating subphenotyping with risk stratification uncovered heterogeneity in anticoagulation treatment effects, supporting more precise and safer therapeutic decision-making in sepsis management. Patients with moderate/severe levels or CIS2 get more substantial benefits from anticoagulant treatment. An open-access, web-based platform (http://smartsepsis.org.cn) facilitates real-time risk stratification and subphenotype identification using only seven low-cost, routinely available coagulation-inflammatory biomarkers. Its simplicity, accessibility, and practical applicability make it a promising tool for improving sepsis management worldwide, particularly in resource-constrained healthcare settings.

## Methods

### Data acquisition

In this study, EHR data were obtained from the First Affiliated Hospital of Chongqing Medical University and three large, publicly available databases, MIMIC-III, MIMIC-IV, and eICU-CRD, forming our four retrospective cohorts. Transcriptomic data were obtained from the Ningbo Medical Center Lihuili Hospital and four publicly available datasets: the GEO, the KEGG, GeneCards, and DisGeNET.

Ethical approval (ID: 2019-312, First Affiliated Hospital of Chongqing Medical University) was granted for the collection of EHR data from adult patients admitted between January 2018 and April 2021. A total of 428 septic patients were enrolled within one hour of admission or within one hour of an acute exacerbation for current inpatients, following the Sepsis-3 definition. This study followed the standards of the Declaration of Helsinki, and written informed consent was obtained from all patients. A total of 11,980 septic patients (MIMIC-III: 2371, MIMIC-IV: 4191, and eICU-CRD: 5418) from publicly available databases were included after the exclusion of incomplete blood test results and those under 18 years of age. Permission to use the data was obtained for all the databases (MIMIC-III No. 36181465, MIMIC-IV No. 46463103, and eICU-CRD No. 12855636). Because of the de-identified nature of the data, informed consent was waived.

With ethical approval (ID: KY2023SL146-01, Ningbo Medical Center Lihuili Hospital) and written informed consent from all participants, blood samples for RT–qPCR were obtained from a total of 29 septic patients over 18 years of age who met the criteria for sepsis-3 and 11 healthy volunteers. The GSE65682 dataset in the GEO database includes 760 sepsis patient samples and 42 healthy control samples. Only 365 samples from survivors and 114 samples from non-survivors were included after excluding septic patients with incomplete 28-day mortality data. CRGs and IRGs, including genes from the hsa04610 and hsa04611 pathways, were obtained from the KEGG database, and DIC-related genes were acquired from trusted databases such as GeneCards and DisGeNET.

Furthermore, to enable real-time and external validation of the risk stratification and subphenotypic classification, we conducted a prospective observational study using our risk stratification and subphenotype platform. This study was approved by the Ethics Committee of Chongqing University Central Hospital (Chongqing Emergency Medical Center) (ID: 2025-55), and written informed consent was obtained from all 40 septic patients. The endpoint event was to observe the actual 28-day mortality rate and obtain the SMART score and subphenotype at the time of enrollment. After obtaining written informed consent, a total of 40 patients with sepsis were enrolled from March 24, 2025, to April 28, 2025.

In the design and implementation of this study, there is no sex or gender difference and no sex or gender bias.

### SepsisFormer: a post-hoc XAI prognostic model based on sepsis predictors

SepsisFormer is a heterogeneity-aware and post-hoc XAI neural network for sepsis prognostic prediction. It consists of three core components: a domain-adaptive generator, an integrated transformer

encoder, and a multilayer perceptron with a loss function. SepsisFormer is detailed below. (a) Input. SepsisFormer employs sepsis predictors, driven by medical knowledge, from the EHR as input. (b) Domain-adaptive generator for fine-tuning. To address the class imbalance and distribution heterogeneity identified through our analysis of mortality outcomes and multi-center covariate distribution shifts, we integrated a domain-adaptive generator module into the prognostic modeling framework for fine-tuning. We specifically implemented and compared several state-of-the-art domain adaptation methods, including Mean–teacher[51], Whitening[52], and Moment Matching[53], to reduce inter-center distributional discrepancies. Crucially, we also propose MMID-SMOTE, an innovative and clinically practical data augmentation strategy. This method incorporates statistical moment alignment and min-max interval constraints. MMID-SMOTE ensures that the synthetic samples generated are both statistically robust and clinically pertinent, thereby substantially enhancing the model's generalizability to unobserved target domains while maintaining clinical reliability and applicability. (c) Encoder. The encoder is composed of 8 integrated Transformer layers, each with eight parallel self-attention heads, three dense layers, and one position feedforward network, all of which are activated by the Gaussian error linear unit (GELU). Through the self-attention head, the model learns the long-term dependencies of different predictors. These extracted features are then connected into dense layers to learn complex mapping relationships. Feedforward networks and the GELU enhance the characterization of potential nonlinear relationships between predictors. (d) Decision-making. The model uses an MLP for decision-making, which is composed of two fully connected hidden layers with 64 and 128 neurons. Finally, the softmax function is used to classify patients into outcomes of survival or nonsurvival. (e) Pretraining Procedure. Cohort 1 (MIMIC-III and eICU-CRD) was partitioned 7:3 for training and internal testing. From this training data, the pre-trained SepsisFormer model was initialized, establishing its core architecture, weights, and hyperparameters. SMOTE was applied for data augmentation to address class imbalance and enhance model robustness during this phase. (f) Fine-tuning Procedure. Subsequently, for domain adaptation and enhanced generalizability, SepsisFormer underwent fine-tuning. Cohort 1 served as the source domain, with fine-tuning specifically targeting external validation cohorts: Cohort 2 (MIMIC-IV) and Cohort 3 (Local ICU) as the target domains. A Domain-adaptive Generator was employed for data augmentation during this transfer learning stage, optimizing performance in the new clinical center. The details of SepsisFormer can be found in Supplementary Method 1.

### Explainability analyses

Multi-view explainability analyses were conducted from three perspectives: two post-hoc approaches (cluster-informed EHR variable-level and AI model-level) and a transcriptomic-level analysis inspired by multi-omics principles. To select key categories that effectively mirror sepsis pathophysiology and are practical for clinical application, we employed unsupervised clustering methods, including GMM, MiniBatchKMeans, K-means, HAC, and Birch. Chord diagrams and radar plots enhance the interpretability of unsupervised clustering results by revealing the variable patterns defining each cluster. Clinical expertise and Pearson's correlation matrix were employed to analyze the associations between individual laboratory measurements within each category.

For model-level explainability, global and local SHAP analyses were employed to investigate the contribution of laboratory measurements for all and individual patients in the cohorts to the prognostic prediction model. Sankey diagrams were utilized to visualize the relationships between the 1-N Transformer neural network in the encoder and laboratory measurements.

Transcriptomic-level explainability explores transcriptomic biomarker evidence to gain valuable insights into cellular processes.

Moreover, univariate and multivariate logistic regression were used to identify genes associated with sepsis diagnosis among the DEGs related to coagulation-inflammation between the sepsis and control groups. RT–qPCR analysis was employed to validate the expression of the aforementioned genes. DEGs were identified between the sepsis survival and nonsurvival groups, with a focus on those associated with DIC. A prognostic prediction model was constructed via least absolute shrinkage and selection operator (LASSO) and multivariate Cox regression analyses, and genes from the model were used to construct a nomogram for predicting the probability of 28-day survival in septic patients. Supplementary Method 2 provides the detailed DEG screening steps and RT–qPCR experimental steps. Supplementary Method 3, 4 provide methods for external validation of DEGs diagnostic performance and prognostic performance, respectively.

### Subphenotype identification

An unsupervised approach was used to derive subphenotypes of sepsis patients on the basis of routine laboratory measurements. This process encompasses two stages: selecting the number of clusters and executing the unsupervised clustering algorithm. The silhouette score[54], Calinski–Harabasz score[55], and Davis–Bouldin score[56] were jointly used to determine the optimal cluster number for subphenotypes. The optimal number of clusters $k$ was strictly determined by maximizing the silhouette and Calinski–Harabasz scores while minimizing the Davis–Bouldin score. Subsequently, $k$ distinct subphenotypes were identified via the unsupervised method, which has the best clustering capability for generating chord diagrams within the subtype-informed EHR variable-level explainability framework. To visualize the identified clusters, fast independent component analysis (FastICA)[57] was applied to transform the cluster data into a new set of maximally independent features, facilitating their projection onto a lower-dimensional space for effective visualization. For each identified subphenotype, assessments were conducted for the SIRI and mortality rates.

### SMART: An automated risk stratification tool for sepsis

SMART is an automated risk stratification tool built using seven coagulation-inflammatory markers and patient age. The process of this heterogeneity-aware method is as follows: MIMIC-III and MIMIC-IV are merged, and SMOTE is employed to balance the class distribution. A supervised bottom-up method, ChiMerge[58], and prior medical knowledge are then combined to discretize continuous variables, mitigating the influence of extreme values and reducing the risk of model overfitting. The weight of evidence and information value are applied to assess and explain the associations between the markers and mortality prediction. Logistic regression with a LASSO penalty is used to mitigate the influence of anomalous data on the model. These markers are subsequently combined with the coefficients from the resulting logistic regression models to establish scores for each marker within the scorecard. An online SMART risk scoring system was developed and implemented on the basis of the final scorecard model results, adhering to the B/S architecture paradigm. Scorecard generation was performed via the scorecard method implemented in the scorecardpy Python package (version 0.1.9.6, https://github.com/ShichenXie/scorecardpy). Supplementary Fig. 13 shows the working principle of the schematic illustration. A more specific description can be found in Supplementary Methods 5–7.

### Statistical analysis

Statistical analyses were performed via Python 3.8 and R 4.3.2. ROC curves were constructed to assess the predictive power of the SepsisFormer, SMART, and transcriptomic diagnostic and prognostic prediction models. Additionally, the performance evaluation of the transcriptomics prognostic prediction models included calibration curves, decision curves, and kappa consistency coefficient analyses. The effect of heparin treatment was evaluated via Kaplan–Meier plots

for 28-day mortality and the Cox proportional hazards model. Kaplan–Meier plots were generated via GraphPad Prism 8.0 to examine the impact of three consecutive days of heparin treatment on survival outcomes in septic patients stratified by subphenotype and risk level. The Cox proportional hazards model was employed via HR to quantify the benefit of heparin across different subphenotypes and risk levels. Continuous variables are presented as medians (interquartile ranges) and were analyzed via non-parametric two-tailed Mann–Whitney U tests. Levene tests were used for variance homogeneity, and two-sided Wilcoxon rank-sum tests were used for violin plots. The log-rank test was applied to the Kaplan–Meier plots. A significance level of $p < 0.05$ was considered statistically significant. DeLong's tests were performed to statistically compare the AUCs of different models using Delong_t-est from the MLstatkit.stats package, in conjunction with roc_auc_-score from scikit-learn.

### Reporting summary

Further information on research design is available in the Nature Portfolio Reporting Summary linked to this article.

## Data availability

Source data are provided (https://github.com/zhuli19031218/SepsisFormer/tree/main/SourceData). In this study, EHR data were obtained from the First Affiliated Hospital of Chongqing Medical University and three publicly available databases: MIMIC-III(https://physionet.org/content/mimiciii/1.4/), MIMIC-IV(https://physionet.org/content/mimiciv/2.2/), and eICU-CRD (https://www.physionet.org/content/eicu-crd/2.0/). For the public EHR data, we adhered to all data use agreements, conducting experiments on observational, retrospective data. All three datasets require user registration and a signed data use agreement for timely access.Transcriptomic data were obtained from the Ningbo Medical Center Lihuili Hospital and four publicly available datasets: the Gene Expression Omnibus (GEO, https://www.ncbi.nlm.nih.gov/geo/), the Kyoto Encyclopedia of Genes and Genomes (KEGG, https://www.genome.jp/kegg/), GeneCards (https://www.genecards.org/), and DisGeNET (https://www.disgenet.org/). To promote transparency, reproducibility, and clinical applicability, we have made the following resources publicly available. We developed an interactive web-based online platform for real-time sepsis subphenotyping and risk prediction, enabling clinicians and researchers to use their own data without coding or technical expertise (http://smartsepsis.org.cn). All raw data, preprocessed datasets, and fully documented source code are available in our GitHub repository(https://github.com/zhuli19031218/SepsisFormer). A detailed step-by-step video tutorial is provided to guide users through the complete process of reproducing the main results presented in this study (https://doi.org/10.5281/zenodo.15634368). Source data are provided with this paper.

## Code availability

Source data are provided (https://github.com/zhuli19031218/SepsisFormer/tree/main/SourceData). In this study, EHR data were obtained from the First Affiliated Hospital of Chongqing Medical University and three publicly available databases: MIMIC-III(https://physionet.org/content/mimiciii/1.4/), MIMIC-IV(https://physionet.org/content/mimiciv/2.2/), and eICU-CRD (https://www.physionet.org/content/eicu-crd/2.0/). For the public EHR data, we adhered to all data use agreements, conducting experiments on observational, retrospective data. All three datasets require user registration and a signed data use agreement for timely access.Transcriptomic data were obtained from the Ningbo Medical Center Lihuili Hospital and four publicly available datasets: the Gene Expression Omnibus (GEO, https://www.ncbi.nlm.nih.gov/geo/), the Kyoto Encyclopedia of Genes and Genomes (KEGG, https://www.genome.jp/kegg/), GeneCards (https://www.genecards.org/), and DisGeNET (https://www.disgenet.

org/). To promote transparency, reproducibility, and clinical applicability, we have made the following resources publicly available. We developed an interactive web-based online platform for real-time sepsis subphenotyping and risk prediction, enabling clinicians and researchers to use their own data without coding or technical expertise (http://smartsepsis.org.cn). All raw data, preprocessed datasets, and fully documented source code are available in our GitHub repository(https://github.com/zhuli19031218/SepsisFormer). A detailed step-by-step video tutorial is provided to guide users through the complete process of reproducing the main results presented in this study (https://doi.org/10.5281/zenodo.15634368).

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

## Acknowledgements

We acknowledge Prof. Craig Coopersmith (Emory Critical Care Center, Emory University), Prof. Teng Fei (Northeastern University), Prof. Yan Xu (Beihang University), Mr Zhendong Zhai, Dr. Chaoqun Zhang, and Dr. Min Wan (Nanchang University) for their constructive comments on this work. We thank Miss Yuting Fan (Chongqing Foreign Language School) for grammar proofreading. This work was supported by the National Natural Science Foundation of China (62461038, 82241059, 82125022), the National Science and Technology Major Project (2025ZD0551300, 2025ZD0551301), the Fundamental Research Funds for the Central Universities (2022CDJQY-002, 2022CDJYGRH-014), the Open Research Program of Chongqing Key Laboratory of Highly Pathogenic Microbes (2025ZDSYSZD002), the Key Project of Chongqing Medical Scientific Research (Joint Project of Chongqing Health Commission and Science and Technology Bureau) (2023ZDXM012), the Yunnan Province Major Science and Technology Special Project (202302AA310039), and the Macao Special Administrative Region Science and Technology Development Fund 0003/2023/RIC.

## Author contributions

L.Z., B.N., and F.G. provided the conceptualization. L.Z., Z.C., and H.C. collected the publicly available ICU data. B.N., H.Z., X.T., J.W., and Y.R. collected the local ICU data. F.G., L.L., J.S., C.H., L.S., and T.L. collected and analyzed the transcriptomic data. L.Z., Z.C., and Z.Y. created SepsisFormer. L.Z., B.N., Z.C., and H.C. conducted the risk stratification and subphenotype analysis. L.Z., B.N., and H.C. established the SMART model and associated website. B.N., Y.L., H.Z., F.Z., and F.L. participated in the clinical discussion and validation. L.Z., Z.C., and H.C. wrote the original manuscript. K.W., Y.C., and W.Y. refined the figures and codes. B.N., L.Z., and F.G. oversaw the investigation. B.N., L.Z., Y.L., and T.L. secured funding acquisition.

## Competing interests

The authors declare no competing interests.

## Additional information

[1]School of Information Engineering, Jiangxi Provincial Key Laboratory of Advanced Signal Processing and Intelligent Communications, Nanchang University, Nanchang, Jiangxi, China. [2]Department of Intensive Care Medicine, Chongqing Emergency Medical Center, Chongqing University Central Hospital, School of Medicine, Chongqing University, Chongqing, China. [3]Department of Laboratory Medicine, Chongqing Center for Clinical Laboratory, Chongqing Academy of Medical Sciences, Chongqing General Hospital, School of Medicine, Chongqing University, Chongqing, China. [4]School of Medicine, Nanchang University, Nanchang, Jiangxi, China. [5]Ningbo Institute of Innovation for Combined Medicine and Engineering (NIIME), The Affiliated Lihuili Hospital of Ningbo University, Ningbo, Zhejiang, China. [6]Department of Intensive Care Medicine, The Affiliated Lihuili Hospital of Ningbo University, Ningbo, Zhejiang, China. [7]Department of Gastroenterology, Qilu Hospital of Shandong University, Jinan, Shandong, China. [8]Department of Critical Care Medicine, The First Affiliated Hospital of Nanchang University, Nanchang University, Nanchang, China. [9]College of Life Science and Laboratory Medicine, Kunming Medical University, Kunming, Yunnan, China. [10]Institute of Translational Medicine, Faculty of Health Sciences & Ministry of Education Frontiers Science Center for Precision Oncology, University of Macau, Macau, China. [11]Chongqing Key Laboratory of Highly Pathogenic Microbes, Chongqing, China. [12]Department of Emergency and Intensive Care Medicine, The First Affiliated Hospital of Chongqing Medical University, Chongqing, China. [13]Department of Surgery, Brigham and Women's Hospital and Harvard Medical School, Boston, MA, USA. [14]These authors contributed equally: Li Zhu, Zengtian Chen, Hong Zhang. [15]These authors jointly supervised this work: Yang Luo, Fei Guo, and Bailin Niu. ✉e-mail: luoy@cqu.edu.cn; lhlguofei@nbu.edu.cn; bniu@cqu.edu.cn

