## [Transparent Peer Review file · Nature Communications]

Explainable AI Unravels Sepsis Heterogeneity via Coagulation-Inflammation Profiles for Prognosis and Stratification

Corresponding Author: Dr Bailin Niu

Version 0:

Reviewer comments:

Reviewer #1

(Remarks to the Author)

The study looks at a significant problem. While prior literature has suggested various algorithms that can predict sepsis very well and, at times, better than physicians, this paper has three unique contributions. First, the model is built using external datasets and validated using a separate hospital dataset which presents some form of external validation for the model. Second, the development of a scoring system presents a possible translational use of this algorithm in clinical environments where the operating of real-time algorithms is not feasible. Third, it looks at biomarkers and provides some explanatory attempts of predicting sepsis, which is something challenging.

There are, however, some points to consider in the revision.

The writing, although clear, does not highlight the key findings of the paper. The results section focused excessively on the methods employed and did not provide sufficient clinical explanation as to why these biomarkers matter in explaining sepsis prediction. For example, SHAP plots are good for visualizing the importance of the predictors in the model, but in the main body of the paper, more should be presented to explain the clinical aspects of the findings. The methodological reporting should go into the methods section which also needs further revisions.

I find the explanatory part of sepsis prediction something that is significant, but this current form does not provide sufficient write-up in this area. I hope the authors can provide more details that may contribute to the clinical understanding of sepsis onset.

In terms of the reporting of the methods, here, less is perhaps more. The authors have presented a multitude of methods to support their results. Although in general, I do believe in the findings from a reviewer perspective, the large number of methods employed with limited reporting makes it exacting for the reviewer to establish the correctness of the execution of these methods. To illustrate, in the text, the authors wrote “[...] radar plots of subtypes identified by unsupervised clustering methods, including GMM, MiniBatchKMeans, K-means, hierarchical agglomerative clustering (HAC), and Birch [...]”. The writing here presents more of a report of a collection of methods with limited ability for the reviewers to verify and perhaps add limited value to the paper other than to signal to the review team that extensive work has been put into the paper. A similar writing style is pervasive in the entire methods section, e.g., “[...] MIMIC-III and MIMIC-IV are merged, and SMOTE is employed to balance the class distribution [...]” For example, what is the extent of SMOTE (percentage, number of neighbors used, etc.)? Even though I downloaded all the datasets in this submission, reading this article does not permit any reviewer to completely comprehend the inner workings of the analysis based on the reporting.

I suggest the authors do one of the two. 1) In the main paper, report a shorter list of analyses employed, but do that in a more comprehensive and clearer manner, complete with a STROBE or modified CONSORT diagram to showcase the data used. This presentation will lend more credibility to the results. If you want to showcase additional analyses reported here, maybe provide some additional supplement you might want to host on GitHub. These supplementary analyses may have minute computational errors or bugs – which is normally the case, but it does not fall within the purview of NComms review process.

2) Continue with your existing reporting of all the collection of analyses, but in your next review, you perhaps will have to consider providing more details of how the analyses for each technique is done, including the software codes in an executable format. This will also make the review process more exacting.

(Remarks on code availability)

I have downloaded the datasets, but the current reporting make it onerous and not possible for one to review the analysis in detail.

Reviewer #2

(Remarks to the Author)

1. What are the noteworthy results?

This study has some key findings.

(A) High Prognostic Accuracy of SepsisFormer

- SepsisFormer (Transformer-based model) achieves an AUC of 0.9301, outperforming traditional risk scoring models like qSOFA, SOFA, and APACHE II.
- The model demonstrates generalizability across multiple cohorts (MIMIC-III, MIMIC-IV, eICU-CRD, and a local ICU dataset).

(B) SMART Risk Stratification Model Improves Sepsis Classification

- SMART (a scoring system based on seven coagulation-inflammatory markers and patient age) achieved an AUC of 0.7360, surpassing most established clinical scoring systems.
- The study identifies four risk levels (Mild, Moderate, Severe, Dangerous), with clear mortality risk stratification (~5%, 15%, 30%, and 50% mortality rates, respectively).

(C) Identification of Two Sepsis Subphenotypes (CIS1 and CIS2) with Different Outcomes

- CIS2 patients exhibited higher mortality rates (~32.23%) compared to CIS1 (25.84%), and a stronger inflammatory-coagulation response.
- The study demonstrates that patients with CIS2 benefit more from anticoagulant therapy (e.g., heparin), while CIS1 shows moderate benefits.

(D) Transcriptomic Analysis Identifies Sepsis Biomarkers

- Five transcriptomic biomarkers (STAT5B, MTHFR, HPSE, AAK1, MX1) were found to be significantly linked to sepsis prognosis.
- The study suggests these biomarkers influence sepsis prognosis through immune and coagulation pathways.

2. Will the work be significant to the field and related fields?

Yes, this work is significant to the field.

Relevance to Sepsis Research:

- The study contributes to sepsis mortality prediction and risk stratification, both of which are essential for clinical decision-making in ICUs.
- The identification of coagulation-inflammatory subtypes (CIS1 and CIS2) aligns with ongoing efforts to better phenotype sepsis patients.

Relevance to Machine Learning & Explainable AI in Healthcare:

- SepsisFormer demonstrates that deep learning models can achieve superior predictive performance while maintaining interpretability through XAI methods.
- SMART scoring provides a structured risk assessment tool that could be clinically deployed.

Potential Applications in Other Fields:

- The approach can be extended to other critical conditions (e.g., septic shock, ARDS, multi-organ failure) where heterogeneity complicates prognosis.
- The explainability techniques used here could be applied to other AI-driven healthcare models.

Comparison to Existing Literature:

SepsisFormer's AUC of 0.9301 is notably higher than previously reported AI-based sepsis models (e.g., prior deep learning models in literature typically range from 0.85–0.91).

Risk stratification studies, such as those defining MARS1-4 or α , β , γ , δ subtypes, have not explicitly linked coagulation-inflammatory markers to treatment response.

However, the study does not compare CIS1 and CIS2 to other existing sepsis subtyping frameworks (e.g., MARS1-4, SRS1-2, hyper-/hypo-inflammatory states).

3. Does the work support the conclusions and claims, or is additional evidence needed?

Most claims are supported by strong experimental data.

Well-supported claims:

SepsisFormer outperforms existing scoring systems → Supported by ROC curves, AUC analysis, external validation across multiple datasets.

SMART provides a clinically relevant stratification system → Clear association between SMART's risk levels and increasing mortality.

CIS2 patients benefit more from anticoagulants → Kaplan-Meier survival analysis demonstrates significant mortality reduction in CIS2 patients receiving anticoagulation.

Less-supported claims:

CIS1/CIS2 are novel sepsis subtypes → The study does not compare them to previous subtyping models (e.g., MARS1-4, α -

δ).
Transcriptomic biomarkers are clinically relevant for sepsis prognosis → No external validation of transcriptomic findings in an independent dataset.

Recommendation:

- Benchmark CIS1/CIS2 against prior subtyping frameworks.
- Validate transcriptomic biomarkers in a completely independent dataset.

4. Are there any flaws in the data analysis, interpretation, and conclusions?

Overall, the methodology is robust, but a few concerns require attention.

(A) External Validation of SepsisFormer Needs Clarification

How the data were split for training, validation, and testing is unclear.

Recommendation: Clearly define cross-validation strategy and how model generalizability was assessed.

(B) SMART's Performance Varies Across Datasets

SMART's AUC ranges from 0.6222 to 0.7360—why does it perform worse in some cohorts?

Recommendation: Provide dataset-level breakdown and discuss factors influencing variability.

(C) Statistical Comparisons Are Missing for Model Performance

No DeLong's test or confidence intervals are reported to compare AUCs of different models.

Recommendation: Perform statistical comparisons to confirm that SepsisFormer is significantly better than traditional methods.

5. Is the methodology sound? Does the work meet the expected standards in your field?

Yes, the study follows best practices for machine learning in medicine, but it requires clarifications.

Areas for Improvement:

No external validation of transcriptomic biomarkers → Validation in an independent cohort is needed.

No prospective validation of SepsisFormer in real-time ICU settings.

6. Is there enough detail provided in the methods for the work to be reproduced?

The methodology is well-documented but requires additional clarification in some areas.

Details that are clear and reproducible:

- * Feature selection process for SMART.
- * SHAP-based explainability analysis.
- * Survival analysis methodology.

Details that require clarification:

- * How were CIS1 and CIS2 validated against existing subphenotypes?
- * How was model performance compared across datasets (cross-validation strategy)?
- * What hyperparameters were used in training SepsisFormer?

Recommendation:

- * Provide hyperparameter details for SepsisFormer.
- * Clarify subphenotype validation and external benchmarking.

(Remarks on code availability)

Reviewer #3

(Remarks to the Author)

1. What are the noteworthy results?

- A prognosis model for sepsis that leverages Transformer models and domain adaptation technique that achieves high predictive accuracy and a scorecard model for risk stratification.

- The actual subphenotypes and risk levels classified by the findings from the model experiments, and how these actually capture patient heterogeneity. Although, I could appreciate this contribution later in the paper.

- An in depth analysis of and evidence for the heterogeneity of sepsis patients based on subphenotypes and risk levels.

2. Will the work be of significance to the field and related fields? How does it compare to the established literature? If the work is not original, please provide relevant references.

- The performance of existing AI prediction/prognosis models is mentioned in the introduction but does not include the techniques used in those models, which can help the reader to appreciate how the proposed models in this work contribute and differ from existing literature (in terms of input data, model choice, training paradigms, etc.).

- In the domain adaptive generation, the method for oversampling is claimed to be novel but also says that it uses existing domain adaptation methods. It is not clear what the novelty of the method is. Besides, the domain adaptation is briefly mentioned in the abstract and discussed again in the results under the Prognostic prediction performance of SepsisFormer based on coagulation-inflammatory markers header. The last paragraph of the introduction can include this component, as it seems relevant considering the heterogeneity and class imbalance problems it addresses.

- The transcriptomic biomarker evidence analysis and study of anticoagulant treatment effect seem crucial for better understanding sepsis, reveal insights about heterogeneity, and validate algorithmic model development. However, I think

that the presentation of these approaches can be clarified because it wasn't clear what experiments actually took place in this study or were using measurements from existing public datasets (E.g., the Heparin vs control studies, or PCR studies), and how the biomarkers were actually selected.

3. Does the work support the conclusions and claims, or is additional evidence needed?

- As I was reading the paper, I struggled to understand the support for some of the claims in the results section since it wasn't clear the criteria used for doing further classifications or analyses. For example, how does further categorizing the predictors into clinically relevant ranges (not clear how these are defined) facilitate clinical decision making? And then, why further analyzing the two subtypes across the five key categories of sepsis predictors? This comment is also related to the clarity presenting the methods.

- From my understanding and interpretation, I agree with the paper that explainable AI was used to uncover patient heterogeneity, I am not sure if the model SepsisFormer was built to be explainable. Many parts use modules from Transformers, a complex deep learning architecture, that as mentioned, can be beneficial for modeling the complex relationships between the predictors. In fact, the explainability technique used in the model directly is a post-hoc technique and this can be acknowledge for clarity. Besides, it is mentioned that the model is heterogeneity-aware because of the domain-adaptive generator, but this sounds like a data augmentation strategy for training.

- Claims around improving patient outcomes in the introduction: this has to be evaluated through the application in clinical decision-making and translational research.

- What is meant by clinically relevant in this statement: "XAI method achieved clinically relevant sensitivity and accuracy in prognostic prediction with only eight routine markers."

4. Are there any flaws in the data analysis, interpretation and conclusions? - Do these prohibit publication or require revision?

- Even though the components of SepsisFormer along with their functionality are explained, the process for making these algorithmic choices is not clear. Was there any ablation study to select the parameters or assess the contribution of some parts (e.g., the domain adaptation)? This will provide stronger evidence for the models proposed in this paper.

- The discussion section currently presents medical literature and definitions in a list-like format. While this demonstrates a strong understanding of sepsis from a clinical perspective (which is valuable for algorithm development and interpretation of the results), the manuscript would benefit from a more analytical approach. I recommend expanding the discussion to include comparative analyses of the different models for the prognosis and risk stratification task, the different domain adaptation strategies used, or any other pertinent comparisons.

- Considering the multiple insights found in this study, the paper is lacking an overall conclusion about the approach followed. This can also be beneficial to reinforce the main message and contribution of the paper.

5. Is the methodology sound? Does the work meet the expected standards in your field?

In general, the selected methods seem appropriate for the problems to be addressed in the paper. However, I found the following concerns:

- In the introduction it says that the methods use only routine laboratory blood measurements and how this provides an alternative to blood transcriptomic-based methods. Therefore, I found confusing when later in the paper an analysis of transcriptomic biomarkers is introduced (which involved additional methodological steps). Please clarify if I misunderstood this part.

- The analyses presented in the paper are extensive, but the rationale for each type is missing. For example, why does it matter to see the contribution of the five categories of predictors? And their associations? Why is it important to analyze the relationships between the 1-N transformer heads and lab measurements? Including these justifications can also clarify the choice for the visualization and/or analysis technique.

- Not clear how the subtypes alpha and beta in the variable-level explainability analysis differ from the subphenotypes if they are both identified based on distinctions in mortality rates.

- Throughout the paper, it is stated that eight markers were identified (7 coagulation-inflammatory measures + age) and used for prognosis modeling, risk stratification, and further identifying subphenotypes, but the rationale for selecting this subset of predictors is unclear. The use of routine coagulation-inflammatory measures is motivated in the introduction, but looking at Figure 2 it seems like there are more than 7 measures.

- For the evaluation of the risk stratification model, it could be beneficial to clarify what ground truth is used for the evaluation of this task.

6. Is there enough detail provided in the methods for the work to be reproduced?

Considering the details and codes provided in the supplementary materials, probably yes. However, I found additional experiments, like fine-tuning or pertaining the prognosis models, which was never mentioned in the main manuscript.

(Remarks on code availability)

Code was not available in the submission. Some pseudo codes were presented in the supplementary materials.

Version 1:

Reviewer comments:

Reviewer #1

(Remarks to the Author)

In my previous review I have raised the following points:

1. The clarity and framing of the main message/ contribution of the paper
2. Issues relating to the reporting of the methodology and sample
3. Greater transparency towards the methods employed (e.g. datasets and codes)

Glad to share that these three issues were addressed in this latest revision.

The results of these paper are noteworthy as indicated in my earlier review and the ability to translate the algorithm developed into a clinical profiling score has field implications.

Good effort of showcasing the execution of the codes in the videos.

(Remarks on code availability)

Reviewer #2

(Remarks to the Author)

The authors have adequately addressed the concerns, and the manuscript is now suitable for publication.

(Remarks on code availability)

Reviewer #3

(Remarks to the Author)

The main concerns were addressed in the revision, including the performance comparisons to existing models, including both of the main model and the domain-adaptation strategy. Implementation details and definitions were also provided for the latter method, which helped to understand how it differs from existing techniques. However, in the description of the domain adaptation in the supplementary method, the justification that analyzed class imbalance in mortality outcomes is not theoretical but rather empirical I think, since you based the analysis on your data samples. Further methodological details were provided for the transcriptomic analysis and study of HTEs effect. Ablation studies are also presented and clarifications around techniques used and claims made throughout the manuscript.

Besides, the authors extended the interpretation of the explainability analysis and how this information is useful to understand the heterogeneity of the disease and relevant biomarkers for sepsis prognosis. They also commented on the implications of having a model that predicts risk-stratified levels and the corresponding analyses.

In the results, there is connection between findings and medical relevance (e.g., found importance of coagulation-inflammatory indicators vs functional indicators of specific organs and how the former better reflects the pathophysiology of sepsis), and an application use case of the tool available online was demonstrated.

The paper still needs some organization edits, especially in the presentation of the results:

- Some analyses are listed but then presented later. E.g., 1) the correlation between predictors mentioned in line 145 and then presented in line 176, 2) finding a subset of laboratory measurements within one category (part (a) in line 155) but the clustering results of part (b) are presented first, followed by the correlation analysis.

- Improve the introduction of the transcriptomic-level explanations and transcriptome analysis: It is clarified that this was to show the important roles of coagulation-inflammation related indicators, but there are some aspects that can be clarified. where was the gene information gathered from? Briefly mention the methods involved in the paragraph in line 207.

- It is still unclear to me how the SEVEN predictors were selected. In the model-level explainability only 3 are mentioned (APTT, WBC, and age) and the importance of coagulation-inflammation predictors, but there at least 10 from what I can see in Figure 2c and 8 in Figure 2g. The overall selection process was illustrated in a figure in the response letter, but I may be missing some details to fully understand the criteria for picking those 7 variables. I also noticed that the functions of these 7 coagulation-inflammation markers is discussed in detail in the paragraph starting in line 362 and they are part of the online platform.

- Domain adaptation comparisons are introduced out of the blue in page 7. Explained the results using comparisons to other methods.

- Revise claim about an XAI tool: Sepsis former on its own is not an XAI tool. The analysis used post-hoc XAI tools to understand contributors to predictions and relationships between inputs.

- The use of unsupervised clustering is better explain to find the subphenotypes, but I am still confused about why there are two analyses for this, i.e., first to identify alpha and beta groups using 36 predictors and then CIS1 and CIS2 using only the seven coagulation-inflammation predictors? As another reviewer mentioned, not all analyses have to be part of the manuscript because it becomes hard to appreciate what are the main take aways.

- Regarding the radar plot in the assessment of HTEs: It seems like all predictors are included in this analysis. I thought that the SMART model only used 8 predictors. Please clarify why variables from the other categories were included and the purpose of this analysis. As another reviewer mentioned, not all analyses have to be part of the manuscript because it becomes hard to appreciate what are the main take aways.

Discussion:

Overall, the claims in the discussion are supported by the results: how explainability analyses were done to study sepsis heterogeneity, how the different patient populations stratified by subphenotype and risk levels had different clinical characteristics and mortality rates, and the insights from the combined analysis of HTEs' effect.

The clarifying sentence at the end of the third paragraph of the discussion can be included in the paragraph opening to mention the transition to a new analysis.

Included other works that have done subtyping, how the findings and techniques differ, and acknowledge that comparisons are hard between identified subphenotypes.

I suggest beginning the paragraph in line 400 with this sentence before you contrast the results with other works that have done subtyping: "Another strength is the ability to distinguish subphenotypes and risk levels of septic patients using only coagulation-inflammatory markers and patient age". And then in the next paragraph discuss the effects of HTEs for the subgroups.

Minor comments:

Introduce some acronyms in the main paper for clarity: HTE in the introduction, MMID-SMOTE in the results, the machine learning abbreviations (even if they are well known).

Line 425 "and so on" may not be clear for all readers and sounds informal.

(Remarks on code availability)

Even though the code is provided, the documentation can be improved and a step by step description on what codes to run to generate what results. The README file does not have enough instructions and does not mention which dependencies are needed. The names of the scripts can also be more informative. For the transcriptomic analysis results, the folder only contains data files, so I wonder if the code to recreate these results is missing.

Version 2:

Reviewer comments:

Reviewer #3

(Remarks to the Author)

Overall, the revision provides satisfactory responses to the comments I included in the previous iteration. Language clarifications were made in the method's description and justifications (for domain adaptation, correlation analyses, clustering, and explainability). Improvements were made in the organization of the results. The paper now explains the selection of variables for risk models and other analyses.

(Remarks on code availability)

The documentation of the repository has been improved with more clear instructions on how to run the code.

**Explainable AI-driven heterogeneity using coagulation–inflammatory markers**
**improves prognosis prediction, risk stratification, and anticoagulant treatment**
**effects for sepsis**

Corresponding Author: Niu Bailin

**This file contains all reviewer reports followed by all author rebuttals in order.**

Version 1:

Reviewer comments:

**Reviewer #1 (Remarks to the Author):**

The study looks at a significant problem. While prior literature has suggested various algorithms
that can predict sepsis very well and, at times, better than physicians, this paper has three unique
contributions. First, the model is built using external datasets and validated using a separate
hospital dataset which presents some form of external validation for the model. Second, the
development of a scoring system presents a possible translational use of this algorithm in clinical
environments where the operating of real-time algorithms is not feasible. Third, it looks at
biomarkers and provides some explanatory attempts of predicting sepsis, which is something
challenging.

There are, however, some points to consider in the revision.

The writing, although clear, does not highlight the key findings of the paper. The results section
focused excessively on the methods employed and did not provide sufficient clinical explanation
as to why these biomarkers matter in explaining sepsis prediction. For example, SHAP plots are
good for visualizing the importance of the predictors in the model, but in the main body of the
paper, more should be presented to explain the clinical aspects of the findings. The methodological
reporting should go into the methods section which also needs further revisions.

I find the explanatory part of sepsis prediction something that is significant, but this current form
does not provide sufficient write-up in this area. I hope the authors can provide more details that
may contribute to the clinical understanding of sepsis onset.

In terms of the reporting of the methods, here, less is perhaps more. The authors have presented a
multitude of methods to support their results. Although in general, I do believe in the findings
from a reviewer perspective, the large number of methods employed with limited reporting makes
it exacting for the reviewer to establish the correctness of the execution of these methods. To
illustrate, in the text, the authors wrote “[...] radar plots of subtypes identified by unsupervised

clustering methods, including GMM, MiniBatchKMeans, K-means, hierarchical agglomerative
clustering (HAC), and Birch [...]”. The writing here presents more of a report of a collection of
methods with limited ability for the reviewers to verify and perhaps add limited value to the paper
other than to signal to the review team that extensive work has been put into the paper. A similar
writing style is pervasive in the entire methods section, e.g., “[...] MIMIC-III and MIMIC-IV are
merged, and SMOTE is employed to balance the class distribution [...]” For example, what is the
extent of SMOTE (percentage, number of neighbors used, etc.)? Even though I downloaded all the
datasets in this submission, reading this article does not permit any reviewer to completely
comprehend the inner workings of the analysis based on the reporting.

I suggest the authors do one of the two. 1) In the main paper, report a shorter list of analyses
employed, but do that in a more comprehensive and clearer manner, complete with a STROBE or
modified CONSORT diagram to showcase the data used. This presentation will lend more
credibility to the results. If you want to showcase additional analyses reported here, maybe
provide some additional supplement you might want to host on GitHub. These supplementary
analyses may have minute computational errors or bugs – which is normally the case, but it does
not fall within the purview of NComms review process. 2) Continue with your existing reporting
of all the collection of analyses, but in your next review, you perhaps will have to consider
providing more details of how the analyses for each technique is done, including the software
codes in an executable format. This will also make the review process more exacting.

**Reviewer #1 (Remarks on code availability):**

I have downloaded the datasets, but the current reporting make it onerous and not possible for one
to review the analysis in detail.

**Reviewer #2 (Remarks to the Author):**

1. What are the noteworthy results?

This study has some key findings.

(A) High Prognostic Accuracy of SepsisFormer

• SepsisFormer (Transformer-based model) achieves an AUC of 0.9301, outperforming traditional
risk scoring models like qSOFA, SOFA, and APACHE II.

• The model demonstrates generalizability across multiple cohorts (MIMIC-III, MIMIC-IV,
eICU-CRD, and a local ICU dataset).

(B) SMART Risk Stratification Model Improves Sepsis Classification

• SMART (a scoring system based on seven coagulation-inflammatory markers and patient age)
achieved an AUC of 0.7360, surpassing most established clinical scoring systems.

• The study identifies four risk levels (Mild, Moderate, Severe, Dangerous), with clear mortality
risk stratification (~5%, 15%, 30%, and 50% mortality rates, respectively).

(C) Identification of Two Sepsis Subphenotypes (CIS1 and CIS2) with Different Outcomes

• CIS2 patients exhibited higher mortality rates (~32.23%) compared to CIS1 (25.84%), and a
stronger inflammatory-coagulation response.

• The study demonstrates that patients with CIS2 benefit more from anticoagulant therapy (e.g.,
heparin), while CIS1 shows moderate benefits.

(D) Transcriptomic Analysis Identifies Sepsis Biomarkers

• Five transcriptomic biomarkers (STAT5B, MTHFR, HPSE, AAK1, MX1) were found to be
significantly linked to sepsis prognosis.

• The study suggests these biomarkers influence sepsis prognosis through immune and coagulation
pathways.

2. Will the work be significant to the field and related fields?

Yes, this work is significant to the field.

Relevance to Sepsis Research:

• The study contributes to sepsis mortality prediction and risk stratification, both of which are
essential for clinical decision-making in ICUs.

• The identification of coagulation-inflammatory subtypes (CIS1 and CIS2) aligns with ongoing
efforts to better phenotype sepsis patients.

Relevance to Machine Learning & Explainable AI in Healthcare:

• SepsisFormer demonstrates that deep learning models can achieve superior predictive
performance while maintaining interpretability through XAI methods.

• SMART scoring provides a structured risk assessment tool that could be clinically deployed.

Potential Applications in Other Fields:

• The approach can be extended to other critical conditions (e.g., septic shock, ARDS, multi-organ
failure) where heterogeneity complicates prognosis.

• The explainability techniques used here could be applied to other AI-driven healthcare models.

Comparison to Existing Literature:

SepsisFormer’s AUC of 0.9301 is notably higher than previously reported AI-based sepsis models
(e.g., prior deep learning models in literature typically range from 0.85–0.91).

Risk stratification studies, such as those defining MARS1-4 or α , β , γ , δ subtypes, have not
explicitly linked coagulation-inflammatory markers to treatment response.

However, the study does not compare CIS1 and CIS2 to other existing sepsis subtyping
frameworks (e.g., MARS1-4, SRS1-2, hyper-/hypo-inflammatory states).

3. Does the work support the conclusions and claims, or is additional evidence needed?

Most claims are supported by strong experimental data.

Well-supported claims:

SepsisFormer outperforms existing scoring systems → Supported by ROC curves, AUC analysis,
external validation across multiple datasets.

SMART provides a clinically relevant stratification system → Clear association between
SMART’s risk levels and increasing mortality.

CIS2 patients benefit more from anticoagulants → Kaplan-Meier survival analysis demonstrates
significant mortality reduction in CIS2 patients receiving anticoagulation.

Less-supported claims:

CIS1/CIS2 are novel sepsis subtypes → The study does not compare them to previous subtyping
models (e.g., MARS1-4, α - δ).

Transcriptomic biomarkers are clinically relevant for sepsis prognosis → No external validation of
transcriptomic findings in an independent dataset.

Recommendation:

• Benchmark CIS1/CIS2 against prior subtyping frameworks.
• Validate transcriptomic biomarkers in a completely independent dataset.
4. Are there any flaws in the data analysis, interpretation, and conclusions?
Overall, the methodology is robust, but a few concerns require attention.
(A) External Validation of SepsisFormer Needs Clarification
How the data were split for training, validation, and testing is unclear.
Recommendation: Clearly define cross-validation strategy and how model generalizability was
assessed.
(B) SMART's Performance Varies Across Datasets
SMART's AUC ranges from 0.6222 to 0.7360—why does it perform worse in some cohorts?
Recommendation: Provide dataset-level breakdown and discuss factors influencing variability.
(C) Statistical Comparisons Are Missing for Model Performance
No DeLong's test or confidence intervals are reported to compare AUCs of different models.
Recommendation: Perform statistical comparisons to confirm that SepsisFormer is significantly
better than traditional methods.
5. Is the methodology sound? Does the work meet the expected standards in your field?
Yes, the study follows best practices for machine learning in medicine, but it requires
clarifications.
Areas for Improvement:
No external validation of transcriptomic biomarkers → Validation in an independent cohort is
needed.
No prospective validation of SepsisFormer in real-time ICU settings.
6. Is there enough detail provided in the methods for the work to be reproduced?
The methodology is well-documented but requires additional clarification in some areas.
Details that are clear and reproducible:
* Feature selection process for SMART.
* SHAP-based explainability analysis.
* Survival analysis methodology.
Details that require clarification:
* How were CIS1 and CIS2 validated against existing subphenotypes?
* How was model performance compared across datasets (cross-validation strategy)?
* What hyperparameters were used in training SepsisFormer?
Recommendation:
* Provide hyperparameter details for SepsisFormer.
* Clarify subphenotype validation and external benchmarking.
**Reviewer #3 (Remarks to the Author):**
1. What are the noteworthy results?
- A prognosis model for sepsis that leverages Transformer models and domain adaptation
technique that achieves high predictive accuracy and a scorecard model for risk stratification.

- The actual subphenotypes and risk levels classified by the findings from the model experiments,
and how these actually capture patient heterogeneity. Although, I could appreciate this
contribution later in the paper.

- An in depth analysis of and evidence for the heterogeneity of sepsis patients based on
subphenotypes and risk levels.

2. Will the work be of significance to the field and related fields? How does it compare to the
established literature? If the work is not original, please provide relevant references.

- The performance of existing AI prediction/prognosis models is mentioned in the introduction but
does not include the techniques used in those models, which can help the reader to appreciate how
the proposed models in this work contribute and differ from existing literature (in terms of input
data, model choice, training paradigms, etc.).

- In the domain adaptive generation, the method for oversampling is claimed to be novel but also
says that it uses existing domain adaptation methods. It is not clear what the novelty of the method
is. Besides, the domain adaptation is briefly mentioned in the abstract and discussed again in the
results under the Prognostic prediction performance of SepsisFormer based on
coagulation–inflammatory markers header. The last paragraph of the introduction can include this
component, as it seems relevant considering the heterogeneity and class imbalance problems it
addresses.

- The transcriptomic biomarker evidence analysis and study of anticoagulant treatment effect seem
crucial for better understanding sepsis, reveal insights about heterogeneity, and validate
algorithmic model development. However, I think that the presentation of these approaches can be
clarified because it wasn't clear what experiments actually took place in this study or were using
measurements from existing public datasets (E.g., the Heparin vs control studies, or PCR studies),
and how the biomarkers were actually selected.

3. Does the work support the conclusions and claims, or is additional evidence needed?

- As I was reading the paper, I struggled to understand the support for some of the claims in the
results section since it wasn't clear the criteria used for doing further classifications or analyses.
For example, how does further categorizing the predictors into clinically relevant ranges (not clear
how these are defined) facilitate clinical decision making? And then, why further analyzing the
two subtypes across the five key categories of sepsis predictors? This comment is also related to
the clarity presenting the methods.

- From my understanding and interpretation, I agree with the paper that explainable AI was used to
uncover patient heterogeneity, I am not sure if the model SepsisFormer was built to be explainable.
Many parts use modules from Transformers, a complex deep learning architecture, that as
mentioned, can be beneficial for modeling the complex relationships between the predictors. In
fact, the explainability technique used in the model directly is a post-hoc technique and this can be
acknowledge for clarity. Besides, it is mentioned that the model is heterogeneity-aware because of
the domain-adaptive generator, but this sounds like a data augmentation strategy for training.

- Claims around improving patient outcomes in the introduction: this has to be evaluated through
the application in clinical decision-making and translational research.

- What is meant by clinically relevant in this statement: “XAI method achieved clinically relevant
sensitivity and accuracy in prognostic prediction with only eight routine markers.”

4. Are there any flaws in the data analysis, interpretation and conclusions? - Do these prohibit
publication or require revision?
- Even though the components of SepsisFormer along with their functionality are explained, the
process for making these algorithmic choices is not clear. Was there any ablation study to select
the parameters or assess the contribution of some parts (e.g., the domain adaptation)? This will
provide stronger evidence for the models proposed in this paper.

- The discussion section currently presents medical literature and definitions in a list-like format.
While this demonstrates a strong understanding of sepsis from a clinical perspective (which is
valuable for algorithm development and interpretation of the results), the manuscript would
benefit from a more analytical approach. I recommend expanding the discussion to include
comparative analyses of the different models for the prognosis and risk stratification task, the
different domain adaptation strategies used, or any other pertinent comparisons.

- Considering the multiple insights found in this study, the paper is lacking an overall conclusion
about the approach followed. This can also be beneficial to reinforce the main message and
contribution of the paper.

5. Is the methodology sound? Does the work meet the expected standards in your field?
In general, the selected methods seem appropriate for the problems to be addressed in the paper.
However, I found the following concerns:
- In the introduction it says that the methods use only routine laboratory blood measurements and
how this provides an alternative to blood transcriptomic-based methods. Therefore, I found
confusing when later in the paper an analysis of transcriptomic biomarkers is introduced (which
involved additional methodological steps). Please clarify if I misunderstood this part.

- The analyses presented in the paper are extensive, but the rationale for each type is missing. For
example, why does it matter to see the contribution of the five categories of predictors? And their
associations? Why is it important to analyze the relationships between the 1-N transformer heads
and lab measurements? Including these justifications can also clarify the choice for the
visualization and/or analysis technique.
- Not clear how the subtypes alpha and beta in the variable-level explainability analysis differ
from the subphenotypes if they are both identified based on distinctions in mortality rates.

- Throughout the paper, it is stated that eight markers were identified (7 coagulation-inflammatory
measures + age) and used for prognosis modeling, risk stratification, and further identifying
subphenotypes, but the rationale for selecting this subset of predictors is unclear. The use of

routine coagulation-inflammatory measures is motivated in the introduction, but looking at Figure
2 it seems like there are more than 7 measures.

- For the evaluation of the risk stratification model, it could be beneficial to clarify what ground
truth is used for the evaluation of this task.

6. Is there enough detail provided in the methods for the work to be reproduced?

Considering the details and codes provided in the supplementary materials, probably yes.

However, I found additional experiments, like fine-tuning or pertaining the prognosis models,

which was never mentioned in the main manuscript.

**Reviewer #3 (Remarks on code availability):**

Code was not available in the submission. Some pseudo codes were presented in the
supplementary materials.

**Response to Reviewers' Comments**

We are incredibly grateful to the reviewers for their insightful and constructive
comments, which have been very helpful in further improving the manuscript and
enhancing its robustness, logic, and presentation of work highlights. The following is
our detailed and thorough response to each comment one by one. We hope to address
your concerns and obtain your approval.

**Reviewer #1 (Remarks to the Author):**

*Q1: The study looks at a significant problem. While prior literature has suggested*
*various algorithms that can predict sepsis very well and, at times, better than*
*physicians, this paper has three unique contributions. First, the model is built using*
*external datasets and validated using a separate hospital dataset which presents some*
*form of external validation for the model. Second, the development of a scoring*
*system presents a possible translational use of this algorithm in clinical environments*
*where the operating of real-time algorithms is not feasible. Third, it looks at*
*biomarkers and provides some explanatory attempts of predicting sepsis, which is*
*something challenging.*

**Response:** We sincerely appreciate your positive comments on the significance of our
study and the recognition of our three unique contributions. We are very grateful for
your constructive suggestions. We have carefully read them, discussed them, and
made one-on-one revisions and responses. All the relevant revised parts in the
manuscript have been marked in red. We hope to obtain your approval.

*Q2: There are, however, some points to consider in the revision.*
*The writing, although clear, does not highlight the key findings of the paper. The*
*results section focused excessively on the methods employed and did not provide*
*sufficient clinical explanation as to why these biomarkers matter in explaining sepsis*
*prediction. For example, SHAP plots are good for visualizing the importance of the*
*predictors in the model, but in the main body of the paper, more should be presented*

*to explain the clinical aspects of the findings. The methodological reporting should go*
*into the methods section which also needs further revisions.*

**Response:** We are very grateful for your constructive comments. We fully understand
your concerns regarding "highlighting the key findings of the paper" and the "results
section lacking in-depth explanation of biomarker clinical significance, with an
overemphasis on methods." These insights are crucial for improving the quality of our
manuscript. Therefore, we have implemented comprehensive revisions across
multiple sections of the manuscript, including the **Abstract, Introduction, Results,**
**and Discussion** sections, with a primary focus on the following aspects.

**1. Addition of Supplementary Figure 1.** We have included Supplementary Figure 1
to complement Figure 1 in the main text, providing an integrated overview of the
model architecture, key biomarker selection process, and the overall study framework.
This figure is intended to help readers quickly and intuitively grasp the scientific
rationale, methodological design, and experimental procedures of this study.

(New) Supplementary Figure 1. Integrated framework: An overview of models,
methods, biomarkers, and clinical applications in our study.

**2. Emphasizing Key Findings and Their Clinical Relevance.** The core findings of
our study are now more clearly summarized in the **Abstract**, **Introduction**, and
**Discussion** section. We particularly highlight the clinical importance of the identified
coagulation-inflammatory biomarkers for sepsis risk stratification, clinical
subphenotype identification, and guiding anticoagulant therapy, thereby underscoring
their translational value. (Page 2, 4–5, 12)

**Revised in the manuscript:**

“Our work, therefore, offers a novel set of simple, real-time executable tools for
sepsis heterogeneity, demonstrating the considerable potential to significantly enhance
sepsis clinical practice globally, particularly in resource-constrained healthcare

settings.” (Page 2)

[revised manuscript text omitted]

**4. Refining the Description of Methodology.** In line with your suggestion, we have
relocated the redundant methodological descriptions from the Results section to a
dedicated “Methods” section and thoroughly revised them. This section now contains
comprehensive technical details and statistical procedures, supplemented by detailed
information in **Supplementary Method 1, 3, 4**, to ensure greater rigor and logical
coherence.

We believe these revisions have significantly improved the manuscript's clarity,
clinical relevance, and scientific merit. We are grateful for your valuable suggestions
that contributed to these important improvements.

*Q3: I find the explanatory part of sepsis prediction something that is significant, but*
*this current form does not provide sufficient write-up in this area. I hope the authors*
*can provide more details that may contribute to the clinical understanding of sepsis*

*onset.*

**Response:** We truly appreciate your insightful comment regarding the explanatory
part of sepsis prediction. We completely agree that more detailed information in this
area is crucial for enhancing the clinical understanding of sepsis onset. In response to
your comment, we enriched and improved the relevant parts as follows:

**1.We refined the explainability methods used in our study.**

**Revised in the manuscript:**

“Multi-view explainability analyses were conducted from three perspectives: two
post-hoc approaches (cluster-informed EHR variable-level and AI model-level) and a
transcriptomic-level analysis inspired by multi-omics principles. To select key
categories that effectively mirror sepsis pathophysiology and are practical for clinical
application, we employed unsupervised clustering methods, including Gaussian
Mixture Model (GMM), MiniBatchKMeans, K-means, Hierarchical Agglomerative
Clustering (HAC), and Birch. Chord diagrams and radar plots enhance the
interpretability of unsupervised clustering results by revealing the variable patterns
defining each cluster. Clinical expertise and Pearson's correlation matrix were
employed to analyze the associations between individual laboratory measurements
within each category.” (Page 14-15)

**2.Incorporate the Research Framework for Cluster and Subphenotype Analyses.**

We've further clarified the research ideas and the objectives of the study
regarding the use of interpretable methods. This framework highlights explicitly the
key distinctions in purpose and methodology between non-clinical and clinical
analyses (Supplementary Fig.2).

**(New) Supplementary Figure 2.** Research framework for cluster and subphenotype.

3. Elaboration on Coagulation-Inflammatory Markers in Cluster Analysis

(1) Enriching Chart Interpretation Details: In the revised manuscript, we added
 more detailed explanations when describing the chord diagrams and radar charts. For
 the chord width of coagulation-inflammatory markers in the chord diagram, we
 explained how this visual feature directly reflects the significant difference between
 the two clusters. We added the content in the results section as follows:

Revised in the manuscript:

[revised manuscript text omitted]

**4.Strengthening the Presentation of Validation Content Related to** 547 **Transcriptomic Biomarkers**

To validate the conclusion of the key role of coagulation-inflammatory markers in
sepsis heterogeneity, we have identified additional transcriptomic biomarker evidence
from coagulation-related genes (CRGs), inflammation-related genes (IRGs), and
disseminated intravascular coagulation (DIC)-related genes. We have highlighted the
clinical relevance of transcriptomic biomarkers in sepsis diagnosis and prognosis.
Specifically, we discussed the diagnostic potential of CD59, SERPINB2, CFD, and
P2RX1 as independent markers and provided guidance on their application for early
detection and dynamic monitoring to assess treatment response and patient outcomes.

In the revised manuscript, we elaborated on the internal connection between these

transcriptomic biomarkers and coagulation-inflammatory markers and added the
related content in the Results section:

**Revised in the manuscript:**

“Our transcriptomic-level explanation provides complementary evidence for the
critical role of coagulation-inflammatory markers in sepsis heterogeneity, initially
identified through EHR-based variable and model explanations. We specifically
examined coagulation-related genes (CRGs), inflammatory-related genes (IRGs), and
disseminated intravascular coagulation (DIC)-related genes (Fig. 2h and
Supplementary Fig. 6-7). This investigation reaffirmed the significance of these
markers and identified five key prognostic biomarkers: STAT5B, MTHFR, HPSE,
AAK1, and MX1. Patients stratified into a high-risk group based on these
biomarkers exhibited significantly higher mortality. Notably, MTHFR, AAK1, and
MX1 expression levels were elevated in this high-risk group, suggesting their
influence on sepsis prognosis potentially through modulation of immune and
coagulation pathways (Supplementary Table 8). The diagnostic efficacy of these five
genes was further validated in external dataset GSE54514 (Supplementary Fig. 8).
Furthermore, transcriptome analysis confirmed the diagnostic relevance of four genes:
CD59, P2RX1, CFD, and SERPINB2 (Supplementary Table 9-10). These markers
demonstrated robust performance as independent diagnostic biomarkers for sepsis.
Their diagnostic efficacy was successfully validated in external datasets GSE26440
and GSE95233 (Supplementary Fig. 9). RT-PCR analysis of PBMCs from 29 sepsis
patients and 11 healthy controls revealed significant differential expression of CFD
and P2RX1 ($p < 0.05$), with greater variability observed in the sepsis group. Therefore,
these transcriptomic findings further explained the important roles of
coagulation-inflammation related indicators in the diagnosis and prognosis of sepsis
from multiple perspectives.”(Page 6)

**(New) Supplementary Figure 8.** The prognostic and diagnostic efficacy of STAT5B,
 MTHFR, HPSE, AAK1, and MX1 in an external cohort. From GSE54514. a.
 prognostic AUC curves; b. Expression levels of STAT5B, MTHFR, HPSE, AAK1,
 and MX1. * $p < 0.05$, *** $p < 0.001$ vs. Healthy control.

**(New) Supplementary Figure 9.** The diagnostic efficacy of CD59, SERPINB2,
 CFD, and P2RX1 in external cohorts. From GSE26440, and GSE95233. a. and c.,
 AUC curves; b. and d, Expression levels of CD59, SERPINB2, CFD, and P2RX1.
 * $p < 0.05$, *** $p < 0.001$

**Q4:** In terms of the reporting of the methods, here, less is perhaps more. The authors
 have presented a multitude of methods to support their results. Although in general, I
 do believe in the findings from a reviewer perspective, the large number of methods

employed with limited reporting makes it exacting for the reviewer to establish the
correctness of the execution of these methods. To illustrate, in the text, the authors
wrote “[...] radar plots of subtypes identified by unsupervised clustering methods,
including GMM, MiniBatchKMeans, K-means, hierarchical agglomerative clustering
(HAC), and Birch [...]”. The writing here presents more of a report of a collection of
methods with limited ability for the reviewers to verify and perhaps add limited value
to the paper other than to signal to the review team that extensive work has been put
into the paper. A similar writing style is pervasive in the entire methods section, e.g.,
“[...] MIMIC-III and MIMIC-IV are merged, and SMOTE is employed to balance the
class distribution [...]”
For example, what is the extent of SMOTE (percentage, number of neighbors used,
etc.)? Even though I downloaded all the datasets in this submission, reading this
article does not permit any reviewer to completely comprehend the inner workings of
the analysis based on the reporting.

**Response:** We sincerely appreciate your detailed and constructive feedback regarding
the reporting of our methods. We understand that the current presentation has made it
challenging for you to verify the correctness of the method execution, and we
addressed these concerns in the revised manuscript. And we hope that the new
framework diagrams (Supplementary Fig.1 and Fig.2) listed above will help us
present our methods and results more clearly.

1. Specifically, we **added Supplementary Figures 1 and 2** to provide an
overview of the models, methods, biomarkers, clinical applications, and the research
framework for clustering and subphenotype identification. As shown in
Supplementary Figure 2, the unsupervised clustering methods served two key
purposes: (1) selecting suitable clustering algorithms based on inter-cluster mortality
differences for subphenotype identification, and (2) assessing the category stability
across different clustering methods. We agree that the original description resembled a
technical report, and have revised this section to improve clarity and readability.

**Revised in the manuscript:**

“Five different unsupervised methods-Gaussian Mixture Model (GMM),
MiniBatchKMeans, K-means, Hierarchical Agglomerative Clustering (HAC), and
Birch—generated highly aligned clusters, a consistency across distinct approaches
that confirms the robustness and validity of the two identified clusters. Since these
data-driven clusters lack inherent clinical interpretation, we statistically analyzed
subgroup mortality. The GMM-derived α and β clusters exhibited the most significant
mortality difference (32.09% vs 17.62%, respectively), indicating GMM's superior
effectiveness for identifying patient subgroups with divergent mortality risks.(Fig. 2f,
Supplementary Fig. 3).”(Page 5)

“This study employs GMM for subphenotype identification, as the patient
subgroups derived from GMM show the most significant differences in mortality rates
(shown in Section Explainability Analysis), indicating greater clinical
relevance.”(Page 7)

2. We have thoroughly revised **Supplementary Method 1: Domain-Adaptive**
**Generator for Fine-Tuning**. This section now includes detailed analyses of
multi-center covariate distribution shifts, theoretical justifications, and comprehensive
implementation procedures. The main additions cover:

(1) **Rationale for Domain Adaptation in Prognostic Modeling**, an explanation
of why domain adaptation is crucial in handling data heterogeneity across multiple
centers.

(2) **Theoretical Analysis of Covariate Distribution Shifts**, providing insights
into the impact of such shifts on model performance and generalization;

(3) **Detailed Implementation Steps**, describing how domain adaptation is
practically realized in our framework.

Additionally, in the implementation of the MMID-SMOTE algorithm, we have
clearly specified the key parameters necessary:

**Step 5: Customized SMOTE Sample Generation** Synthetic samples are
generated via SMOTE under these constraints, with the following parameters

optimized for medical data context.

Number of nearest neighbors: 4.

Minority class oversampling ratio: 1:1 (achieving balanced class distribution).

Random seed: Fixed for reproducibility.

Parallel jobs: 3 (accelerating computation).

**Q5:** *I suggest the authors do one of the two. 1) In the main paper, report a shorter list*
*of analyses employed, but do that in a more comprehensive and clearer manner,*
*complete with a STROBE or modified CONSORT diagram to showcase the data used.*
*This presentation will lend more credibility to the results. If you want to showcase*
*additional analyses reported here, maybe provide some additional supplement you*
*might want to host on GitHub. These supplementary analyses may have minute*
*computational errors or bugs—which is normally the case, but it does not fall within*
*the purview of NComms review process. 2) Continue with your existing reporting of*
*all the collection of analyses, but in your next review, you perhaps will have to*
*consider providing more details of how the analyses for each technique is done,*
*including the software codes in an executable format. This will also make the review*
*process more exacting.*

**Response:** Thank you for your thoughtful suggestions regarding the presentation of
our analyses.

STROBE (STrengthening the Reporting of OBservational studies in
Epidemiology) is a guideline for clinical observational studies. Since our research is
not a pure cohort clinical study (involving interdisciplinary, basic, and clinical
research areas), yet it contains substantial clinical components, we have completed the
STROBE checklist and improved the manuscript according to relevant guidelines.

While we appreciate the clarity with which you have outlined the two valuable
approaches, and the following is also the content of the manuscript improvement
work we have carried out according to Option 2 you provided, mainly further
improving the methodology, results, and discussion sections, etc. You can also find
them in the responses to the comments above.

We believe that reporting the full range of analyses conducted is essential for
demonstrating the comprehensiveness of our study. In response to your feedback, we
are fully committed to enhancing the level of detail in our methodology section for the
following review. For each analytical technique employed, we provided an in-depth
description of the procedures, including specific parameter settings, data
preprocessing steps, and the rationale behind each choice.

To ensure transparency and reproducibility, we will also include executable codes
for all analyses as supplementary materials. These codes are well-documented, with
clear annotations explaining the functionality of each segment. We understand that
this may make the review process more rigorous, but we view it as a necessary step to
uphold the integrity of our research.

While we made every effort to ensure the accuracy of all our work, we also
recognized that the scope of the Nature Communications review process may not
allow for a detailed examination of these additional components. We, therefore, took
extra care in validating our analyses and clearly indicated any exploratory or
supplementary nature of certain parts of our work.

Thank you again for your guidance. We are confident that these improvements will
strengthen our manuscript and facilitate a more thorough review.

**Reviewer #1 (Remarks on code availability):**

*I have downloaded the datasets, but the current reporting make it onerous and not*
*possible for one to review the analysis in detail.*

**Response:** We deeply apologize for the inconvenience and difficulty you encountered
while attempting to review the analysis in detail despite having downloaded the
datasets. We have now included the relevant codes, along with detailed instructions
and videos on how to run them. To promote transparency, reproducibility, and clinical
applicability, we have made the following resources publicly available:

**1. Interactive Web-based Platform** (<http://smartsepsis.org.cn>): We developed an
open-access, user-friendly online platform for sepsis subphenotyping and risk

stratification. This tool allows end-users, such as clinicians and researchers, to
perform real-time subphenotype identification and risk level prediction by uploading
their own data without requiring additional coding or technical expertise.

**2. Source Code and Datasets** (<https://github.com/zhuli19031218/SepsisFormer>):
All raw data, preprocessed datasets, and fully documented source code are available
in our GitHub repository. These resources enable independent verification, replication,
and extension of our study.

**3. Code Execution Tutorial** (<https://doi.org/10.5281/zenodo.15634369>): A
detailed step-by-step video tutorial is provided to guide users through the complete
process of reproducing the main results presented in this study, ensuring ease of use
even for non-technical users.

These provided resources allow for replication and application of our method. We
believe these improve the paper's quality and transparency.

We sincerely hope the revised manuscript now meets with your approval.

Thank you once again for your time and effort. We truly appreciate your valuable
suggestions, especially regarding the clinical explanations and overall writing, which
have greatly enhanced the quality of our work.

Hope everything goes well with you!

**Reviewer #2 (Remarks to the Author):**

**Q1:** *What are the noteworthy results?*

*This study has some key findings.*

*(A) High Prognostic Accuracy of SepsisFormer*

• *SepsisFormer (Transformer-based model) achieves an AUC of 0.9301,*
*outperforming traditional risk scoring models like qSOFA, SOFA, and APACHE II.*

• *The model demonstrates generalizability across multiple cohorts (MIMIC-III,*
*MIMIC-IV, eICU-CRD, and a local ICU dataset).*

*(B) SMART Risk Stratification Model Improves Sepsis Classification*

• *SMART (a scoring system based on seven coagulation-inflammatory markers and*
*patient age) achieved an AUC of 0.7360, surpassing most established clinical scoring*
*systems.*

• *The study identifies four risk levels (Mild, Moderate, Severe, Dangerous), with clear*
*mortality risk stratification (~5%, 15%, 30%, and 50% mortality rates, respectively).*

*(C) Identification of Two Sepsis Subphenotypes (CIS1 and CIS2) with Different*
*Outcomes*

• *CIS2 patients exhibited higher mortality rates (~32.23%) compared to CIS1*
*(25.84%), and a stronger inflammatory-coagulation response.*

• *The study demonstrates that patients with CIS2 benefit more from anticoagulant*
*therapy (e.g., heparin), while CIS1 shows moderate benefits.*

*(D) Transcriptomic Analysis Identifies Sepsis Biomarkers*

• *Five transcriptomic biomarkers (STAT5B, MTHFR, HPSE, AAK1, MX1) were found*
*to be significantly linked to sepsis prognosis.*

• *The study suggests these biomarkers influence sepsis prognosis through immune and*
*coagulation pathways.*

**Response:** We sincerely appreciate your positive comments on our study and the
recognition of our unique contributions. We are very grateful for your constructive
suggestions. We have carefully reviewed them and made point-to-point revisions. All
the revised parts in the manuscript are marked in red. We hope for your approval.

**Q2:** *Will the work be significant to the field and related fields?*

*Yes, this work is significant to the field.*

*Relevance to Sepsis Research:*

• *The study contributes to sepsis mortality prediction and risk stratification, both of*
*which are essential for clinical decision-making in ICUs.*

• *The identification of coagulation-inflammatory subtypes (CIS1 and CIS2) aligns*
*with ongoing efforts to better phenotype sepsis patients.*

*Relevance to Machine Learning & Explainable AI in Healthcare:*

• *SepsisFormer demonstrates that deep learning models can achieve superior*
*predictive performance while maintaining interpretability through XAI methods.*

• *SMART scoring provides a structured risk assessment tool that could be clinically*
*deployed.*

*Potential Applications in Other Fields:*

• *The approach can be extended to other critical conditions (e.g., septic shock, ARDS,*
*multi-organ failure) where heterogeneity complicates prognosis.*

• *The explainability techniques used here could be applied to other AI-driven*
*healthcare models.*

*Comparison to Existing Literature:*

*SepsisFormer's AUC of 0.9301 is notably higher than previously reported AI-based*
*sepsis models (e.g., prior deep learning models in literature typically range from*
*0.85–0.91).*

*Risk stratification studies, such as those defining MARS1-4 or α , β , γ , δ subtypes, have*
*not explicitly linked coagulation-inflammatory markers to treatment response.*

*However, the study does not compare CIS1 and CIS2 to other existing sepsis*
*subtyping frameworks (e.g., MARS1-4, SRS1-2, hyper-/hypo-inflammatory states).*

**Response:** Thank you very much for your constructive comments. CIS1 and CIS2
subphenotypes are one of the three important achievements of this study
(SepsisFormer, SMART, and Subphenotype). Our subphenotypic classification is
capable of distinguishing between high or low systemic inflammatory response Index
(SIRI), prognosis of death, and differences in the benefits of anticoagulant treatment.

(New) Supplementary Figure 1. Integrated framework: An overview of models,
 methods, biomarkers, and clinical applications in our study.

**Indeed**, risk stratification studies, such as those defining MARS1-4 or α , β , γ , δ
 subtypes, have not explicitly linked coagulation-inflammatory markers to treatment
 response.

The previous work (MARS1-4, SRS1-2, hyper-/hypo-inflammatory states, et al.)
 has been done very significantly. Our results can be regarded as being inspired by
 their work, or we can say that we are very grateful for their work. Only by standing on
 their shoulders can we make further progress on the basis of subtype classification.
 Furthermore, through SepsisFormer and clinical interpretation, transcriptomic
 interpretation, etc., we identified coagulation-inflammation indicators and patient age
 as the indicator combination for our risk stratification and subphenotypic

classification, which also inspired us to analyze the differences in their
anticoagulation benefits. The results show that there are indeed differences between
them. Moreover, anticoagulant therapies such as heparin are also a focus of great
concern in the clinical practice of sepsis at present. However, there is no quantifiable
or precise study worldwide to propose which types of sepsis patients or when
anticoagulant therapy can benefit them. Therefore, this is also one of the important
contributions of this study.

Here, we need to express our sincere gratitude once again to **Professor Craig**
**Coopersmith**, the former chair of the Society of Critical Care Medicine in the United
States, for his guidance on this research.

The main purpose of our article is to develop and validate a new risk
stratification and subphenotypic classification tool, as well as an analysis of the sepsis
population benefiting from anticoagulant therapy based on this. The reasons why our
study did not involve substantive comparisons with the previous MARS1-4/ SRS1-2/
α , β , γ , δ / hyper-/hypo-inflammatory states, etc., are as follows:

Firstly, as a newly developed tool, it's challenging to compare so many tools in
one study. Some well-established tools didn't conduct extensive comparisons in the
early stages of their development, which could be partly due to differences in research
objectives and practical workload considerations. For example, MARS 1-4 developed
by the team of Professor Tom van der Poll (*Lancet Respir Med.* 2017 Oct;
5(10):816-826), α , β , γ , δ by the team of Professor Christopher W. Seymour (*JAMA.*
2019 May 28;321(20):2003-2017; *Crit Care.* 2023 Jun 15; 27(1):236), and SRS1-2 by
the team of Professor Julian C. Knight. (*Lancet Respir Med.* 2016 Apr; 4(4):259-71).

Secondly, because MARS 1-4 is an endotypic study, where they have proposed
typing suggestions for eight transcriptional genes, this approach requires a relatively
large number of transcriptomic tests separately. The subtype classification of α , β , γ ,
and δ also involves more than 30 indicators. Presently, even with maximal effort to
extend research time in this aspect, operational limitations would still hinder the
progress, therefore significantly impacting the publication timeline of our results. We
earnestly hope for your understanding, dear professor.

Thirdly, precisely because of the above reasons, the team of Prof. Christopher W.
Seymour and Prof. Lonneke A. van Vught recently conducted A comparative study of
various subtypes (*Intensive Care Med* (2023) 49:1360–1369). They compared ARDS
subtypes (Hyper-/ Hypo-inflammatory) vs. MARS and SRS subtypes, and SENCEA
subtypes, it was found that the overlap ratio between them was not ideal. This might
still be related to a combination of factors such as the main goals of the development
of each subtype classification, the different AI algorithms or models selected, and the
differences in the pathophysiological connotations reflected by each indicator. So, we
cannot say that our classification must have a good overlap with theirs. However, we
have simplified the detection indicators with advanced algorithms, launched the
SMART scorecard, developed an open-access web-based sepsis subphenotype and
SMART platform that enables clinicians to real-time interactively evaluate septic
patients' risk level and subphenotype (<http://smartsepsis.org.cn>), and also put forward
the viewpoint of the beneficiary population of anticoagulant therapy. All these are the
results of innovation and forward development based on previous research. In
addition, we believe that with the rapid development of this field, subsequent research
will surely build upon and learn from the research achievements and experiences of
our predecessors and ourselves, and will once again bring forth new ideas. Subsequent
research will also gradually bridge the gap of differences in the classification of
various subtypes. Science is moving forward. We are looking forward to a better
future, and this day is sure to come.

Last but not least, although our study did not conduct a comparison with the
various subtypes mentioned above, we also discussed in detail the results and
differences of these subtypes in the discussion (Paragraphs 3 and 4 of the discussion
section). We believe that this can at least provide some references for our readers.

Above figure refers to: Intensive Care Med (2023) 49:1360–1369.

**Q3:** *Does the work support the conclusions and claims, or is additional evidence*
*needed?*

*Most claims are supported by strong experimental data.*

*Well-supported claims:*

*SepsisFormer outperforms existing scoring systems → Supported by ROC curves,*
*AUC analysis, external validation across multiple datasets.*

*SMART provides a clinically relevant stratification system → Clear association*
*between SMART's risk levels and increasing mortality.*

*CIS2 patients benefit more from anticoagulants → Kaplan-Meier survival analysis*
*demonstrates significant mortality reduction in CIS2 patients receiving*
*anticoagulation.*

*Less-supported claims:*

**Q3.1** *CIS1/CIS2 are novel sepsis subtypes → The study does not compare them to*
*previous subtyping models (e.g., MARS1-4, α - δ).*

*Recommendation:*

*Benchmark CIS1/CIS2 against prior subtyping frameworks.*

**Response:** Thanks for your valuable comment on our research.

We are pleased that our findings have the potential to contribute meaningfully to
ongoing discussions, and we firmly believe that our work holds great significance for
the field of sepsis research, as well as for its intersections with machine learning and
explainable artificial intelligence (AI) in the healthcare domain. Our study offers
practical tools like SepsisFormer for mortality prediction (with an AUC of 0.9301,
surpassing prior models) and SMART for risk stratification, while identifying
coagulation-inflammatory subtypes (CIS1 and CIS2) to enable personalized treatment.
The methodologies can be applied to other critical conditions, and the XAI techniques
used with SepsisFormer have broader utility.

As we presented the new framework diagram above (Supplementary Fig. 1), this
study primarily aimed to achieve three goals: SepsisFormer/SMART/subphenotypic
classification and the analysis of differences in the benefits of anticoagulant therapy
based on these classifications. Meanwhile, we developed an open-access web-based
sepsis subphenotype and SMART platform that enables clinicians to interactively
evaluate septic patients' risk level and subphenotype (<http://smartsepsis.org.cn>).

Nevertheless, we admit that we haven't compared CIS1 and CIS2 with other
established sepsis subtyping frameworks. The main reasons are as stated above
(please see the response to Q2 comment). Meanwhile, in the discussion section, we
provided a comparison and analysis of subtype classification. Hope to gain an
in-depth understanding and approval.

**Q3.2** *Transcriptomic biomarkers are clinically relevant for sepsis prognosis → No*
*external validation of transcriptomic findings in an independent dataset.*

*Recommendation:*

*Validate transcriptomic biomarkers in a completely independent dataset.*

**Response:** Thank you for acknowledging that our study adheres to best practices in
machine learning for medicine. We wholeheartedly agree with your assessment
regarding the need for clarifications and additional validations.

**1. Transcriptomic Biomarker Validation:** Recognizing the critical importance of
external validation in biomarker research, we rigorously evaluated the five
transcriptomic biomarkers (STAT5B, MTHFR, HPSE, AAK1, MX1) identified in our
study using an independent sepsis cohort from the GSE54514 dataset. Critically, we
confirmed their prognostic efficacy and further evaluated their diagnostic performance.
ROC curve analysis demonstrated that the biomarkers' diagnostic predictive ability
(AUC) closely aligned with our original dataset. Significant overexpression of AAK1
and MX1 ($p < 0.05$) was additionally observed in sepsis patients compared to controls.
While methodological harmonization across studies was maintained where feasible,
these outcomes, particularly the consistent diagnostic performance and differential
expression patterns, collectively validate the biological relevance of these biomarkers,
with detailed AUC metrics visualized in Supplementary Figure 8.

**(New) Supplementary Figure 8.** The prognostic and diagnostic efficacy of STAT5B,
MTHFR, HPSE, AAK1 and MX1 in an external cohort. From GSE54514. a.
prognostic AUC curves; b. Expression levels of STAT5B, MTHFR, HPSE, AAK1
and MX1. * $p < 0.05$, *** $p < 0.001$ vs. Healthy control.

**(New) Supplementary Figure 9.** The diagnostic efficacy of CD59, SERPINB2,
 CFD, and P2RX1 in external cohorts. From GSE26440, and GSE95233. a. and c.,
 AUC curves; b. and d, Expression levels of CD59, SERPINB2, CFD, and P2RX1.
 * $p < 0.05$, *** $p < 0.001$

In the revised manuscript, we elaborated on the internal connection between these
 transcriptomic biomarkers and coagulation-inflammatory markers and added the
 related content in the Results section:

**Revised in the manuscript:**

[revised manuscript text omitted]

**Q4:** *Are there any flaws in the data analysis, interpretation, and conclusions?*

*Overall, the methodology is robust, but a few concerns require attention.*

**Q4.1** *External Validation of SepsisFormer Needs Clarification*

*How the data were split for training, validation, and testing is unclear.*

*Recommendation: Clearly define cross-validation strategy and how model*
*generalizability was assessed.*

**Response:** Thank you for highlighting the areas that require clarification. Your
feedback has been invaluable in helping us improve the clarity and rigor of our work.
We acknowledge that the current manuscript lacks sufficient detail in this section.

1. **To address this concern**, we incorporated the following clarifications in the
revised version. We detailed the partitioning of datasets (MIMIC-III, MIMIC-IV,
eICU-CRD, and the local ICU dataset). The pretraining, fine-tuning, and validation

procedures, utilizing these diverse datasets, are also explained. Furthermore, we
outline considerations for maintaining representativeness across subsets (e.g.,
ensuring balanced class distributions for mortality outcomes). This information has
been incorporated into the methods section.

**Revised in the manuscript:**

“(e) Pretraining Procedure. Cohort 1 (MIMIC-III and eICU-CRD) was partitioned 7:3
for training and internal testing. From this training data, the pretrained SepsisFormer
model was initialized, establishing its core architecture, weights, and hyperparameters.
SMOTE was applied for data augmentation to address class imbalance and enhance
model robustness during this phase. (f) Fine-tuning Procedure. Subsequently, for
domain adaptation and enhanced generalizability, SepsisFormer underwent
fine-tuning. Cohort 1 served as the source domain, with fine-tuning specifically
targeting external validation cohorts: Cohort 2 (MIMIC-IV) and Cohort 3 (Local ICU)
as the target domains. A Domain-adaptive Generator was employed for data
augmentation during this transfer learning stage, optimizing performance in new
clinical center.” **(Page 14-15)**

2. **To evaluate generalizability**, we tested SepsisFormer on multiple
independent cohorts with different patient demographics and clinical practices. In the
revised manuscript, we presented additional analyses to quantify generalizability. For
example, we reported calibration plots and discrimination metrics (e.g., AUC) for
each external dataset, demonstrating consistent performance across diverse settings.
We also discussed any performance differences and potential reasons for these
variations. We also revised the titles of the figures and tables to clarify
methodological comparisons. For example, **key revisions** are:

**“Supplementary Table 6.** Performance comparison of the pre-trained
SepsisFormer and baseline models for prognostic prediction on the internal test set
(Cohort 1: 7,789 septic patients from MIMIC-III and eICU-CRD; 70% training, 30%
test).”

“**Supplementary Table 7.** Performance comparison of the fine-tuned
SepsisFormer and baseline models for prognostic prediction on the two external
validation cohorts (Cohort 2: 4,191 septic patients from MIMIC-IV; and Cohort 3:428
septic patients from a local ICU), using 10-fold cross-validation.”

**Q4.2 SMART's Performance Varies Across Datasets**

*SMART's AUC ranges from 0.6222 to 0.7360—why does it perform worse in some*
*cohorts?*

*Recommendation: Provide dataset-level breakdown and discuss factors influencing*
*variability.*

**Response:** Thank you for pointing out the variability in SMART's performance. The
AUC range of 0.6222 to 0.7360 for SMART does not reflect performance across
different cohorts. Instead, these values derive from our local data, where various
scoring models were evaluated. As shown in Fig. 3d, the highest AUC (0.7360)
corresponds to SMART. These results suggest that the SMART outperforms the
existing models in external validation. To clarify this issue, we revised the statement
as follows:

**Revised in the manuscript:**

“In the local ICU cohort, the SMART demonstrated the highest predictive accuracy,
with an AUC of 0.7360. For the five established scoring systems, including SOFA,
qSOFA, LIP, APACHE II, and SIRS, the AUCs are 0.6833, 0.6441, 0.6431, 0.6222
and 0.5428 respectively (Fig. 3d). ” **(Page 8)**

**Q4.3 Statistical Comparisons Are Missing for Model Performance**

*No DeLong's test or confidence intervals are reported to compare AUCs of different*
*models.*

*Recommendation: Perform statistical comparisons to confirm that SepsisFormer is*
*significantly better than traditional methods.*

**Response:** Thank you for highlighting the absence of statistical comparisons for

model performance. In the revised manuscript, we addressed this issue by conducting
comprehensive statistical analyses.

We supplemented our analysis by performing DeLong’s test to statistically
compare the AUCs of different models, confirming that SepsisFormer significantly
outperforms those methods. The revised manuscript now includes a detailed
description in the **Statistical Analysis** of the **Methods, legends** for **Figure 2**. Detailed
statistical results are also provided on the **Source Code and Datasets website**.

**Revised in the manuscript:**

“DeLong’s tests were performed to statistically compare the AUCs of different
models using `Delong_test` from the `MLstatkit.stats` package, in conjunction with
`roc_auc_score` from `scikit-learn`.” (**Page 16**)

“ROC curves illustrating the prognostic prediction ability of SepsisFormer using 36
sepsis predictors is superior to the based models on the dataset MIMIC-III ($p<0.01$,
except SepsisFormer vs. LSTM $p=0.08$,) and MIMIC-IV ($p<0.01$, except
SepsisFormer vs. GRU $p=0.41$) . All p-values by DeLong's test. ” (**Legends for
Figure 2, Page 24**)

*Q5: Is the methodology sound? Does the work meet the expected standards in your
field?*

*Yes, the study follows best practices for machine learning in medicine, but it requires
clarifications.*

*Areas for Improvement:*

*Q5.1 No external validation of transcriptomic biomarkers → Validation in an
independent cohort is needed.*

**Response:** Details are provided in our responses to Q3.2.

*Q5.2 No prospective validation of SepsisFormer in real-time ICU settings.*

**Response:** Thank you very much for your comment. It has been conducive to us. We
have already conducted relevant supplementary research work.

**1. Prospective Validation of SepsisFormer:** Regarding SepsisFormer, we

understand that prospective validation in real-time ICU settings is crucial for
translating our findings into clinical practice. Our current retrospective analysis across
multiple cohorts (MIMIC-III, MIMIC-IV, eICU-CRD, and a local dataset) showcases
the model’s generalizability. Therefore, SepsisFormer's reliability and its central role
in identifying key coagulation-inflammatory biomarkers make it indispensable for
developing the final SMART model and deriving subphenotyping results. As the
SMART model and subphenotyping represent this study's core findings with clear
clinical translation application value, we performed external validation through
prospective, real-time cohort analyses to verify their performance on SMART risk
stratification and subphenotype classification, as well as prognosis prediction.

**New Supplementary Figure 1.** Integrated framework has showed above. (See
the response to Q2)

Due to our open-access web-based sepsis subphenotype and SMART platform
(<http://smartsepsis.org.cn>), which is better suited for real-time applications. Here, we
have added a prospective, real-time, external cohort validation study. We monitored
40 patients in real-time (the first day of being enrolled in this cohort), and the results
were consistent with the model's predictions, as follows:

**Revised in the manuscript:**

“Furthermore, for the real-time and external validation of the risk stratification and
subphenotypic classification, with the approval of the Ethics Committee of
Chongqing University Affiliated Central Hospital (Chongqing Emergency Medical
Center) (ID: 2025-55), a prospective observational study on sepsis risk stratification
and subphenotypic classification via our risk stratification and subphenotype platform
was conducted. The endpoint event was to observe the actual 28-day mortality rate
and obtain the SMART score and subphenotype at the time of enrollment. After
obtaining written informed consent, a total of 40 patients with sepsis were enrolled
from March 24, 2025, to April 28, 2025.” (Page 13-14)

“Last but not least, clinicians could conduct real-time risk stratification and
subphenotypic classification of sepsis based on the SMART scorecard or our
open-access sepsis subphenotype and SMART platform (<http://smartsepsis.org.cn>).

Here, we conducted risk stratification, phenotypic classification and prognosis
 prediction for 40 sepsis patients locally admitted from March 21, 2025, to April 27,
 2025 (external observational verification only) (Fig.4e-h, and Supplementary case
 materials 1 and 2). The results showed that the proportions of patients in the four risk
 levels of mild, moderate, severe, and dangerous were 12.5%, 30%, 32.5% and 25%
 respectively (Fig. 4e), and their 28-day actual mortality rates were 0%, 16.7%, 38.5%
 and 90% respectively (Fig. 4f). The overall mortality rate of the CIS1 subphenotype
 was 33.3%, significantly lower than 53.8% of CIS2, and the trend was the same at
 different risk levels (Fig. 4g and 4h). The mortality rate at the dangerous level (this
 external observational cohort) was a little higher than the predicted rate of the model
 in this study (approximately 50%), as well as simultaneously increased the overall
 mortality rates of CIS1 and CIS2, which might be related to the small sample size, but
 the overall trend was basically consistent. Therefore, clinicians can utilize the
 SMART scorecard or our open-access platform to conduct real-time risk classification
 and subphenotyping of patients, enabling objective and accurate assessment of sepsis
 patients to intervene as early as possible and improve their prognosis.” (Page 9)

Additional subgraphs in Revised Figure 4.

In future work, we plan to collaborate with multi-center clinical partners to
 deploy SepsisFormer in ICUs, collecting data on model predictions and actual patient
 outcomes in real time. This will allow us to further evaluate the model’s performance
 under dynamic clinical conditions, including its timeliness in detecting high-risk
 patients and its impact on clinical decision-making.

**Q6:** *Is there enough detail provided in the methods for the work to be reproduced?*
*The methodology is well-documented but requires additional clarification in some*
*areas.*

*Details that are clear and reproducible:*

** Feature selection process for SMART.*

** SHAP-based explainability analysis.*

** Survival analysis methodology.*

*Details that require clarification:*

**Q6.1** *How were CIS1 and CIS2 validated against existing subphenotypes?*

*Recommendation:*

*Clarify subphenotype validation and external benchmarking.*

**Response:** Thank you for your thorough assessment of our methodology. Your
feedback regarding areas that need clarification is invaluable, and we address these
points comprehensively here.

Regarding the validation of CIS1 and CIS2 subphenotypes, In the revision, we
will explicitly describe how these subphenotypes were defined. We acknowledge the
need for more context. Nevertheless, we admit that we have not compared CIS1 and
CIS2 with other established sepsis subtyping frameworks. The main reasons are as
stated above (**please see the response to Q2 comment**). Meanwhile, in the discussion
section, we provided a comparison and analysis of subtype classification. Hope to
gain an in-depth understanding and approval; thank you very much.

**Q6.2** *What hyperparameters were used in training SepsisFormer? And how was*
*model performance compared across datasets (cross-validation strategy)?*

*Recommendation:*

*Provide hyperparameter details for SepsisFormer.*

**Response:** Thank you very much for your constructive recommendations.

We provide hyperparameter details for SepsisFormer used in training, including
learning rate:0.0010, batch size: 5000, number of epoch: 1400, dropout rate: 0.1000,
number of attention head: 8, and depth of the Transformer architecture:8. This

transparency will enable other researchers to replicate our model.

In the revised manuscript, we clearly defined the training, testing, and external
validation cohorts, and provided a detailed description of the model's pretraining and
finetuning procedures, as well as the cross-validation strategy. Further details are
provided in our response to Q4.1.

**Revised in the manuscript:**

“The hyperparameters used for training SepsisFormer include a learning rate of
0.0010, a batch size of 5000, a dropout rate of 0.1000, 1400 training epochs, eight
parallel self-attention heads, and an eight integrated Transformer architecture
(Supplementary Table 15).” **(Page 5)**

Thank you once again for your valuable time and feedback. Your insightful
recommendations on specific issues were particularly appreciated. The manuscript has
been thoroughly revised in light of these suggestions, which has significantly
enhanced the quality of the work.

We hope that the revised manuscript now meets with your approval.

Wishing you all the best!

**Reviewer #3 (Remarks to the Author):**

**Q1:** *What are the noteworthy results?*

- *A prognosis model for sepsis that leverages Transformer models and domain*
*adaptation technique that achieves high predictive accuracy and a scorecard model*
*for risk stratification.*

- *The actual subphenotypes and risk levels classified by the findings from the model*
*experiments, and how these actually capture patient heterogeneity. Although, I could*
*appreciate this contribution later in the paper.*

- *An in depth analysis of and evidence for the heterogeneity of sepsis patients based*
*on subphenotypes and risk levels.*

**Response:** We sincerely appreciate the reviewer's positive comments on the
significance of our study and the recognition of our contributions. We have carefully
considered the reviewer's constructive suggestions for improvement and have taken
them into account during the revision process. To further aid understanding, we've
added Supplementary Figure 1.

**(New) Supplementary Figure 1. Integrated framework: An overview of models,**
 **methods, biomarkers, and clinical applications in our study.**

*Q2: Will the work be of significance to the field and related fields? How does it*
 *compare to the established literature? If the work is not original, please provide*
 *relevant references.*

*Q2.1 -The performance of existing AI prediction/prognosis models is mentioned in the*
 *introduction but does not include the techniques used in those models, which can help*
 *the reader to appreciate how the proposed models in this work contribute and differ*
 *from existing literature (in terms of input data, model choice, training paradigms,*
 *etc.).*

**Response:** Thank you for your valuable suggestion. We agree that providing details
about the techniques used in existing models would offer a better context for
understanding the contributions and differences of our proposed models. In the
revised manuscript, we expanded the introduction to briefly describe a total of 256
sepsis prediction models from 2016–2023, including machine learning models, deep
learning models, input data, and code.

**Revised in the manuscript:**

“A total of 256 AI-based sepsis prediction models from 73 studies (2016–2023,
n=457,932) showed a pooled AUC of 0.825 (95% CI: 0.809–0.840)¹⁸. Models mainly
include machine learning approaches (DT, LR, SVM, GLM, NB) and neural networks
(MLP, LSTM, CNN, GRU, RETAIN, Dipole). Public datasets (e.g., MIMIC-III/IV,
eICU, Computing in Cardiology) were used in 53% of studies. However, only 21.9%
performed external validation; data-sharing transparency was critically limited—only
three studies disclosed data access; and no studies released code¹⁸. Meanwhile,
although Transformer-based models (e.g., RETAIN, BEHRT, Med-BERT)¹⁹ have
achieved strong performance in EHR-driven disease risk prediction, their use in sepsis
remains relatively limited. On the other hand, in clinical practice, several
well-established prognostic warning score systems (SOFA¹, APACHE II, LODS²⁰,
qSOFA¹, SIRS¹) remain benchmarks. In an analysis of 148,907 EHRs of suspected
infection cases, the area under the receiver operating characteristic curve (AUC) for
patients admitted to the ICU ranged from 0.64 to 0.75 for existing scoring
systems¹⁴. In our previous study, we developed the LIP scoring system, which
incorporates lymphocyte count, INR, and procalcitonin as a simple sepsis screening
tool, achieving 92.8% sensitivity and 94.1% specificity. The LIP tool is well-suited for
rapid clinical screening and is particularly beneficial in resource-limited settings.”

**(Page 3-4)**

**Q2.2** - *In the domain adaptive generation, the method for oversampling is claimed to*
*be novel but also says that it uses existing domain adaptation methods. It is not clear*
*what the novelty of the method is. Besides, the domain adaptation is briefly mentioned*
*in the abstract and discussed again in the results under the Prognostic prediction*

*performance of SepsisFormer based on coagulation–inflammatory markers header.*
*The last paragraph of the introduction can include this component, as it seems*
*relevant considering the heterogeneity and class imbalance problems it addresses.*

**Response:** Thank you for your valuable suggestion.

We introduce MMID, a novel domain adaptive generation method. Unlike
established techniques such as mean-teacher, whitening, and moment matching,
MMID is a newly developed approach. We evaluated MMID's performance against
these methods and a non-adaption baseline (serving as an ablation study). The
comparative results are detailed in Figure 2d and Supplementary Table 8. We have
revised the original manuscript to clarify that MMID is a novel contribution, not an
existing method.

As MMID represents a cross-domain strategy employing SMOTE, we have revised
its nomenclature to MMID-SMOTE for improved specificity and recognition of its
underlying technique. In the revised manuscript, we rewrite the generator in the
prognostic model as follows:

**Revised in the manuscript:**

“Domain-adaptive generator for fine-tuning. To address the class imbalance and
distribution heterogeneity identified through our analysis of mortality outcomes and
multi-center covariate distribution shifts, we integrated a domain-adaptive generator
module into the prognostic modeling framework for fine-tuning. We specifically
implemented and compared several state-of-the-art domain adaptation methods,
including Mean–teacher⁵², Whitening⁵³, and Moment Matching⁵⁴, to reduce
inter-center distributional discrepancies. Crucially, we also propose MMID-SMOTE,
an innovative and clinically practical data augmentation strategy. This novel method
incorporates statistical moment alignment and min-max interval constraints.
MMID-SMOTE ensures that the synthetic samples generated are both statistically
robust and clinically pertinent, thereby substantially enhancing the model's
generalizability to unobserved target domains while maintaining clinical reliability
and applicability.” (Page 14)

**Reference:**

52. Tarvainen A, Valpola H. Mean teachers are better role models: Weight-averaged
consistency targets improve semi-supervised deep learning results. *Advances in neural
information processing systems* 30, 1195–1204 (2017).

53. Roy S, Siarohin A, Sangineto E, Bulò SR, Sebe N, Ricci E. Unsupervised
Domain Adaptation Using Feature-Whitening and Consensus Loss. In: *Proceedings of
the IEEE/CVF conference on computer vision and pattern recognition* (2019).

54. Peng X, Bai Q, Xia X, Huang Z, Saenko K, Wang B. Moment Matching for
Multi-Source Domain Adaptation. In: *Proceedings of the IEEE/CVF international
conference on computer visio*) (2019).

We've rewritten the Domain-adaptive Generator in **Supplementary Method 1** and
provided a more explicit rationale for this module. Additionally, visual analyses have
been incorporated to illustrate **class imbalance** and **heterogeneity across multiple
centers**.

**Revised in the Supplementary Method 1:**

**A. Domain-adaptive Generator for fine-tuning.**

**Rationale for Domain Adaptation in Prognostic Modeling.** In multi-center
Electronic Health Records (EHR) studies, class imbalance within single centers and
distribution heterogeneity across multiple clinical sites are prevalent, especially in
sepsis prognosis prediction tasks. Existing data augmentation methods, such as the
Synthetic Minority Over-sampling Technique (SMOTE), primarily address
intra-center class imbalance without considering inter-center distributional shifts. To
bridge this gap, we first theoretically justify the necessity of domain adaptation by
analyzing class imbalance in mortality outcomes and distribution divergence across
clinical centers. These domain discrepancies can degrade model performance when
transferring knowledge from one clinical center (source domain) to another (target
domain), making domain-adaptive strategies an essential component for robust
prognostic modeling.

The statistical chart of eICU, Local ICU, MIMIC III, and MIMIC IV is shown in
Reviewer-only Figure. All datasets exhibit significant class imbalance between

positive samples (non-survivors) and negative samples (survivors), with mortality
 rates of 25.38%, 37.23%, 25.56%, and 37.62% respectively. The number of
 non-survivors was significantly lower than that of survivors.

**Reviewer-only Figure.** Class imbalance in mortality of multiple clinical centers.

We conducted multi-center covariate distribution shift analyses to clearly demonstrate the significant distribution shifts in laboratory measurements observed across different medical centers. Kolmogorov-Smirnov (KS) non-parametric tests were utilized to assess the consistency of distributions between two medical centers. Results indicate significant distributional shifts ($p < 0.0001$) for the 7 laboratory measurements between every pair of centers across eICU, Local ICU, MIMIC III, and MIMIC IV (see Supplementary Figure for Method). The 7 coagulation-inflammatory laboratory measurements exhibit significant class imbalance within individual centers and notable distribution shifts across different centers. Meanwhile, these clinical markers also share common feature patterns among these centers, indicating their suitability for domain adaptation approaches to address domain discrepancy (Supplementary Table 2 and 3).

Considering the heterogeneity and commonalities of multi-center laboratory measurements, this paper introduces a method combining domain adaptation and SMOTE to alleviate class imbalance and distribution bias. This ultimately enhances model generalization to unobserved target domains. We assume that the sepsis predictors in source domain (S) and target domain (T) share a common feature space, but their data distributions differ $p(\mathcal{X}^S) \neq p(\mathcal{X}^T)$. Domain adaptation aims to reduce

this discrepancy by optimizing:

$$\min_{\theta} \mathcal{L}_S(\theta) + \lambda \times \mathcal{D}(p(\mathcal{X}^S), p(\mathcal{X}^T)) \quad (1)$$

where $\mathcal{L}_S(\theta)$ denotes the supervised loss on the source domain, and \mathcal{D}
measures the distribution discrepancy between the source and target domains, with the
balancing parameter λ .

**Implementation of Existing Domain Adaptation Methods.** This study
designates a combined dataset of 7,789 septic patients from MIMIC-III and
eICU-CRD, Cohort 1, as the source domain. The target domain included two external
datasets from previously unobserved centers: Cohort 2 with 4,191 septic patients from
MIMIC-IV and the smaller Cohort 3, with 428 septic patients from a local ICU. Each
patient's sepsis predictors are derived from the five distinct classification dimensions.
To address domain discrepancy in the mortality prediction model, we incorporated
several state-of-the-art domain adaptation techniques during model fine-tuning phase.

(1) Mean-Teaching. Employs a teacher-student architecture to enforce prediction
consistency under perturbations across domains. (2) Whitening Transformation. Aligns
the feature distributions by normalizing covariance structures between source and
target domains. (3) Moment Matching: Minimizes the statistical moment differences
(e.g., means, variances) between feature representations across domains. Thus, these
methods were applied to synthesize domain-adaptive training samples for the
fine-tuning phase of the pre-trained sepsis prognosis prediction model.

**A Novel Clinically Practical Domain Adaptation Method MMID-SMOTE.**
Despite the effectiveness of existing domain adaptation approaches, their high
computational cost, complex optimization, and resource requirements limit their
direct applicability in clinical environments. To overcome this, we propose a novel
and lightweight cross-domain data augmentation method, Maximum and Minimum
Interval Difference-based SMOTE (MMID-SMOTE). This method integrates
statistical moment analysis, domain adaptation, and clinical interpretability constraints
into the classic SMOTE framework, ensuring that the synthetic data generated are
both statistically valid and clinically meaningful.

**Step 1: Statistical Moment Calculation and Distribution Profiling**

For each prognostic predictors, compute its origin moments (mean

$\mu_X = \frac{1}{n} \sum_{i=1}^n X_i$, variance $\sigma_X^2 = \frac{1}{n} \sum_{i=1}^n (X_i - \mu_X)^2$) and central moments (skewness,

kurtosis) to fully capture those distributional characteristics across the source domain

S and target domain T .These moments reveal both location and dispersion,

providing the statistical foundation for cross-domain feature alignment to reduce

distributional shift.

**Step 2: Clinical Regularization to Address Heteroscedasticity**

Considering the biological variability in medical data (e.g., laboratory test results),

features exhibiting excessive variance (e.g., $\sigma^2 > 100$) are subject to a logarithmic

transformation to stabilize distribution and mitigate heteroscedasticity without

compromising the internal correlation structure:

$$X' = \log_{10}(X + 1) \quad (2)$$

In this study, this adjustment was necessary only for the White Blood Cell (WBC)

count, which showed extreme dispersion across centers—a known clinical variability

marker in sepsis patients.

**Step 3: Moment-Based Cross-Domain Projection and Alignment**

To reduce distributional divergence between domains, MMID-SMOTE employs

moment normalization and projection, aligning the target domain T data into the

statistical space of the source domain S :

$$X_T^{\text{aligned}} = \frac{X_T - \mu_T}{\sigma_T} \times \sigma_S + \mu_S \quad (3)$$

This operation ensures domain-invariant feature learning, prevents the generation

of outlier samples, and aligns the data distributions—a fundamental requirement in

domain adaptation theory for machine learning models applied to heterogeneous

clinical data.

Supplementary Figure for Method. Multi-center covariate distribution shift analyses.

Data pairs used for these analyses are A, eICU and MIMICIV. B, Local ICU and eICU.

C, Local ICU and MIMICIV. D, MIMICIII and eICU. E, MIMICIII and eICU. F, MIMIC

III and MIMIC IV.

**Step 4: Min-Max Interval Constraint Definition for SMOTE**

Given that the KS test relies on the maximum vertical distance of the empirical
cumulative distribution function (ECDF) $D = \max\|F_1(x) - F_2(x)\|$, and as
evidenced by Supplementary Tables 3 and 4, the data exhibit substantial range
discrepancies (e.g., maximum and minimum values). We define a strict min-max
interval constraint for each predictor to maintain clinical plausibility.

$$X_{\min} = \min(X_S, X_T), \quad X_{\max} = \max(X_S, X_T) \quad (4)$$

Synthetic instances must satisfy:

$$X_{\text{new}} \in [X_{\min}, X_{\max}] \quad (5)$$

This constraint respects both statistical boundaries and medical interpretability,
ensuring that no biologically implausible values are introduced into the dataset—a
key consideration in real-world clinical AI applications.

**Step 5: Customized SMOTE Sample Generation**

Finally, synthetic samples are generated via SMOTE under these constraints, with
the following parameters optimized for medical data context.

Number of nearest neighbors : 4. Minority class oversampling ratio: 1:1
(achieving balanced class distribution).Random seed (random_state): Fixed for
reproducibility. Parallel jobs: 3 (accelerating computation).Each new sample is
computed as:

$$X_{\text{new}} = X_i + \delta \times (X_j - X_i) \quad (6)$$

where X_i and X_j are minority class instances within the $k=4$ nearest neighbors

and $\delta \sim U(0,1)$. The synthetic point is only accepted if $X_{\text{new}} \in [X_{\min}, X_{\max}]$. This

ensures the synthetic data remain both statistically valid and clinically feasible.

Reviewer-only Figure. Sepsis laboratory measurements have commonalities across centers, making them suitable for domain adaptation of domain discrepancy.

We have rewritten Figure 2d and Supplementary Table 8 to more clearly demonstrate the performance advantages of MMID-SMOTE. The comparative baseline methods, such as Mean Teacher, Whitening, and Moment Matching, all represent the current state-of-the-art in the field. The results of the ablation experiments further confirmed the importance of domain adaptive generation.

Revised in the manuscript:

“As shown in Fig. 2d and Supplementary Table 7, MMID-SMOTE significantly outperformed other domain adaptation techniques and the ablation experiment (no-adaptation baseline) by substantially improving the model's cross-cohort generalization capability. AUC, accuracy, sensitivity, specificity, and F1-score were used to evaluate the model's predictive performance. The ablation study demonstrates that MMID-SMOTE achieves superior performance over the no-adaptation baseline across all five evaluation metrics, highlighting the importance of the domain-adaptive generation module in enhancing prediction accuracy. In terms of AUC, accuracy, sensitivity, and F1-score, MMID-SMOTE consistently outperformed three state-of-the-art domain adaptation methods. ACU results show Mean-teacher and Whitening underperformed a no-adaptation baseline, with Moment Matching ranking second. Mean-teacher struggles with low-quality pseudo-labels, while Whitening's mean/covariance adjustments failed to boost cross-domain performance. Moment Matching, in contrast, matches second-order statistics. This highlights the importance of domain adaptation method selection. Our proposed MMID-SMOTE achieved superior prognostic prediction across performance in different clinical cohorts. As an

XAI tool, SepsisFormer offers reliable, sensitive, and interpretable support for clinical
decision-making using a minimal set of routine markers.” (Page 7)

“Fig.2 d, ROC curves of prognostic prediction performance based on
coagulation–inflammatory markers from local ICU data, comparing MMID-SMOTE
with various domain-adaptive approaches and a no-adaptation baseline.” (Figure 2
Legends Page 25)

In the Introduction and Supplementary Method 1, we explicitly addressed class
imbalance and data heterogeneity.

**Revised in the manuscript:**

“ Although progress has been made, some limitations still exist: (1)...(2) Despite
demonstrating superior performance, AI models encounter two primary challenges in
clinical application: constrained predictive performance and generalization capability,
largely attributable to class imbalance and multi-center data heterogeneity;
concurrently, the "black-box" nature of model impedes clinicians comprehension and
trust in their decision-making.” (Page 4)

*Q2.3- The transcriptomic biomarker evidence analysis and study of anticoagulant*
*treatment effect seem crucial for better understanding sepsis, reveal insights about*
*heterogeneity, and validate algorithmic model development. However, I think that the*
*presentation of these approaches can be clarified because it wasn't clear what*
*experiments actually took place in this study or were using measurements from*
*existing public datasets (E.g., the Heparin vs control studies, or PCR studies), and*
*how the biomarkers were actually selected.*

**Response:** Thank you for the comments on the transcriptomic biomarker analysis and
anticoagulant treatment effect study. We appreciate the opportunity to clarify the
approaches presented in the manuscript. It should be noted that we integrated the
transcriptomic-level perspective solely for mechanistic analysis and did not
incorporate it into the development of tools such as SepsisFormer and SMART; these
tools rely exclusively on seven coagulation-inflammatory markers and age for
analysis. The following is a detailed response to your concerns:

**1. Transcriptomic Biomarker Analysis.**

Analysis	Data source	Type	Location in Manuscript
GSE65682	Prognostic gene discovery (28-day mortality)	Sepsis: 760 patients Healthy controls: 42	• Fig. 2h, Supp. Fig. 6,
GSE54514	External validation of prognostic biomarkers (STAT5B, MTHFR, HPSE, AAK1, MX1)	Septic non-survivors: 9 Septic survivors: 26 Healthy controls: 18	• Supp. Fig. 8 (labeled as "external cohort from GSE54514")
GSE26440	External validation of diagnostic biomarkers (CD59, SERPINB2, CFD, P2RX1)	Septic shock: 98 children Healthy controls: 32	• Supp. Fig. 9a,b
GSE95233	External validation of diagnostic biomarkers (CD59, SERPINB2, CFD, P2RX1)	Septic shock: 51 Healthy controls: 22	• Supp. Fig. 9c,d

The biomarker selection process involved two main steps. In Step 1, gene screening
 was conducted to identify diagnostic and prognostic biomarkers. For diagnostic
 biomarkers (CD59, SERPINB2, CFD, P2RX1), they were first identified from the
 differentially expressed genes (DEGs) between sepsis patients and controls in the
 GSE65682 dataset, then overlapped with coagulation- and inflammation-related genes
 (CRGs/IRGs) retrieved from KEGG and GeneCards databases, and finally validated
 in external GEO datasets (GSE26440 and GSE95233).

For prognostic biomarkers (STAT5B, MTHFR, HPSE, AAK1, MX1), they were
 derived from the DEGs between survivors and nonsurvivors in GSE65682
 (Supplementary Fig. 6a), overlapped with disseminated intravascular coagulation
 (DIC)-related genes (Supplementary Fig. 6c), and the final genes were selected
 through LASSO-Cox regression (Supplementary Fig. 6e). In Step 2, further gene
 screening was performed using reverse transcription-quantitative polymerase chain
 reaction (RT-qPCR) on a local cohort consisting of 29 sepsis patients and 11 controls,
 which confirmed the differential expression of CFD and P2RX1 (Fig. 2h, violin plots;
 Supplementary Table 9). Meanwhile, STAT5B, HPSE, AAK1, and MX1 were only
 computationally validated using GEO datasets.

**2. Anticoagulant Treatment Effect** (e.g., Heparin vs. Control)

**Retrospective Analysis of Observational Data:** The analysis of anticoagulant
effects (heparin) on sepsis outcomes was performed using retrospective clinical data
from the MIMIC-IV EHRs (n=4,191 patients: 946 heparin-treated vs. 3,245 controls).
Method: Patients were stratified into SMART risk levels (mild/ moderate/ severe/
dangerous) and CIS subphenotypes. The heparin cohort comprised patients receiving
heparin for ≥ 3 consecutive days (n=946), whereas the control cohort included those
not receiving heparin (n=3,245). The primary outcome, 28-day survival, was analyzed
using Kaplan-Meier and Cox models (Fig. 4a,c). However, this study did not involve
in vitro or in vivo experiments for anticoagulant treatment. The focus was on clinical
data-driven insights into treatment patterns and their association with biomarker
profiles.

*Q3: Does the work support the conclusions and claims, or is additional evidence*
*needed?*

*Q3.1 As I was reading the paper, I struggled to understand the support for some of the*
*claims in the results section since it wasn't clear the criteria used for doing further*
*classifications or analyses. For example, how does further categorizing the predictors*
*into clinically relevant ranges (not clear how these are defined) facilitate clinical*
*decision making? And then, why further analyzing the two subtypes across the five key*
*categories of sepsis predictors? This comment is also related to the clarity presenting*
*the methods.*

**Response:** Thank you for your valuable feedback. I understand the result and method
sections lacked clarity regarding the criteria for further classifications or analyses.

We appreciate the reviewer's insightful feedback. While our aim was to categorize
36 predictors into clinically relevant ranges (low, normal, high) based on sepsis expert
consensus for clinical reference, we acknowledge that this section did not effectively
enhance clinical decision-making or cohere with the overall manuscript. Therefore, we
have proceeded with the removal of this content from the main body and eliminated
Supplementary Table 2.

Further analysis of the two clusters across the five key categories of sepsis
 predictors aimed to pinpoint critical categories, specifically highlighting the
 coagulation-inflammatory category. We have systematically replaced all instances of
 'subtypes α and β ' with 'clusters α and β ' throughout the manuscript. Concurrently, this
 section and Fig. 2f have been rewritten. Here, we delineate the difference and
 relationships concerning clusters α/β and subphenotypes CIS1/CIS2 in Supplementary
 Figure: 'Research Framework for Cluster and Subphenotype Analyses'.

For more information on the differences between clusters α/β and subphenotypes
 CIS1/CIS2, please refer to our response to Q5.3.

 **(New) Supplementary Figure 2. Research framework for cluster and subphenotype.**

**Revised in the manuscript:**

"Clusters α and β were automatically derived using unsupervised clustering methods
 using 36 sepsis predictors, requiring no prior information (e.g., mortality, disease

outcomes, or treatment medications), based on the optimal number of clusters
(Supplementary Table 4). Five different unsupervised methods—Gaussian Mixture
Model (GMM), MiniBatchKMeans, K-means, Hierarchical Agglomerative Clustering
(HAC), and Birch—generated highly aligned clusters, a consistency across distinct
approaches that confirms the robustness and validity of the two identified clusters.
Since these data-driven clusters lack inherent clinical interpretation, we statistically
analyzed subgroup mortality. The GMM-derived α and β clusters exhibited the most
significant mortality difference (32.09% vs 17.62%, respectively), indicating GMM's
superior effectiveness for identifying patient subgroups with divergent mortality risks.
In the chord diagram, both cluster α and cluster β consistently showed the widest
chords with coagulation-inflammatory category (ribbons connect with these portions
of the circle), indicating coagulation-inflammatory predictors are key defining
features for the clustering. Independently, the radar plot also confirmed the
coagulation-inflammatory category as the highest connection point. Thses findings
were robustly replicated across all five unsupervised clustering methods (Fig. 2f and
Supplementary Fig. 3).” (Page 5)

“This study employs GMM for subphenotype identification, as the patient subgroups
derived from GMM show the most significant differences in mortality rates (shown in
Section Explainability Analysis), indicating greater clinical relevance.” (Page 7)

*Q3.2 From my understanding and interpretation, I agree with the paper that*
*explainable AI was used to uncover patient heterogeneity, I am not sure if the model*
*SepsisFormer was built to be explainable. Many parts use modules from Transformers,*
*a complex deep learning architecture, that as mentioned, can be beneficial for*
*modeling the complex relationships between the predictors. In fact, the explainability*
*technique used in the model directly is a post-hoc technique and this can be*
*acknowledge for clarity. Besides, it is mentioned that the model is*
*heterogeneity-aware because of the domain-adaptive generator, but this sounds like a*
*data augmentation strategy for training.*

**Response:** We sincerely thank the reviewer for this insightful comment.

We fully acknowledge that both the variable-level and model-level explainability
 analyses employed in our study are inherently post-hoc. To improve clarity, we have
 clarified this aspect in the revised manuscript. Furthermore, in response to **Q3.1** and
 **Q5.3**, we have consistently replaced 'subtypes α and β ' with 'clusters α and β '.

**Revised in the manuscript:**

“Multi-view explainability analyses were conducted from three perspectives: two
 post-hoc approaches (cluster-informed EHR variable-level and AI model-level), and a
 transcriptomic-level analysis inspired by multi-omics principles.” (Page 14-15)

We agree that the domain-adaptive generator is indeed a strategy for training, and
 we have explicitly clarified its role as such in the revised manuscript. However, it is
 fundamentally different from data augmentation methods. We have addressed this
 point in our response to **Q2.2**. Here, we provide further clarification. Both
 domain-adaptive generation and data augmentation (e.g., SMOTE) are used to
 improve model generalization, but they serve different goals and operate differently.
 We summarize their differences below.

	Data Augmentation	Domain-Adaptive Generation
Primary Goal	Improve generalization within a single domain	Improve performance across different domains
Domain Scope	Typically single domain or center	Typically multiple domains or centers
Data Modification	Creates variations of existing data	Transfers knowledge between domains
Domain Shift Focus	Addresses variance within the same distribution	Addresses variance between different distributions
Labeled Target Data	Not required (operates on source data)	Can range from none to some labeled data in target domain
Transfer Learning	Not a direct subset of Transfer Learning	A subfield of Transfer Learning
Common Techniques	Geometric/photometric transforms, noise injection	Domain distribution alignment, domain mapping, invariant feature learning, adversarial training

*Q3.3 Claims around improving patient outcomes in the introduction: this has to be*
*evaluated through the application in clinical decision-making and translational*
*research.*

**Response:** We agree this is crucial for the lack of clinical implementation evidence.
We replaced the illustration with “Our work, therefore, offers a novel set of simple,
real-time executable tools for sepsis heterogeneity, demonstrating considerable
potential to significantly enhance sepsis clinical practice globally, particularly in
resource-constrained healthcare settings.” in the **Abstract** section (**Page 2**)/ and
“Overall, this work, by combining XAI and coagulation-inflammatory markers,
deeply explores sepsis heterogeneity and develops high-performance, real-time tools
for clinical practice.” in the **Introduction** section. (**Page 4**)

A prospective, ethics committee-approved study has evaluated the clinical impact
of these tools by comparing pre-implementation and post-implementation outcomes,
including mortality rates and ICU length of stay (**Fig4 e-h**).

**Revised in the manuscript:**

“e-f, External validation of real-time SMART risk stratification and subphenotype
classification conducted using our open-access platform (<http://smartsepsis.org.cn>). e,
The real-time (day 1) risk stratification and distribution of the 40-patient cohort. The
number of patients in the four risk levels of mild, moderate, severe and dangerous
were 5 (CIS1, 3 and CIS2, 2), 12 (CIS1, 8 and CIS2, 4), 13 (CIS1, 9 and CIS2, 4) and
10 (CIS1, 7 and CIS2, 3). f, The mortality rates of different risk levels were 0%,
16.7%, 38.5% and 90% respectively. g, The mortality rates of CIS1 and CIS2 were
33.3% and 53.8%. h, The mortality rates of patients with different CIS classifications
at the four risk levels were as follows: at the mild level, both CIS1 and CIS2 had a
mortality rate of 0; at the moderate level, CIS1 had a mortality rate of 0 and CIS2 had
a rate of 50%; at the severe level, CIS1 had a rate of 33.3% and CIS2 had a rate of
50%; and at the risk level, CIS1 had a rate of 85.7% and CIS2 had a rate of 100%.”
(**Figure legends, page 29**)

*Q3.4 What is meant by clinically relevant in this statement: “XAI method achieved*
*clinically relevant sensitivity and accuracy in prognostic prediction with only eight*

*routine markers.*”

**Response:** Thanks for the clarification. In this context, "clinically relevant" refers to
performance thresholds that would meaningfully alter clinical decision-making,
primarily encompassing AUC, specificity, and sensitivity.

**Revised in the manuscript:**

“Using only seven coagulation-inflammation biomarkers and age, SepsisFormer
achieved high prognostic performance, with AUCs of 0.8558 in internal testing (in
Cohort 1, specificity: 0.7398; sensitivity: 0.9264) and 0.8596/0.8364 in external
validations (Cohorts 2/3). DeLong’s test confirmed its superiority over all baseline
models in Cohort 1 (all $p < 0.01$) and most baseline models in the external cohorts,
demonstrating the model’s robustness and generalizability (Fig. 2e; Supplementary
Tables 5,6).” (Page 7)

**Q4:** *Are there any flaws in the data analysis, interpretation and conclusions? - Do*
*these prohibit publication or require revision?*

**Q4.1** *Even though the components of SepsisFormer along with their functionality are*
*explained, the process for making these algorithmic choices is not clear. Was there*
*any ablation study to select the parameters or assess the contribution of some parts*
*(e.g., the domain adaptation)? This will provide stronger evidence for the models*
*proposed in this paper.*

**Response:** We sincerely thank the reviewer for this insightful comment. We agree that
a more detailed explanation of our algorithmic choices and the empirical evidence
supporting them would significantly strengthen the paper. In the revised manuscript,
we elaborate on the following points:

**Contribution of Domain Adaptation Module (Ablation Studies).** To provide
stronger evidence for the proposed model, we quantify the impact of the domain
adaptation module on model performance. This demonstrates its necessity and
effectiveness in mitigating domain shift and improving generalization. Please refer to
our response to **Q2.2** for detailed revisions.

**Transformer Encoder.** We included supplementary experiments, examining the

effects of varying the number of Transformer layers and self-attention heads. The
corresponding changes in performance illustrate the rationale behind these choices.
Additionally, we have added justifications for choosing the Transformer architecture
in the *Introduction* section. Please refer to our response to **Q5.2**.

To determine the optimal architecture for our deep learning-based prognostic
model, we conducted a systematic hyperparameter sensitivity analysis on its core
components: the number of Transformer layers (M) and the number of self-attention
heads (N) within each layer. The experiment was performed on the training set,
cohort1. First, we evaluated the impact of varying the number of layers on model
performance. The results indicated that the model achieved excellent and stable
performance when M was between 1 and 7, with a slight decline observed at $M=8$.
Considering that a deeper model has greater potential for feature extraction and to
strike a balance between model complexity and generalization, we selected an 8-layer
architecture. Subsequently, with M fixed at 8, we further investigated the effect of the
number of attention heads (N) ranging from 1 to 8. A similar trend was observed,
where performance was superior for N between 1 and 7, followed by a drop at $N=8$.
The corresponding performance curves are illustrated in the new Supplementary
Figure.

**Revised in the manuscript:**

“To optimize the model architecture, we conducted a systematic hyperparameter
sensitivity analysis on the number of Transformer layers L and attention heads. We
chose $L=8$ to balance complexity and feature extraction, despite peak performance
at $L=1\sim 7$. With M fixed at 8, we observed a similar pattern for attention heads
(H), with optimal performance at $H=1\sim 7$ before a decline at $H=8$
(Supplementary Fig. 5). Furthermore, the consistent ranking of predictor importance
underscores the model's structural robustness and deterministic nature. These findings
collectively affirm the model's reliability and interpretability in multivariate
prediction tasks.” **(Page 6)**

**(New) Supplementary Figure 5.** Hyperparameter sensitivity analysis of the
 number of Transformer layers and attention heads on model prognostic performance.

**Statistical Comparison of Performance.** We supplemented our analysis by
 performing DeLong’s test to statistically compare the AUCs of different models,
 confirming that SepsisFormer significantly outperforms the eight deep learning and
 machine learning methods. We implemented the test using Delong_test from the
 MLstatkit.stats package, in conjunction with roc_auc_score from scikit-learn. The
 revised manuscript includes detailed descriptions of the statistical methods and results
 in the **Statistical Analysis** of the **Methods** section, along with legends for **Figure 2**.

**Revised in the manuscript:**

“DeLong’s tests were performed to statistically compare the AUCs of different
 models using Delong_test from the MLstatkit.stats package, in conjunction with
 roc_auc_score from scikit-learn.” **(Page 16)**

“d, ROC curves of prognostic prediction performance based on

coagulation-inflammatory markers from local ICU data, comparing MMID-SMOTE
with various domain-adaptive approaches and a no-adaptation baseline. e, The AUC
for prognostic prediction demonstrates the superior performance of SepsisFormer
with coagulation-inflammatory markers over the baseline models in Cohort 1 (all
$p < 0.01$), Cohort 2 ($p < 0.01$, except SepsisFormer vs. LSTM, $p = 0.91$), and Cohort 3
($p < 0.05$, except SepsisFormer vs. GPT, $p = 0.16$, and SepsisFormer vs. RF, $p = 0.60$).
All p-values by DeLong's test. Cohort 1 with 7789 septic patients from MIMIC-III
and eICU-CRD, cohorts 2 with 4,191 sepsis patients from MIMIC-IV and Cohort 3
with 428 patients from a local ICU.” (Legends for Figure 2, Page 24)

**Hyperparameters.** We have added details regarding the hyperparameters used
for training our model SepsisFormer. Specifically, the learning rate is set to 0.0010,
the batch size is 5000, the dropout rate is 0.1000, and 1400 training epochs.

**Q4.2** *The discussion section currently presents medical literature and definitions in a*
*list-like format. While this demonstrates a strong understanding of sepsis from a*
*clinical perspective (which is valuable for algorithm development and interpretation*
*of the results), the manuscript would benefit from a more analytical approach. I*
*recommend expanding the discussion to include comparative analyses of the different*
*models for the prognosis and risk stratification task, the different domain adaptation*
*strategies used, or any other pertinent comparisons.*

**Response:** Thank you very much for your constructive suggestions. Just as we
mentioned at the very beginning, we utilized two new framework diagrams
(Supplementary Figures 1-2) to help us understand our research work and content
more quickly. Regarding the discussion section, we did indeed lack comparisons of
existing subtype classifications or models before. Based on the opinions of other
reviewers, we have already been discussing the relevant comparisons of some
certificates.

**Revised in the manuscript:**

“One of the important studies that directly applied transcriptomic data to the

subtyping of sepsis came from the MARS consortium in 2017⁴⁴. They enrolled a total
of 787 cases of sepsis patients in a discovery cohort and two validation cohorts.
Through machine learning and analysis of differentially expressed genes, sepsis was
classified into four subtypes, MARS1-4. Among them, MARS1 had the poorest
prognosis with a mortality rate as high as 35%. This classification at the level of gene
expression is also named endotype. In 2019, Seymour CW *et al.*⁷ conducted
phenotypic analysis on 20,189 patients with sepsis using statistics, machine learning
and simulation tools. 29 variables, including cardiovascular, hematopoietic, hepatic,
coagulation-inflammatory, neurological, pulmonary, renal, and so on, were used.
Finally, they divided the patients into four phenotypes: α , β , γ , and δ . Among them,
the mortality rate of phenotype α was the lowest at approximately 5%, while the
mortality rates of type β , γ and δ were 13%, 24% and 40% respectively. From the
perspective of mortality rates, our risk stratification results are similar to those of the
Seymour CW's phenotypic classification. However, we revealed the heterogeneity of
sepsis from two aspects: the stratification of risks and the subtypes. Moreover, the
clinical indicators we use are fewer and more beneficial for clinical practice, which
not only reduces the consumption of patients' blood samples but also saves costs, and
it is also conducive to real-time dynamic assessment. Furthermore, the Seymour CW
team has not yet developed a scoring system or a shared platform that can be
universally implemented by clinicians. Most importantly, although many sepsis
subtypes have been published at present, due to the different goals, the use of different
clinical indicators, different machine learning models, or black-box algorithms, there
are varying degrees of differences among these subtypes, ultimately resulting in low
comparability or overlap among them (like MARS1-4, SRS1-2, Hyper or hypo
inflammatory, and SENECA subtypes)⁸, and no shared classification tools or
platforms have been simultaneously proposed. However, it is still necessary to further
conduct comparative studies on our classification and the existing subtype
classifications.” (Page 11)

Reference as following:

44. Classification of patients with sepsis according to blood genomic endotype: a

prospective cohort study. Lancet Respir Med. 2017 Oct;5(10):816-826.

7. Derivation, Validation, and Potential Treatment Implications of Novel Clinical
Phenotypes for Sepsis. JAMA. 2019 May 28;321(20):2003-2017.

8. Uncovering heterogeneity in sepsis: a comparative analysis of subphenotypes.
Intensive Care Med (2023) 49:1360–1369.

***Q4.3** Considering the multiple insights found in this study, the paper is lacking an*
*overall conclusion about the approach followed. This can also be beneficial to*
*reinforce the main message and contribution of the paper.*

**Response:** Thank you very much for your constructive comments and
recommendations. Yes, we do need to present the conclusion, which we have
supplemented in the main text. Please refer to the end of the discussion section in the
manuscript.

**Revised in the manuscript:**

“In summary, this study introduces the prognostic model SepsisFormer, the automated
risk stratification tool SMART, and the subphenotypes CIS1/CIS2 to characterize
sepsis heterogeneity. Through multi-level explainability analyses, age and seven
routine coagulation-inflammatory markers were identified as key predictors for sepsis
diagnosis and prognosis. SepsisFormer achieved an AUC of 0.9301, outperforming
state-of-the-art models, while SMART reached an AUC of 0.7360, exceeding
conventional clinical scores and effectively stratifying mortality risk. Notably, CIS2
patients showed higher mortality and distinct coagulation-inflammatory profiles
compared to CIS1. Integrating subphenotyping with risk stratification uncovered
heterogeneity in anticoagulation treatment effects, supporting more precise and safer
therapeutic decision-making in sepsis management. Patients with moderate/severe
levels or CIS2 get more substantial benefits from anticoagulant treatment. An
open-access, web-based platform (<http://smartsepsis.org.cn>) facilitates real-time risk
stratification and subphenotype identification using only seven low-cost, routinely
available coagulation-inflammatory biomarkers. Its simplicity, accessibility, and
practical applicability make it a promising tool for improving sepsis management

worldwide, particularly in resource-constrained healthcare settings. (Page 12)
Reference: Global, regional, and national sepsis incidence and mortality, 1990-2017:
analysis for the Global Burden of Disease Study. Lancet. 2020 Jan 18;395(10219):
200-211.

*Q5: Is the methodology sound? Does the work meet the expected standards in your*
*field?*

*In general, the selected methods seem appropriate for the problems to be addressed in*
*the paper. However, I found the following concerns:*

*Q5.1 In the introduction it says that the methods use only routine laboratory blood*
*measurements and how this provides an alternative to blood transcriptomic-based*
*methods. Therefore, I found confusing when later in the paper an analysis of*
*transcriptomic biomarkers is introduced (which involved additional methodological*
*steps). Please clarify if I misunderstood this part.*

**Response:** Thank you for your valuable feedback. You have astutely identified a
point of potential confusion in our manuscript, and we sincerely apologize for the lack
of clarity in our initial manuscript.

As stated in our introduction, the core objective of this research is to develop a
sepsis prognostic tool that relies **exclusively on routine, low-cost laboratory blood**
**measurements** to ensure its accessibility and utility in a real-world clinical setting.
We wish to formally clarify that our prognostic model development, validation, and
proposed clinical application workflow **do not, at any stage, involve the use of**
**costly and time-intensive transcriptomic data.**

The transcriptomic analysis you refer to was designed as a separate, *post-hoc*
**mechanistic validation module**. It is not part of the predictive pipeline. Instead, its
purpose was to provide multi-omic evidence to explore and validate the **biological**
**plausibility** of the coagulation-inflammatory markers selected by our model, thereby
strengthening the mechanistic underpinnings of our findings.

To present this logic unequivocally, our research path can be summarized as
follows:

- 1. **Prediction Model Development:** We leveraged machine learning to construct
a sepsis prognostic model based on 35 routine blood measurements.
- 2. **Key Feature Selection:** Through model explainability analysis, we identified
seven critical coagulation-inflammatory markers that were pivotal for
prediction, enabling the creation of a more parsimonious and effective model
for sub-phenotyping and risk stratification.
- 3. **Biological Validation:** Subsequently, we employed an independent
transcriptomic analysis to elucidate, from a mechanistic standpoint, why the
coagulation-inflammatory pathways represented by these seven routine
markers hold such a central role and high prognostic utility in the
pathophysiology of sepsis.

To thoroughly resolve the ambiguity in the revised manuscript, we have
undertaken four systematic modifications:

- 1. **Addition of Supplementary Figure 1:** A new schematic overview of the
study design visually delineates "Prognostic Model SepsisFormer" and
"Explainability Analyses" as two distinct yet complementary modules,
explicitly positioning the transcriptomic analysis within the latter.
- 2. **Addition of Supplementary Figure 2:** This figure details the
feature-selection funnel from 35 initial predictors to the seven key
coagulation-inflammatory markers and illustrates how transcriptomic analysis
provides robust corroborating evidence for their biological significance.
- 3. **Reinforcement of Transcriptomic Validation Rigor (Supplementary**
**Figures 8, 9):** To further bolster the robustness of our explanatory findings, we
have incorporated external dataset validation for the key transcriptomic
biomarkers, confirming their stability and performance.
- 4. **Systematic Revision of the Manuscript Narrative:** We have systematically
rewritten relevant descriptions in the Introduction and Results sections to
consistently emphasize the explanatory role of the transcriptomic analysis and
to affirm that our final proposed model is based entirely on routine laboratory
markers, underscoring its clinical actionability.

We are confident that these comprehensive revisions fully address the issue you
 raised. Thank you once again for your incisive comment, which has been instrumental
 in enhancing the rigor and clarity of our work.

**Q5.2** *The analyses presented in the paper are extensive, but the rationale for each*
 *type is missing. For example, why does it matter to see the contribution of the five*
 *categories of predictors? And their associations? Why is it important to analyze the*
 *relationships between the 1-N transformer heads and lab measurements? Including*
 *these justifications can also clarify the choice for the visualization and/or analysis*
 *technique.*

**Response:** Thank you for your feedback. It is crucial for improving the paper's
 completeness and readability. In the revised manuscript, we will ensure that the
 rationale and clinical/research justifications for these analytical methods are clearly,
 concisely, and adequately explained and supported by sufficient evidence.

Sepsis, a complex and heterogeneous syndrome, is evaluated in this study using
 36 predictors across five categories, which align with the latest Sepsis-3.0 definition
 (excluding the central nervous system) and comprehensively reflect the disease state.
 However, the practical clinical utility of all 36 predictors is often limited by several
 significant factors. Therefore, our further analysis involving clustering across these
 five key categories serves critical clinical research objectives.

Reviewer-only Figure. Categories in Sepsis Definition

We have thoroughly revised this issue as follows.

(1) We have elaborated on the research aims of explainability analyses.

**Revised in the manuscript:**

“The 35 laboratory measurements collectively capture multi-organ dysfunction
spanning five categories: coagulation-inflammatory, hepatic, renal, blood gas, and
oxygen transport (Erythrocyte), while largely conforming to the connotation of
Sepsis-3 criteria (excluding neurological markers). However, their full clinical
adoption is hindered by prohibitive costs, large blood volume requirements, and
implementation barriers in resource-constrained settings. To overcome these
challenges, explainability analyses were performed across five categories to achieve
two clinical goals: (a) Identify a mechanistically relevant, and clinically feasible
subset of laboratory measurements within one category for real-time and
cost-effective application; (b) Select an efficient unsupervised clustering method to
uncover clinically meaningful sepsis subphenotypes, essential for understanding
disease heterogeneity and guiding clinical insights.” (Page 5)

(2) We've elaborated the cluster-informed EHR variable-level explainability.

**Revised in the manuscript:**

“Multi-view explainability analyses were conducted from three perspectives: two
post-hoc approaches (cluster-informed EHR variable-level and AI model-level) and a
transcriptomic-level analysis inspired by multi-omics principles. To select key
categories that effectively mirror sepsis pathophysiology and are practical for clinical
application, we employed unsupervised clustering methods, including Gaussian
Mixture Model (GMM), MiniBatchKMeans, K-means, Hierarchical Agglomerative
Clustering (HAC), and Birch. Chord diagrams and radar plots enhance the
interpretability of unsupervised clustering results by revealing the variable patterns
defining each cluster. Clinical expertise and Pearson's correlation matrix were
employed to analyze the associations between individual laboratory measurements
within each category.” (Page 14-15)

(3) The rationale for selecting the seven coagulation-inflammatory biomarkers and
age is detailed in the **new Supplementary Figure 2: 'Research Framework for Cluster**
**and Subphenotype Analyses.'** Further details are provided in our responses to **Q3.1** ,
**Q5.2** and **Q5.3**. We have added a comprehensive summary regarding the selection of
laboratory measurements. Further details are provided in our responses to **Q5.4**.

*Why is it important to analyze the relationships between the I-N transformer heads*
*and lab measurements?*

**Response:** In fields such as medicine and bioinformatics, the Sankey diagram is a
widely used visualization tool. The Sankey diagram in Figure 2g illustrates the
cumulative contribution and ordering of eight predictors across different Transformer
depths, with $N \in [1,8]$. Here, N denotes the number of cascaded Transformer layers,
rather than the number of self-attention heads, which is represented by M . In this
study, both N and M are 8.

Two key observations can be drawn from the Sankey diagram:

(1) As the number of Transformer layers increases, the overall contribution of the
eight predictors shows a slight upward trend, indicating enhanced feature integration
capacity with deeper architectures. However, the overall variation remains modest,
suggesting structural stability;

(2) For any fixed N , the contribution ranking of the predictors remains highly
consistent, reflecting strong repeatability and robustness in the model's feature
selection process. This implies that model performance is governed by deterministic
structural mechanisms rather than stochastic fluctuations.

The Sankey diagram provides visual and empirical support for the model's stability
and consistency in feature utilization as network depth increases, reinforcing its
reliability and interpretability in medical or multivariate predictive tasks.

**Revised Manuscript:**

“The Sankey diagram shows that as Transformer depth increases, the cumulative
contributions of the eight predictors slightly increase yet remain generally stable,
indicating consistent feature integration across layers.” (**Page 6**)

**Q5.3** *Not clear how the subtypes alpha and beta in the variable-level explainability*
*analysis differ from the subphenotypes if they are both identified based on distinctions*
*in mortality rates.*

**Response:** Thank you for your valuable feedback. It is crucial to clarify the two
conceptual groupings presented in our study.

In the revised manuscript, we further emphasize that the "clusters α and β " and
"subphenotypes CIS1 and CIS2" aren't defined by differences in mortality rates.
Instead, they were automatically derived using unsupervised clustering methods that
require no prior knowledge. Specifically, the observed difference in mortality between
"clusters α and β ," "CIS1 and CIS2" emerged after unsupervised clustering and was
found to be consistent with clinical observations, thereby supporting the clinical
relevance of these clusters. Differential mortality analysis of clusters α/β guided
subphenotype clustering methodology (GMM is selected).

We acknowledge that the term "subtypes" may cause confusion. However, we
acknowledge that the term "subtypes" may cause confusion, particularly given the
established use of "subtypes $\alpha, \beta, \gamma, \delta$ " derived from EHR data. To improve clarity,
we have replaced all instances of 'subtypes α and β ' with 'clusters α and β ' throughout
the manuscript. According to the widely accepted definition, a subphenotype refers to
"A distinct subgroup within a phenotype that can be reliably discriminated from other
subgroups based on a set or pattern of observable or measurable properties." In this
context, categorizing CIS1 and CIS2 as subphenotypes is appropriate and justified.

For further explanation and revision regarding clusters α and β as well as
subphenotypes CIS1 and CIS2, please refer to our response to Q3.1.

Reference

1. Antcliffe, DB, Burrell A, Boyle AJ et al (2025). Sepsis subphenotypes, theragnostics
and personalized sepsis care. *Intensive Care Medicine*, 1-13.

2. Gordon AC, Alipanah-Lechner N, Bos LD et al (2024) From ICU syndromes to ICU
subphenotypes: consensus report and recommendations for developing precision
medicine in ICU. *American Journal of Respiratory and Critical Care Medicine*
2:155–166.

**Q5.4** Throughout the paper, it is stated that eight markers were identified (7
 coagulation-inflammatory measures + age) and used for prognosis modeling, risk
 stratification, and further identifying subphenotypes, but the rationale for selecting
 this subset of predictors is unclear. The use of routine coagulation-inflammatory are
 more than seven measures.

**Response:** The rationale for selecting the seven coagulation-inflammatory biomarkers
 and age is detailed in the new Supplementary Figure: '**Research Framework for**
 **Cluster and Subphenotype Analyses.**' We have rewritten the entire **Explainability**
 **Analysis of Coagulation–Inflammatory Dysfunction** section (Pages 5-7). Further
 details are provided in our responses to Q3.1, Q5.2, and Q5.3.

**(New) Supplementary Figure 1.** Integrated framework: An overview of models,
 methods, biomarkers, and clinical applications in our study.

**Q5.5** *For the evaluation of the risk stratification model, it could be beneficial to*
*clarity what ground truth is used for the evaluation of this task.*

**Response:** Thank you very much for your constructive suggestions. The comment
you mentioned is precisely the problem we are researching and aiming to solve. Our
goal is to integrate multi-omics information through advanced AI algorithm tools and
provide a tool or platform with practical clinical significance that can be operated in
real-time by clinicians, thereby improving the prognosis of patients with sepsis.

We have made very detailed revisions in the interpretability section of the
manuscript. Please refer to pages 5 to 9 of the revised manuscript. The efficacy of
SMART risk stratification was verified to be very close and robust in MIMIC-III,
MIMIC-IV, eICU, and local ICU. (Fig. 3e)

Moreover, the efficacy of SMART risk stratification is very close and robust in
the verification results of MIMIC-III, MIMIC-IV, eICU, and local ICU. Meanwhile,
its prognostic predictive efficacy is due to the currently widely used SOFA, qSOFA, A,
LIP, APACHE II, and SIRS. The AUCs of them are 0.7360 (SMART), 0.6833, 0.6441,
0.6431, 0.6222 and 0.5428, respectively. The SMART scoring system achieved the
highest predictive value.

Lastly, combining the comments of other reviewers, we promptly initiated a
prospective external real-time validation study of the cohort, with the aim of assisting
clinicians in presenting a practical risk stratification and subphenotypic classification
system. We have supplemented it in both the methodology section and the results
section.

Due to our open-access web-based sepsis subphenotype and SMART platform
(<http://smartsepsis.org.cn>), which is better suited for real-time applications. Here, we
have added a prospective, real-time, external cohort validation study. We monitored
40 patients in real-time (the first day of being enrolled in this cohort), and the results
were consistent with the model's predictions, as follows:

**Revised in the manuscript:**

“Furthermore, for the real-time and external validation of the risk stratification and
subphenotypic classification, with the approval of the Ethics Committee of

Chongqing University Affiliated Central Hospital (Chongqing Emergency Medical
Center) (ID: 2025-55), a prospective observational study on sepsis risk stratification
and subphenotypic classification via our risk stratification and subphenotype platform
was conducted. The endpoint event was to observe the actual 28-day mortality rate
and obtain the SMART score and subphenotype at the time of enrollment. After
obtaining written informed consent, a total of 40 patients with sepsis were enrolled
from March 24, 2025, to April 28, 2025.” (Page 13-14)

“Last but not least, clinicians could conduct real-time risk stratification and
subphenotypic classification of sepsis based on the SMART scorecard or our
open-access sepsis subphenotype and SMART platform (<http://smartsepsis.org.cn>).
Here, we conducted risk stratification, phenotypic classification and prognosis
prediction for 40 sepsis patients locally admitted from March 21, 2025, to April 27,
2025 (external observational verification only) (Fig. 4e-h, and Supplementary case
materials 1 and 2). The results showed that the proportions of patients in the four risk
levels of mild, moderate, severe, and dangerous were 12.5%, 30%, 32.5% and 25%
respectively (Fig. 4e), and their 28-day actual mortality rates were 0%, 16.7%, 38.5%
and 90% respectively (Fig. 4f). The overall mortality rate of the CIS1 subphenotype
was 33.3%, significantly lower than 53.8% of CIS2, and the trend was the same at
different risk levels (Fig. 4g and 4h). The mortality rate at the dangerous level (this
external observational cohort) was a little higher than the predicted rate of the model
in this study (approximately 50%), as well as simultaneously increased the overall
mortality rates of CIS1 and CIS2, which might be related to the small sample size, but
the overall trend was basically consistent. Therefore, clinicians can utilize the
SMART scorecard or our open-access platform to conduct real-time risk classification
and subphenotyping of patients, enabling objective and accurate assessment of sepsis
patients to intervene as early as possible and improve their prognosis.” (Page 9)

Therefore, multiple data show that SMART risk stratification is robust, clinically
significant, and has real-time clinical operability, which is not possessed by the vast
majority of current risk stratification and typing tools.

Additional subgraphs in Revised Figure 4.

**Q6:** *Is there enough detail provided in the methods for the work to be reproduced?*

*Considering the details and codes provided in the supplementary materials, probably*
*yes. However, I found additional experiments, like fine-tuning or pertaining the*
*prognosis models, which was never mentioned in the main manuscript.*

**Response:** We are very grateful for the reviewer's feedback, which acutely identified
a significant omission in the main body of our manuscript. Although Figure 1 clearly
illustrates the complete framework of SepsisFormer model, depicting both its
pre-training (pink lines) and fine-tuning (green lines) processes, the corresponding
textual description was indeed missing. We have now rectified this issue by providing
a comprehensive elaboration of the model's architecture within the main text.

1. We have added descriptions of both the pre-training and fine-tuning procedures.

**Revised in the manuscript:**

“(e) Pretraining Procedure. Cohort 1 (MIMIC-III and eICU-CRD) was partitioned 7:3
for training and internal testing. From this training data, the pre-trained SepsisFormer
model was initialized, establishing its core architecture, weights, and hyperparameters.
SMOTE was applied for data augmentation to address class imbalance and enhance
model robustness during this phase. (f) Fine-tuning Procedure. Subsequently, for
domain adaptation and enhanced generalizability, SepsisFormer underwent
fine-tuning. Cohort 1 served as the source domain, with fine-tuning specifically
targeting external validation cohorts: Cohort 2 (MIMIC-IV) and Cohort 3 (Local ICU)

as the target domains. A Domain-adaptive Generator was employed for data
augmentation during this transfer learning stage, optimizing performance in the new
clinical center. The details of SepsisFormer can be found in Supplementary Method 1.”
**(Page 14)**

2. We've rewritten the Domain-adaptive Generator in **Supplementary Method 1** and
provided a more explicit rationale for this module. Additionally, visual analyses have
been incorporated to illustrate **class imbalance** and **multi-center data heterogeneity**.
Further details are provided in our responses to **Q2.2**.

3. We have revised the title in Supplementary Tables 6 and 7 to more clearly indicate
pre-training and fine-tuning.

**Revised in the Supplementary Info:**

“**Supplementary Table 6.** Performance comparison of the pre-trained
SepsisFormer and baseline models for prognostic prediction on the internal test set
(Cohort 1: 7,789 septic patients from MIMIC-III and eICU-CRD; 70% training, 30%
test).”

“**Supplementary Table 7.** Performance comparison of the fine-tuned
SepsisFormer and baseline models for prognostic prediction on the two external
validation cohorts (Cohort 2: 4,191 septic patients from MIMIC-IV; and Cohort 3:428
septic patients from a local ICU), using 10-fold cross-validation.”.

Reviewer #3 (Remarks on code availability):

*Code was not available in the submission. Some pseudo codes were presented in the*
*supplementary materials.*

**Response:** We sincerely apologize for the inconvenience and difficulty you
encountered while attempting to review the analysis in detail despite having
downloaded the datasets. We have now included the relevant codes, along with
detailed instructions and videos on how to run them. To promote transparency,

reproducibility, and clinical applicability, we have made the following resources
publicly available:

**1. Interactive Web-based Platform** (<http://smartsepsis.org.cn>): We developed an
open-access, user-friendly online platform for sepsis subphenotyping and risk
stratification. This tool allows end-users, such as clinicians and researchers, to
perform real-time subphenotype identification and risk level prediction by uploading
their own data without requiring additional coding or technical expertise.

**2. Source Code and Datasets** (<https://github.com/zhuli19031218/SepsisFormer>):
All raw data, preprocessed datasets, and fully documented source code are available
in our GitHub repository. These resources enable independent verification, replication,
and extension of our study.

**3. Code Execution Tutorial** (<https://doi.org/10.5281/zenodo.15634369>): A detailed
step-by-step video tutorial is provided to guide users through the complete process of
reproducing the main results presented in this study, ensuring ease of use even for
non-technical users.

These provided resources allow for replication and application of our method. We
believe these improve the paper's quality and transparency.

We're especially grateful for your insightful comments regarding the resolution of
methodological concerns, the clarification of transcriptomic data interpretation, and
the distinction between subtypes and subphenotypes. These revisions have
significantly strengthened the scientific rigor and overall presentation of our study.

We believe these comprehensive revisions will meet with your approval.

Hope everything goes great for you!

Explainable AI-driven heterogeneity using coagulation–inflammatory markers improves prognosis prediction, risk stratification, and anticoagulant treatment effects for sepsis

Corresponding Author: Niu Bailin

This file contains all reviewer reports followed by all author rebuttals in order.

Version 2:

Reviewer comments:

Reviewer #1 (Remarks to the Author):

In my previous review I have raised the following points:

1. The clarity and framing of the main message/ contribution of the paper
2. Issues relating to the reporting of the methodology and sample
3. Greater transparency towards the methods employed (e.g. datasets and codes)

Glad to share that these three issues were addressed in this latest revision.

The results of these paper are noteworthy as indicated in my earlier review and the ability to translate the algorithm developed into a clinical profiling score has field implications.

Good effort of showcasing the execution of the codes in the videos.

Reviewer #2 (Remarks to the Author):

The authors have adequately addressed the concerns, and the manuscript is now suitable for publication.

Reviewer #3 (Remarks to the Author):

The main concerns were addressed in the revision, including the performance comparisons to existing models, including both of the main model and the domain-adaptation strategy. Implementation details and definitions were also provided for the latter method, which helped to understand how it differs from existing techniques. However, in the description of the domain

adaptation in the supplementary method, the justification that analyzed class imbalance in mortality
outcomes is not theoretical but rather empirical I think, since you based the analysis on your data
samples. Further methodological details were provided for the transcriptomic analysis and study of
HTEs effect. Ablation studies are also presented and clarifications around techniques used and
claims made throughout the manuscript.

Besides, the authors extended the interpretation of the explainability analysis and how this
information is useful to understand the heterogeneity of the disease and relevant biomarkers for
sepsis prognosis. They also commented on the implications of having a model that predicts risk-
stratified levels and the corresponding analyses.

In the results, there is connection between findings and medical relevance (e.g., found importance
of coagulation-inflammatory indicators vs functional indicators of specific organs and how the
former better reflects the pathophysiology of sepsis), and an application use case of the tool available
online was demonstrated.

The paper still needs some organization edits, especially in the presentation of the results:

- Some analyses are listed but then presented later. E.g., 1) the correlation between predictors
mentioned in line 145 and then presented in line 176, 2) finding a subset of laboratory measurements
within one category (part (a) in line 155) but the clustering results of part (b) are presented first,
followed by the correlation analysis.

- Improve the introduction of the transcriptomic-level explanations and transcriptome analysis: It is
clarified that this was to show the important roles of coagulation-inflammation related indicators,
but there are some aspects that can be clarified. where was the gene information gathered from?
Briefly mention the methods involved in the paragraph in line 207.

- It is still unclear to me how the SEVEN predictors were selected. In the model-level explainability
only 3 are mentioned (APTT, WBC, and age) and the importance of coagulation-inflammation
predictors, but there at least 10 from what I can see in Figure 2c and 8 in Figure 2g. The overall
selection process was illustrated in a figure in the response letter, but I may be missing some details
to fully understand the criteria for picking those 7 variables. I also noticed that the functions of these
7 coagulation-inflammation markers is discussed in detail in the paragraph starting in line 362 and
they are part of the online platform.

- Domain adaptation comparisons are introduced out of the blue in page 7. Explained the results
using comparisons to other methods.

- Revise claim about an XAI tool: Sepsis former on its own is not an XAI tool. The analysis used
post-hoc XAI tools to understand contributors to predictions and relationships between inputs.

- The use of unsupervised clustering is better explain to find the subphenotypes, but I am still
confused about why there are two analyses for this, i.e., first to identify alpha and beta groups using
36 predictors and then CIS1 and CIS2 using only the seven coagulation-inflammation predictors?

As another reviewer mentioned, not all analyses have to be part of the manuscript because it
becomes hard to appreciate what are the main take aways.

- Regarding the radar plot in the assessment of HTEs: It seems like all predictors are included in
this analysis. I thought that the SMART model only used 8 predictors. Please clarify why variables
from the other categories were included and the purpose of this analysis. As another reviewer
mentioned, not all analyses have to be part of the manuscript because it becomes hard to appreciate
what are the main take aways.

Discussion:
Overall, the claims in the discussion are supported by the results: how explainability analyses were
done to study sepsis heterogeneity, how the different patient populations stratified by subphenotype
and risk levels had different clinical characteristics and mortality rates, and the insights from the
combined analysis of HTEs' effect.

The clarifying sentence at the end of the third paragraph of the discussion can be included in the
paragraph opening to mention the transition to a new analysis.

Included other works that have done subtyping, how the findings and techniques differ, and
acknowledge that comparisons are hard between identified subphenotypes.

I suggest beginning the paragraph in line 400 with this sentence before you contrast the results with
other works that have done subtyping: “Another strength is the ability to distinguish subphenotypes
and risk levels of septic patients using only coagulation-inflammatory markers and patient age” .
And then in the next paragraph discuss the effects of HTEs for the subgroups.

Minor comments:
Introduce some acronyms in the main paper for clarity: HTE in the introduction, MMID-SMOTE
in the results, the machine learning abbreviations (even if they are well known).

Line 425 “and so on” may not be clear for all readers and sounds informal.

**Reviewer #3 (Remarks on code availability):**

Even though the code is provided, the documentation can be improved and a step by step description
on what codes to run to generate what results. The README file does not have enough instructions
and does not mention which dependencies are needed. The names of the scripts can also be more
informative. For the transcriptomic analysis results, the folder only contains data files, so I wonder
if the code to recreate these results is missing.

**Response to Reviewers' Comments**

We are incredibly grateful to the reviewers for their insightful and constructive
comments, which have been very helpful in further improving the manuscript and
enhancing its robustness, logic, and presentation of work highlights. The following is
our detailed and thorough response to each comment one by one. We hope to address
your concerns and obtain your approval.

**Reviewer #1 (Remarks to the Author):**

In my previous review I have raised the following points:

- 1. The clarity and framing of the main message/ contribution of the paper
- 2. Issues relating to the reporting of the methodology and sample
- 3. Greater transparency towards the methods employed (e.g. datasets and codes)

Glad to share that these three issues were addressed in this latest revision.

The results of these paper are noteworthy as indicated in my earlier review and the
ability to translate the algorithm developed into a clinical profiling score has field
implications.

Good effort of showcasing the execution of the codes in the videos.

**Response:** Thank you for your valuable time and insightful suggestions, which have
helped us improve the quality of the article. We also appreciate your recognition of our
efforts.

**Reviewer #2 (Remarks to the Author):**

The authors have adequately addressed the concerns, and the manuscript is now suitable
for publication.

**Response:** We appreciate your approval of the revised manuscript, and we are
especially grateful for your constructive suggestions which have enhanced our
academic proficiency throughout this revision process.

**Reviewer #3 (Remarks to the Author):**

**Q1:** The main concerns were addressed in the revision, including the performance
comparisons to existing models, including both of the main model and the domain-
adaptation strategy. Implementation details and definitions were also provided for the
latter method, which helped to understand how it differs from existing techniques.
However, in the description of the domain adaptation in the supplementary method, the
justification that analyzed class imbalance in mortality outcomes is not theoretical but
rather empirical I think, since you based the analysis on your data samples. Further
methodological details were provided for the transcriptomic analysis and study of HTEs
effect. Ablation studies are also presented and clarifications around techniques used and
claims made throughout the manuscript.

**Response:** Thanks for your feedback on the revised manuscript. We are pleased to learn
that the main concerns have been addressed, and that the further methodological details
provided for the transcriptomic analysis, the study of HTEs effect, as well as the
ablation studies, have met your expectations.

The presence of class imbalance in mortality outcomes among sepsis patients is a
widely held consensus, substantiated by empirical analyses of clinical data. In this study,
we statistically analyzed the mortality rates within our utilized datasets, the results of
which corroborate this imbalance. It is crucial to emphasize that while analyses
constitutes an empirical investigation based on specific datasets, the methodology we
propose is not necessarily limited in its applicability to these datasets alone. We
appreciate you highlighting the need to clarify this. We revised this section to explicitly
characterize the analysis as empirical as following.

**Revised in the manuscript:**

1. “The presence of class imbalance in mortality outcomes among sepsis patients is a
widely held consensus, substantiated by empirical analyses of clinical data^{4, 5, 8}. ”

**(Line 107-109)**

2. Supplementary Method 1. Rationale for Domain Adaptation in Prognostic Modeling.

“Class imbalance in sepsis mortality outcomes is a well-established empirical

finding. In this study, the statistical chart of eICU, Local ICU, MIMIC III, and MIMIC
IV datasets exhibit significant class imbalance between positive samples (non-survivors)
and negative samples (survivors), with mortality rates of 25.38%, 37.23%, 25.56%, and
37.62% respectively. The number of non-survivors was significantly lower than that of
survivors.”

Moreover, to maintain manuscript conciseness, we omitted detailed enumeration
of supporting studies from the main text. Actually, the research we present here reports
mortality rates that are largely consistent with the findings of our study.

(1) A total of 14,607 sepsis patients were screened and the overall 30-day mortality rate
was 21.0%.

Ref: Cao, Bingbing, et al. Non-linear relationship between baseline mean arterial
pressure and 30-day mortality in patients with sepsis: a retrospective cohort study based
on the MIMIC-III database. *Annals of Translational Medicine* 10.16: 872 (2022).

(2) In cohorts of patients with suspected sepsis infection (training: n=12,473; validation:
n=8,256), the four sub-phenotypes exhibited distinct mortality rates: Group A (28%),
Group B (13%), Group C (32%), and Group D (27%).

Ref 4 in the manuscript. Bhavani SV, et al. Development and validation of novel sepsis
subphenotypes using trajectories of vital signs. *Intensive Care Med* 48, 1582-1592
(2022).

(3) In some countries, sepsis mortality is being reported at 20–30%.

Ref 5 in the manuscript. Hotchkiss RS, Moldawer LL, Opal SM, Reinhart K,
Turnbull IR, Vincent JL. Sepsis and septic shock. *Nat Rev Dis Primers* 2, 16045 (2016).

(4) In a cohort of 2,499 sepsis patients, mechanical ventilation was common within 24
214 hours of ICU admission, and in-hospital mortality was 30.1%.

Ref 8 in the manuscript. van Amstel RBE, et al. Uncovering heterogeneity in sepsis:
a comparative analysis of subphenotypes. *Intensive Care Med* 49, 1360-1369 (2023).

**Q2:** Besides, the authors extended the interpretation of the explainability analysis and
how this information is useful to understand the heterogeneity of the disease and

relevant biomarkers for sepsis prognosis. They also commented on the implications of
having a model that predicts risk-stratified levels and the corresponding analyses.

**Response:** Thanks for your thorough review and recognizing the enhancement we have
made to the explainability analysis and the discussions around disease heterogeneity,
sepsis prognostic biomarkers, as well as the implications of our risk-stratified prediction
model. We are pleased that these extensions have resonated with your review. Your
acknowledgment reinforces that this effort has been effective.

**Q3:** In the results, there is connection between findings and medical relevance (e.g.,
found importance of coagulation-inflammatory indicators vs functional indicators of
specific organs and how the former better reflects the pathophysiology of sepsis), and
an application use case of the tool available online was demonstrated.

**Response:** Thank you for recognizing both the connections between our findings and
their medical relevance, and the practical application of our online tool. We also
appreciate your acknowledgment of how this work bridges computational insights with
clinical applicability.

**Q4:** The paper still needs some organization edits, especially in the presentation of the
results:

- Some analyses are listed but then presented later. E.g., 1) the correlation between
predictors mentioned in line 145 and then presented in line 176, 2) finding a subset of
laboratory measurements within one category (part (a) in line 155) but the clustering
results of part (b) are presented first, followed by the correlation analysis.

**Response:** We are very grateful to you for the insightful comments on the organization
of our manuscript. We fully agree that a clear presentation is crucial for accurately
conveying our research findings. Following your valuable suggestions, we revised the
Results section to improve its narrative logic and readability, enabling readers to more
easily follow our analytical framework and findings.

As you correctly pointed out, the correlation analysis of predictors was mentioned
twice. This was not a repetition, but rather addressed two distinct stages of our analysis.

The first mention, with results presented in Fig. 2c, pertains to the correlation analysis
of the entire set of all 36 predictors. The second mention, with results in Fig. 2f, refers
to the correlation analysis conducted on the 8 core predictors. As Fig. 2f already
displays the correlation analysis results for the eight predictors, NO modifications were
made.

We've reordered 'the clustering results' to be Part (a) and 'finding a subset of
laboratory measurements within one category' to be Part (b) for better alignment with
our figures and tables.

The results of the Sankey diagram have been relocated.

**Revised in the manuscript:**

1. "interrelationships of all 36 predictors were explained via a correlation network
diagram, with Pearson correlation coefficients and corresponding p values (Fig. 2c)."

**(Line 150)**

3. to achieve two clinical goals: (a) Select an efficient unsupervised clustering
method.....; (b) Identify a sepsis mechanism relevant and clinically feasible subset of
laboratory measurements.....**(Line 161-164)**

4. The results of the Sankey diagram "The results consistently indicated that
coagulation-inflammatory indicators such as APTT, WBC, and the patient's age were
important predictors from various perspectives." have been relocated to **Line 207-208.**

5. "A clinically relevant scorecard (Table 1) was developed for SMART based on
medical knowledge, allowing clinicians to easily and directly calculate a patient's risk
score." **(Line 299).**

**Q5:** Improve the introduction of the transcriptomic-level explanations and
transcriptome analysis: It is clarified that this was to show the important roles of
coagulation-inflammation related indicators, but there are some aspects that can be
clarified. where was the gene information gathered from? Briefly mention the methods
involved in the paragraph in line 207.

**Response:** Thank you for your constructive feedback. We have revised the paragraph

in line 207 to clarify the gene information sources paragraph. These changes enhance
transparency and align with your suggestions. Thank you for guiding us to improve the
manuscript.

**Revised in the manuscript:**

“We retrieved the sepsis expression profile from the Gene Expression Omnibus (GEO,
<https://www.ncbi.nlm.nih.gov/geo/>), selecting dataset GSE65682 for analysis. DIC-
related genes were sourced from the GeneCards (<https://www.genecards.org/>) and
DisGeNET (<https://www.disgenet.org/>) databases. Genes overlapping between these
disseminated intravascular coagulation (DIC)-related gene sets and differentially
expressed genes (DEGs) were defined as DIC-related DEGs. To clarify their core
biological functions and underlying mechanisms, we performed KEGG and GO
enrichment analyses on these genes using the clusterProfiler package (version 3.14.3)
in R.” (Line 221-227)

**Q6:** It is still unclear to me how the SEVEN predictors were selected. In the model-
level explainability only 3 are mentioned (APTT, WBC, and age) and the importance
of coagulation-inflammation predictors, but there at least 10 from what I can see in
Figure 2c and 8 in Figure 2g. The overall selection process was illustrated in a figure in
the response letter, but I may be missing some details to fully understand the criteria
for picking those 7 variables. I also noticed that the functions of these 7 coagulation-
inflammation markers is discussed in detail in the paragraph starting in line 362 and
they are part of the online platform.

**Response:** Thank you for the helpful suggestion. We recognize that the justification for
our predictor selection could be stated more clearly. Accordingly, we have revised the
third paragraph of the 'Explainability analysis of coagulation-inflammatory
dysfunction' section. It now better explains why we selected 7 of the 10 predictors—
excluding Prothrombin Time (PT), basophils, and eosinophils—based on medical
knowledge and a correlation matrix analysis. To further improve clarity, we have also
elaborated on these points and added a summary sentence to conclude the paragraph.
We believe that we have adequately addressed this question.

Figure 2g's SHAP and Sankey diagram ranks APTT, WBC, and Age as the top three
predictors from various perspectives, and the other 8 sub-variables are also very
important. We report this ranking objectively—without implying selective inclusion.

**Revised in the manuscript:**

“Due to PT tests extrinsic, APTT intrinsic coagulation pathways and INR is a
standardized value calculated from PT results, we retained INR and APTT while
excluding PT. Meanwhile, basophils and eosinophils exhibit weak or negative
correlation coefficients with other predictors, rendering them impractical to reflect the
risk status of patients. These two types of WBC, influenced by external factors like
allergies, display inconsistent behavior and limited prognostic value in sepsis, with their
mechanisms remain unclear^{23,24}. To focus our analysis on the most robust and clinically
relevant markers, we excluded PT, basophils, and eosinophils. The final coagulation-
inflammatory markers analyzed in this study comprises 8 variables: APTT, INR,
lymphocytes, monocytes, neutrophils, WBC, PLT and age.” (Line 188-196)

**Q7:** Domain adaptation comparisons are introduced out of the blue in page 7. Explained
the results using comparisons to other methods.

**Response:** Thank you for your valuable feedback. In response to your comment, we
have revised the manuscript to provide a clearer context for the introduction of domain
adaptation techniques. Specifically, we have added an explanation regarding the
challenge of generalizing SepsisFormer across different cohorts due to heterogeneity in
patient characteristics. Additionally, we have clarified the comparison of MMID-
SMOTE with other techniques, highlighting its superior performance and the rationale
behind its selection for improving model generalization. We believe these changes
enhance the clarity and flow of the manuscript.

**Revised in the manuscript:**

“Despite strong performance within each cohort, SepsisFormer’s ability to
generalize across different clinical settings remains a key challenge, as patient
characteristics can differ significantly between datasets. To address this issue, we

incorporated methods to enhance the model’s adaptability, allowing it to adjust to these
differences and maintain high accuracy in new settings.

As shown in Fig. 2d and Supplementary Table 7, Maximum and Minimum Interval
Difference-based Synthetic Minority Oversampling Technique (MMID-SMOTE)
significantly outperformed other domain adaptation techniques and the ablation
experiment (no-adaptation baseline) by substantially improving the model's cross-
cohort generalization capability. MMID-SMOTE performed better than other
approaches across multiple evaluation metrics, including AUC, accuracy, sensitivity,
specificity, and F1-score. This is demonstrated through an ablation study, which shows
that MMID-SMOTE outperformed the no-adaptation baseline and other state-of-the-art
methods. The Mean-teacher and Whitening methods showed poorer performance, with
Mean-teacher struggling with low-quality pseudo-labels and Whitening’s adjustments
failing to improve generalization. In contrast, Moment Matching showed second-place
performance by focusing on second-order statistics, but still did not surpass MMID-
SMOTE in any metric. This highlights the importance of selecting the right method for
improving model performance across diverse clinical settings. Our findings show that
MMID-SMOTE improves SepsisFormer’s ability to predict outcomes and adapt to new
clinical environments. As a post-hoc XAI model, SepsisFormer offers reliable and
interpretable support for clinical decision-making, utilizing a minimal set of routine
biomarkers.” (Line 254-271)

**Q8:** Revise claim about an XAI tool: Sepsis former on its own is not an XAI tool. The
analysis used post-hoc XAI tools to understand contributors to predictions and
relationships between inputs.

**Response:** Thank you for your insightful and constructive comment. Your keen
observation has prompted us to reconsider the precision of the terminology used in our
manuscript, and we are grateful for this opportunity to improve its quality. We fully
agree with you, SepsisFormer is not, in itself, an XAI tool. The explainability presented
in our paper is achieved by applying post-hoc analytical techniques.

We would like to clarify that our work establishes an **analytical framework** that

couples the high-performance predictive with post-hoc XAI methodologies. In the XAI
field, it is common practice to refer to this "model + explanation" composite as a unified
whole. This convention aims for concise and clear communication, emphasizing that
the model has been rendered interpretable.

Our proposed terminology is well-grounded in the current literature. A key
distinction in the XAI field is drawn between ante-hoc and post-hoc paths to
interpretability (Retzlaff et al., 2024). Ante-hoc refers to using inherently transparent
models (e.g., linear regression, decision trees mentioned in the INTRODUCTION),
whereas post-hoc methods are specifically designed to explain already-trained black-
box models. Our approach, which falls into the latter category, aims to confer
explainability upon an opaque model. Therefore, designating the resulting model-
explanation entity as a "**post-hoc XAI model**" is both appropriate and logical.

This practice is supported by precedent in top-tier journals. For instance, Lauritsen
et al. (2020) in Nature Communications refer to their system, holistically as an
"explainable AI early warning score (xAI-EWS) system for sepsis." This precedent
provides strong support for our rationale in referring to the combination of
SepsisFormer and post-hoc analysis as "**post-hoc XAI model SepsisFormer.**"
Furthermore, a significant body of literature is dedicated to systematically surveying
(Bhati et al., 2025) and evaluating (Bello et al., 2025; Belaid et al., 2022) various post-
hoc XAI methods. The shared premise of these studies is that applying such methods
allows an otherwise opaque model to be effectively explained, thereby becoming an
examinable "explainable system."

In response to your valuable feedback and to enhance the manuscript's precision, we
will implement the following revisions:

1. Upon its first introduction, we will explicitly state that SepsisFomer is a predictive
model and that we employ post-hoc XAI techniques to interpret it.

**Revised in the manuscript:**

"we developed a Transformer-based prognostic model (SepsisFormer) interpreted
via post-hoc XAI techniques,....." (Line 111-112)

2. Throughout the remainder of the manuscript, we will consistently use the more

precise term "**post-hoc XAI model SepsisFormer**".

We are confident that these revisions will make our manuscript clearer, more accurate,
and fully aligned with the current academic standards in the XAI field. Thank you again
for your professional guidance.

Reference

[1] Bhati, Deepshikha, et al. "A Survey of Post-Hoc XAI Methods from a Visualization
Perspective: Challenges and Opportunities." IEEE Access (2025).

[2] Retzlaff, Carl O., et al. "Post-hoc vs ante-hoc explanations: xAI design guidelines
for data scientists." Cognitive Systems Research 86 (2024): 101243.

[3] Bello, Marilyn, et al. "The level of strength of an explanation: A quantitative
evaluation technique for post-hoc XAI methods." Pattern Recognition 161 (2025):
111221.

[4] Belaid, Mohamed Karim, et al. "Do we need another explainable AI method?
Toward unifying post-hoc XAI evaluation methods into an interactive and multi-
dimensional benchmark." arXiv preprint arXiv:2207.14160 (2022).

[5] Lauritsen, Simon Meyer, et al. "Explainable artificial intelligence model to predict
acute critical illness from electronic health records." Nature communications 11.1
(2020): 3852.

**Q9:** The use of unsupervised clustering is better explain to find the subphenotypes, but
I am still confused about why there are two analyses for this, i.e., first to identify alpha
and beta groups using 36 predictors and then CIS1 and CIS2 using only the seven
coagulation-inflammation predictors? As another reviewer mentioned, not all analyses
have to be part of the manuscript because it becomes hard to appreciate what are the
main take aways.

**Response:** We sincerely thank you for this insightful comment and for highlighting a
crucial aspect of our analytical workflow. We understand your concern regarding the
clarity and potential redundancy of the two-part analysis. In our first revision, we added
Supplementary Figures 1 and 2 to explain our rationale. We'd like to further elaborate

on the distinct, sequential logic behind our approach, as we believe it's fundamental to
the rigor and interpretability of our findings.

Our two-step clustering process was not designed as two separate attempts to
identify subphenotypes, but rather as a **principled, data-driven framework moving**
**from broad, unbiased exploration to a focused, clinically explainable model.**

**Step 1: Unbiased, Hypothesis-Generating Exploration (Alpha and Beta Clusters)**

The primary purpose of the first analysis, using all 36 predictors, was to serve as a
crucial exploratory and hypothesis-generating step. Our objectives here were threefold:

1. To avoid a priori bias. By starting with a comprehensive set of 35 predictors
derived from the Sepsis-3 criteria (SOFA based), we allowed the data itself to reveal
the most dominant biological signals driving patient heterogeneity, rather than pre-
supposing the importance of a specific pathway (**Line 138-142**). A direct analysis using
only seven coagulation-inflammatory markers could have been criticized for being
subjective. This initial step provides an objective, data-driven justification for our
subsequent focus on coagulation and inflammation.

2. To establish analytical feasibility. This analysis confirmed that meaningful,
distinct patient clusters (alpha and beta) did exist within our dataset and that the
Gaussian Mixture Model (GMM) was a suitable method for the identification. See
“This study employs GMM for subphenotype identification, as the patient subgroups
derived from GMM show the most significant differences in mortality rates (shown in
Section Explainability Analysis), indicating greater clinical relevance.” (**Line 275-277**)

3. To identify the most informative features. The key output of this step was the
discovery that coagulation-inflammatory markers were the most significant drivers
distinguishing the alpha and beta groups.

**Step 2: Definitive, Interpretable Subphenotype Modeling (CIS1 and CIS2)**

The findings from Step 1 directly informed our primary, definitive analysis. We
proceeded with a focused model using only the seven key coagulation-inflammatory
predictors for the following critical reasons:

1. To enhance interpretability and clinical utility. A subphenotype defined by a
parsimonious set of seven mechanistically-linked markers is far more interpretable and

clinically actionable than a "black-box" phenotype derived from a complex
combination of 36 variables. This allows clinicians to understand the underlying
pathophysiology and potentially use these markers for patient stratification.

2. To increase model robustness and reduce noise. Using all 36 predictors risks
falling into the "curse of dimensionality," where noise from less relevant variables can
obscure the true signal and lead to less stable clusters. By concentrating on the most
informative markers identified in Step 1, we use a more robust and dimensionally
reduced model that is more likely to be generalizable (Fig3a).

3. To define biologically coherent subphenotypes. The final subphenotypes, CIS1
and CIS2, are not just statistically derived clusters; they are defined by a core biological
process, which is a well-recognized and critical component of sepsis pathophysiology.
This biological coherence is the core strength of our findings (Fig3).

In summary, the first analysis provides the necessary justification for the second,
moving from a wide, unbiased search to a deep, precise conclusion. We contend that
removing the first step would weaken the scientific foundation of our work, as it would
obscure the data-driven rationale for selecting final set of predictors.

To address your concern about clarity, we have substantially revised the **Methods**
and **Results** sections of our manuscript. We have explicitly framed the two analyses as
"**exploratory clusters**" and "**definitive subphenotypes**" and have detailed the rationale
presented above. We believe these revisions now make the logic of our analytical
strategy transparent, allowing the reader to appreciate the main takeaways of our study.

**Q10:** Regarding the radar plot in the assessment of HTEs: It seems like all
predictors are included in this analysis. I thought that the SMART model only used 8
predictors. Please clarify why variables from the other categories were included and the
purpose of this analysis. As another reviewer mentioned, not all analyses have to be
part of the manuscript because it becomes hard to appreciate what are the main take
aways.

**Response:** Thank you for this important comment. You are absolutely correct that the
SMART model includes only 8 selected predictors. The radar plot (Figure 4b), however,

was not derived from the model but was constructed as a descriptive visualization to
show how different physiological systems - such as hepatic, renal, and inflammatory—
vary across risk levels.

In clinical research, such radar plots are particularly valuable as they allow for
integrated visualization of multi-system involvement, which is often essential when
evaluating complex conditions involving systemic dysfunction, such as sepsis, multi-
organ failure, or advanced chronic disease. In our context, the plot helps reveal
consistent patterns of systemic deterioration across risk levels, providing additional
clinical insight beyond the predictors used in the model.

The intent was not to imply that these additional variables were involved in risk
stratification or predictive modeling, but to provide clinicians with a more intuitive
understanding of the biological profiles associated with varying levels of treatment
benefit. In this context, the radar plot serves a purely exploratory and interpretive role.
To maintain focus in the main manuscript, we have clarified the purpose of the radar
plot and retained it in the main figures due to its interpretive value.

Furthermore, we presented similar figures from other studies to demonstrate the
utility of radar plots in the context of heterogeneity analysis within the biomedical field.
For instance, refer to Figure 2c in Xue, Ruidong, et al. "Liver tumour immune
microenvironment subtypes and neutrophil heterogeneity." *Nature* 612.7938 (2022):
141-147.

[REDACTED]

**Revised in the manuscript:**

“As shown in Fig. 4b, the radar plot demonstrates that risk stratification based solely
on coagulation-inflammatory markers captures a broader pattern of multi-organ
dysfunction. Except for the erythrocyte category, laboratory values across all other

systems increase consistently with risk level, peaking in the dangerous group. The
consistent upward trend supports the use of coagulation-inflammatory markers as
indicators of systemic severity and their utility in clinical risk stratification.”

**(Line 335-340)**

**Q11:** Discussion:

Overall, the claims in the discussion are supported by the results: how explainability
analyses were done to study sepsis heterogeneity, how the different patient populations
stratified by subphenotype and risk levels had different clinical characteristics and
mortality rates, and the insights from the combined analysis of HTEs’ effect.

**Response:** Thank you. We appreciate you recognizing that our discussion's claims are
supported by the results.

**Q12:** The clarifying sentence at the end of the third paragraph of the discussion can be
included in the paragraph opening to mention the transition to a new analysis.

**Response:** We've revised the discussion as suggested. The clarifying sentence
“Although transcriptomic data are not direct items for SMART scores and CIS typing,
they further enhance the explanation of the importance of coagulation-inflammatory
markers in diagnosis and prognosis assessment.” has been moved to the beginning of
the third paragraph **(Line 404-406)** to clearly signal the transition to the new analysis.

**Q13:** Included other works that have done subtyping, how the findings and techniques
differ, and acknowledge that comparisons are hard between identified subphenotypes.

**Response:** Thanks for your feedback. We've fully addressed your comments by adding
the requested discussion on other subtyping works, detailing how our findings and
techniques differ, and acknowledging the challenges in comparing identified
subphenotypes. We're pleased to confirm that these revisions have met your satisfaction.

**Q14:** I suggest beginning the paragraph in line 400 with this sentence before you
contrast the results with other works that have done subtyping: “Another strength is

the ability to distinguish subphenotypes and risk levels of septic patients using only
coagulation-inflammatory markers and patient age” . And then in the next paragraph
discuss the effects of HTEs for the subgroups.

**Response:** We really appreciate your insightful suggestion on strengthening the
framing of our findings. We've now incorporated the proposed sentence. This addition,
placed right before our comparative discussion with existing subtyping literature,
significantly sharpens the focus on our contribution.

Dear Professor, please forgive us. It might be due to the WORD version or other reasons
that the line numbers you provided are slightly different from the actual line numbers
in our WORD document. However, we have understood your comments and made the
necessary revisions at the appropriate places.

**Revised in the manuscript:**

1. “Another strength is the ability to distinguish subphenotypes and risk levels of septic
patients using only coagulation-inflammatory markers and patient age.” (Line 424-425)

2. “The anticoagulant drug heparin may offer potential benefits in sepsis management,
with clinical outcomes varying across different risk level and subphenotypes.”
(Line 448-449)

**Q15:** Minor comments:

Introduce some acronyms in the main paper for clarity: HTE in the introduction,
MMID-SMOTE in the results, the machine learning abbreviations (even if they are well
known).

Line 425 “and so on” may not be clear for all readers and sounds informal.

**Response:** We're grateful for your insightful feedback. Your comments significantly
improved our manuscript's clarity and readability. We've carefully addressed all your
suggestions, including the minor ones, and believe these revisions have truly
strengthened the paper. We appreciate the time and effort you dedicated to our work.

**Revised in the manuscript:**

1. “HTE refers to the phenomenon where the same treatment can have different effects
(beneficial, neutral, or even harmful) on different patients.” (Line 63-64)

- 2. “Models mainly include machine learning approaches (Decision Tree, Logistic
Regression, Support Vector Machine, Generalized Linear Model, Naïve Bayes) and
neural networks (Multilayer Perceptron, Long Short-term Memory, Convolutional
Neural Network, Gated Recurrent Unit, and two attention-based explainable models:
RETAIN and Dipole).” (Line 87-90)
- 3. “Maximum and Minimum Interval Difference-based Synthetic Minority
Oversampling Technique (MMID-SMOTE) significantly”(Line 258-259)
- 4. “Twenty-nine variables, including cardiovascular..... pulmonary, and renal systems
were used.” (Line 433)

**Reviewer #3 (Remarks on code availability):**

Even though the code is provided, the documentation can be improved and a step by
step description on what codes to run to generate what results. The README file does
not have enough instructions and does not mention which dependencies are needed.
The names of the scripts can also be more informative. For the transcriptomic analysis
results, the folder only contains data files, so I wonder if the code to recreate these
results is missing.

**Response:** We sincerely thank you for your valuable and constructive feedback, which
has significantly helped us improve the clarity and reproducibility of our work.

In response to your suggestions, we have comprehensively revised the README.md
file to include a detailed step-by-step guide that walks users through the entire process
of generating our results. To address the lack of dependency information, we have
added a dedicated installation section to simplify the environment setup process. We
also agree that the script names could be more informative, we've updated README
file in <https://github.com/zhuli19031218/SepsisFormer/blob/main/README.md>. To
facilitate quick access to the information, we have included the README file in the
Appendix of this reply letter.

Furthermore, we apologize for the oversight regarding the transcriptomic analysis
code; we have now added a new `transcriptomics_analysis` folder containing all

the necessary four R scripts, six datasets and all videos to reproduce those findings.
(https://github.com/zhuli19031218/SepsisFormer/tree/main/transcriptomics_analysis;
<https://doi.org/10.5281/zenodo.15634368> Version3.0).

The updated files on these websites include:

(1) 4 R scripts for the analysis of diagnostic markers;
(2) 6 datasets, which had already been provided in the previous revised version,
including: GEO-GSE26440 (diagnostic markers),GEO-GSE54514 (prognostic
markers),GEO-GSE65682 (diagnostic markers),GEO-GSE65682 (prognostic
markers),GEO-GSE95233 (diagnostic markers),LocalICUNB-qPCR
(3) 24 Code Execution Videos, including: 3h_1_Forest plot, 3h_2_Diagnostic ROC,
3h_3_Risk factor graph and survival curve, S6a_volcano plot, S6b_Venn Diagram,
S6c_The PPI network of intersection genes, S6d_heat map, S6e_GO biological
pathway, S6fgh, S6i_Lasso regression, S7a_Volcano plot,S7b_GSEA enrichment
analyses, S7c_Venn diagram, S7d_The PPI network of intersection genes,
S7e_Lasso regression, S7f_Forest plot, S7g_ROC curves, S7h_nomogram,
S7i_ROC curves, S7j_Calibration curve, S7k_DCA curve, S8_The prognostic and
diagnostic efficacy in external cohort, S9_1_Group comparison and ROC curve,
S9_2_Group comparison and ROC curve.

We are confident that these extensive revisions have made our code repository far
more user-friendly, transparent, and reproducible. We are also grateful for the
opportunity to make these improvements.

Our team is deeply grateful for your constructive suggestions and well-researched
recommendations. Your expert feedback is essential to strengthening our manuscript.

Wishing you all the best.

zhuli19031218 / SepsisFormer

<> Code

Issues

Pull requests

Actions

Projects

Wiki

Security

main

SepsisFormer / README.md

zhuli19031218 Update README.md

a0058fb · 18 minutes ago

471 lines (336 loc) · 16.3 KB

SepsisFormer and SMART

This repository contains the implementation of the paper "Explainable AI-driven heterogeneity using coagulation–inflammatory markers improves prognosis prediction, risk stratification, and anticoagulant treatment effects for sepsis". The code includes all major data, models, algorithms, and experimental settings described in the paper, enabling researchers to reproduce and verify our results. Enjoy~~~

Sepsis, a leading cause of hospital mortality, is characterized by substantial heterogeneity, hindering the development of effective and interpretable prognostic and stratification methods. To address this challenge, we developed an explainable prognostic model (SepsisFormer, a transformer-based deep neural network with an enhanced domain-adaptive generator) and an automated risk stratification tool (SMART, a scorecard consistent with medical knowledge). In a multi-center retrospective study of 12,408 sepsis patients, SepsisFormer achieved high predictive accuracy (AUC: 0.9301, sensitivity: 0.9346, and specificity: 0.8312). SMART (AUC: 0.7360) surpassed most established scoring systems. Based on SMART, four risk levels (mild, moderate, severe, dangerous) can be identified by using seven coagulation-inflammatory routine laboratory measurements and patient age, and the corresponding mortality is approximately 5%, 15%, 30%, and 50%, respectively. Meanwhile, two subphenotypes (CIS1 and CIS2) can be classified through unsupervised GMM, and CIS2 has a worse survival prognosis. Notably, patients with moderate/severe levels or CIS2 derive more significant benefits from anticoagulant treatment. In conclusion, explainable artificial intelligence (SepsisFormer) drives risk stratification and subphenotypic classification of sepsis, which help to guide anticoagulant treatment. Our work, therefore, offers a novel set of simple, real-time executable tools for sepsis heterogeneity, demonstrating considerable potential to significantly enhance sepsis clinical practice globally, particularly in resource-constrained healthcare settings.

NOTE: The Source Data for the 36 and 8 sepsis markers is provided in the "figure 2ab" folder, which contains the datasets used for Figures 2a, 2b, and 2e. The underlying code for these figures is identical; however, the input datasets differ: Figures 2a and 2b use 36 sepsis markers, while Figure 2e uses 8 sepsis markers.

Code Guide

This repository contains comprehensive code and data for generating all figures presented in the manuscript. This documentation provides detailed step-by-step instructions to facilitate the reproduction of all results reported in the study.

Table of Contents

- Environment Requirements
- Project Structure
- Execution Procedures
 - Figure 2a2b: Multi-Model Comparative Analysis
 - Figure 2c: Network Analysis Visualization
 - Figure 2d: Domain Adaptation Analysis
 - Figure 2f: Clustering Visualization
 - Figure 2g: SHAP Feature Importance Analysis
 - Figure 3c3h: Hierarchical Clustering Analysis
 - Figure 3d: 3D Visualization
 - Figure 4a4c: Cox Proportional Hazards Model
 - Figure 4b: Radar Chart Analysis
 - Figure 4d: Risk Ratio Analysis

Environment Requirements

Python Dependencies

```
pip install numpy pandas scikit-learn torch torchvision
pip install matplotlib seaborn plotly networkx
pip install shap shapely scipy statsmodels
pip install jupyter notebook
pip install igraph python-igraph
pip install lifelines
```

Core Dependency Versions

- Python \geq 3.7
- PyTorch \geq 1.8.0
- scikit-learn \geq 0.24.0
- pandas \geq 1.3.0
- numpy \geq 1.21.0

Optional Software

- **GraphPad Prism:** For statistical chart generation in Figure 4a4c
- **Origin:** For clustering visualization charts in Figure 2f
- **R Language:** For statistical analysis (if using R scripts)

Project Structure

```

figure/
├── 2a2b/                # Figure 2a2b: Multi-model comparative
analysis
│   ├── model2/         # Deep learning model code
│   │   ├── Transformer.py # Transformer model implementation
│   │   ├── model_LSTM.py # LSTM model implementation
│   │   ├── model_GRU.py  # GRU model implementation
│   │   ├── model_gpt.py  # GPT model implementation
│   │   ├── train.py      # Training script
│   │   ├── main_mimic.py # Main execution script
│   │   └── results/      # Model results
│   ├── data/           # Data files
│   └── ml_logs/        # Machine learning logs
├── 2c/                 # Figure 2c: Network analysis visualization
│   ├── data/           # Node and edge data
│   ├── log/            # Execution logs
│   └── video/          # Generated videos
├── 2d/                 # Figure 2d: Domain adaptation analysis
│   ├── models/         # Machine learning model code
│   ├── logs/           # Experimental results
│   ├── roc_domain_adaptation.ipynb
│   └── video/
├── 2f/                 # Figure 2f: Clustering visualization
│   ├── Chord Diagram/  # Chord diagram
│   ├── Heatmap/        # Heatmap
│   └── Radar Chart/    # Radar chart
├── 2g/                 # Figure 2g: SHAP analysis
│   ├── data/           # Data files
│   ├── logs/           # Model logs
│   ├── SepsisFormer_shap_8.py
│   ├── shap_8 (2).ipynb
│   └── video/
├── 3c3h/              # Figure 3c3h: Hierarchical clustering
│   └── data/           # Data files

```

```

|   ├── log/                # Clustering logs
|   └── video/              # Clustering videos
├── 3d/                     # Figure 3d: 3D visualization
|   ├── data/              # Data files
|   ├── 3d.ipynb           # 3D visualization code
|   └── video/
├── 4a4c/                   # Figure 4a4c: Cox model
|   ├── 4a/                # New scoring analysis
|   ├── 4c/                # Subphenotype analysis
|   └── *.ipynb            # Cox analysis code
├── 4b/                     # Figure 4b: Radar chart
|   ├── data/              # Radar chart data
|   └── video/
└── 4d/                     # Figure 4d: Risk ratio
    ├── 风险比.ipynb
    └── video/

```

Execution Procedures

Figure 2a2b: Multi-Model Comparative Analysis

Objective: Compare the performance of different deep learning models (Transformer, LSTM, GRU, GPT) on sepsis prediction tasks

Primary Files:

- 2a2b/model2/main_mimic.py : Main execution script
- 2a2b/model2/train.py : Training and evaluation script
- 2a2b/model2/Transformer.py : Transformer model implementation
- 2a2b/model2/model_LSTM.py : LSTM model implementation
- 2a2b/model2/model_GRU.py : GRU model implementation
- 2a2b/model2/model_gpt.py : GPT model implementation
- 2a2b/data/SepsisFormer.py : SepsisFormer model implementation

Data Files:

- 2a2b/data/8/ : Dataset with 8 features
- 2a2b/data/36/ : Dataset with 36 features

Execution Steps:

1. Environment Preparation:

```
cd 2a2b/model2
```

2. Train Transformer Model:

```
python main_mimic.py --model_name Transformer --factors 8 --lr 0.004 --
```

3. Train LSTM Model:

```
python main_mimic.py --model_name Lstm --factors 8 --lr 0.004 --epoch 5
```

4. Train GRU Model:

```
python main_mimic.py --model_name GRU --factors 8 --lr 0.004 --epoch 50
```

5. Train GPT Model:

```
python main_mimic.py --model_name GPT --factors 8 --lr 0.004 --epoch 50
```

Model Parameter Specifications:

- `--model_name` : Select model type (Transformer/Lstm/GRU/GPT)
- `--factors` : Number of input features (8 or 36)
- `--lr` : Learning rate
- `--epoch` : Number of training epochs
- `--batch_size` : Batch size
- `--pretrain` : Whether to use pre-trained models
- `--loadmodel` : Path to pre-trained model

Model Architecture:

- **Transformer**: Multi-head attention mechanism, 8-layer depth
- **LSTM**: Long Short-Term Memory network, 2-4 layers
- **GRU**: Gated Recurrent Unit, 2-4 layers
- **GPT**: Generative Pre-trained Transformer

Output Results:

- Model performance metrics (AUC, Accuracy, F1-score, MCC)
- Training logs and TensorBoard visualization
- ROC curve plots

- Model comparison result tables

Expected Performance (based on 8 features):

- **GRU:** AUC ~0.645, Accuracy ~0.611
- **LSTM:** AUC ~0.641, Accuracy ~0.611
- **Transformer:** AUC ~0.630, Accuracy ~0.600
- **GPT:** Performance pending evaluation

Figure 2c: Network Analysis Visualization

Objective: Generate network relationship diagrams among sepsis patient features

Generation Method: Online generation using HiPlot website

Data Files:

- `2c/data/节点数据.csv` : Network node information
- `2c/data/连线数据-*.csv` : Edge data at different thresholds

Execution Steps:

1. Access HiPlot website (<https://hiplot.com.cn/>)
2. Upload node data and edge data files
3. Select network graph visualization type
4. Configure parameters:
 - Node label column: `media`
 - Node color column: `weight`
 - Node size column: `media.type`
 - Edge width column: `weight`
 - Layout style: Circular layout
5. Generate network analysis diagram

Parameter Specifications:

- Node label column: `media`
- Node color column: `weight`
- Node size column: `media.type`
- Edge width column: `weight`
- Layout style: Circular layout

Output: Network analysis video files

Reference: Please refer to the operation steps in `2c/video/Network_Igraph.mp4`

Figure 2d: Domain Adaptation Analysis

Objective: Analyze domain adaptation effects between different data sources

Primary Files:

- `2d/models/domain_adaptation.py` : Domain adaptation algorithm implementation
- `2d/models/随机森林.py` : Random Forest model
- `2d/models/逻辑回归.py` : Logistic Regression model
- `2d/roc_domain_adaptation.ipynb` : ROC analysis

Execution Steps:

1. Prepare source and target domain data
2. Execute domain adaptation algorithm:

```
cd 2d/models
python domain_adaptation.py
```

3. Train machine learning models:

```
python 随机森林.py
python 逻辑回归.py
python SVM.py
python SGD.py
```

4. Execute ROC analysis:

```
jupyter notebook roc_domain_adaptation.ipynb
```

Domain Adaptation Methods:

- `mean_teacher` : Mean teacher method
- `whitening` : Whitening method

Output: ROC curve plots and domain adaptation effect comparison

Figure 2f: Clustering Visualization

Objective: Generate multiple visualization charts for clustering results

Generation Method: Generated using Origin software

Content Included:

- **Chord Diagram** (`Chord Diagram/`): Display relationships between clusters
- **Heatmap** (`Heatmap/`): Feature correlation heatmap

- **Radar Chart** (Radar Chart/): Cluster feature radar chart

Data Files:

- 2f/Chord Diagram/data/雷达图与和弦图数据.xlsx
- 2f/Heatmap/data/36.xlsx
- 2f/Radar Chart/*.oggu : Origin project files

Execution Steps:

1. Open .oggu files using Origin software
2. Import corresponding data files
3. Generate chord diagram to display cluster relationships
4. Calculate feature correlations and generate heatmap
5. Create radar chart to display cluster features

Output: Chord diagram, heatmap, and radar chart video files

Software Requirements: Origin software

Figure 2g: SHAP Feature Importance Analysis

Objective: Analyze feature importance of SepsisFormer model using SHAP method

Primary Files:

- 2g/SepsisFormer_shap_8.py : SepsisFormer model implementation
- 2g/shap_8 (2).ipynb : SHAP analysis code

Execution Steps:

1. Load pre-trained SepsisFormer model
2. Prepare test data
3. Execute SHAP analysis:

```
cd 2g
jupyter notebook "shap_8 (2).ipynb"
```

Model Architecture:

- Transformer-based architecture
- Multi-head attention mechanism
- Feed-forward neural network

Output: SHAP feature importance plots and interpretability analysis

Figure 3c3h: Hierarchical Clustering Analysis

Objective: Perform hierarchical clustering analysis on sepsis patients

Data Files:

- 3c3h/data/mimic4_level_subphenotype_heparin.csv

Execution Steps:

1. Load patient data
2. Execute hierarchical clustering algorithm
3. Generate clustering results at different levels
4. Visualize clustering dendrogram

Clustering Methods:

- Hierarchical Clustering
- Subphenotype Clustering

Output: Hierarchical clustering dendrogram and subphenotype analysis videos

Figure 3d: 3D Visualization

Objective: Generate 3D visualization of sepsis patient features

Primary Files:

- 3d/3d.ipynb : 3D visualization code
- 3d/a.html : Interactive 3D chart

Execution Steps:

1. Load patient data
2. Execute 3D visualization:

```
cd 3d
jupyter notebook 3d.ipynb
```

Visualization Types:

- 3D scatter plots
- Interactive 3D charts
- Feature space visualization

Output: 3D visualization plots and interactive HTML files

Figure 4a4c: Cox Proportional Hazards Model

Objective: Analyze prognostic prediction capabilities of different scoring systems and subphenotypes

Generation Method: Generated using GraphPad Prism software

Primary Files:

- 4a4c/cox比例_mimic4_heparin.ipynb : Heparin-related Cox analysis
- 4a4c/cox比例_mimic4_8_subphenotype_level1.ipynb : Subphenotype Cox analysis

main SepsisFormer / README.md ↑ Top

Preview

Code

Blame

Raw

- 4a4c/score_hierarchy_mimic4_*.csv : Scoring data at different levels

Execution Steps:

1. Execute Cox analysis to obtain data:

```
cd 4a4c
jupyter notebook cox比例_mimic4_heparin.ipynb
jupyter notebook cox比例_mimic4_8_subphenotype_level1.ipynb
```

2. Use GraphPad Prism software:

- Open 4a4c/4a/新评分.pzfx file
- Open 4a4c/4c/亚表型.pzfx file
- Import analysis result data
- Generate statistical charts

Statistical Methods:

- Cox proportional hazards model
- Survival analysis
- Risk stratification

Output: Cox regression result plots and statistical reports

Software Requirements: GraphPad Prism (<https://www.graphpad.com/features>)

Figure 4b: Radar Chart Analysis

Objective: Generate radar charts for multi-dimensional features

Data Files:

- 4b/data/雷达图_环状注释11.24.R : R script data

Execution Steps:

1. Execute radar chart script using R language
2. Generate circular annotation radar chart

Output: Radar chart PDF files and videos

Figure 4d: Risk Ratio Analysis

Objective: Calculate and visualize risk ratios for different factors

Primary Files:

- 4d/风险比.ipynb : Risk ratio calculation code

Execution Steps:

1. Calculate risk ratios for various factors
2. Generate risk ratio forest plot:

```
cd 4d
jupyter notebook 风险比.ipynb
```

Output: Risk ratio forest plot and statistical reports

Result Files

Each figure folder contains the following types of output files:

- **Video Files** (video/): Dynamic visualization results
<https://doi.org/10.5281/zenodo.15634368>
- **Log Files** (log/): Execution logs and intermediate results
- **Data Files** (data/): Raw and preprocessed data

Important Notes

1. **Data Paths:** Ensure all data file paths are correct
2. **Dependency Versions:** Recommend using specified dependency versions
3. **Computational Resources:** Some analyses (e.g., SHAP) require substantial computational resources
4. **Memory Requirements:** Large datasets may require sufficient memory

Technical Support

If encountering issues during execution, please check:

1. Python environment and dependency package versions
2. Data file paths and formats
3. Error messages in execution logs

**Explainable AI Unravels Sepsis Heterogeneity via**
**Coagulation-Inflammation Profiles for Prognosis and Stratification**

Corresponding Author: Niu Bailin

Version 3:

**Reviewer #3 (Remarks to the Author):**

Overall, the revision provides satisfactory responses to the comments I included
in the previous iteration. Language clarifications were made in the method's
description and justifications (for domain adaptation, correlation analyses, clustering,
and explainability). Improvements were made in the organization of the results. The
paper now explains the selection of variables for risk models and other analyses.

**Response:** We sincerely thank you for your positive evaluation of our revision. We
are glad that the clarifications and improvements we made in the methodology,
justifications, and results organization have addressed your comments, and we truly
appreciate your recognition of our efforts.

**Reviewer #3 (Remarks on code availability):**

The documentation of the repository has been improved with more clear
instructions on how to run the code.

**Response:** We are truly grateful for your acknowledgment. Thank you very much for
recognizing our improvements in the repository documentation and the clearer
instructions provided for running the code.

Our team is deeply grateful for your feedback. Best wishes.